# CONCEPT-DRIVEN OFF POLICY EVALUATION

## ABSTRACT

Evaluating off-policy decisions using batch data poses significant challenges due to high variance and limited sample sizes, making reliable evaluation difficult. To improve Off-Policy Evaluation (OPE) performance, we must identify and address the sources of this variance. Recent research on Concept Bottleneck Models (CBMs) shows that using human-explainable concepts can improve predictions and provide better understanding. We propose incorporating concepts into OPE to reduce variance through targeted interventions. Shared disease characteristics, for example, could help identify better treatment options, despite variations in patient vitals. Our work introduces a family of concept-based OPE estimators, proving that they remain unbiased and reduce variance when concepts are known and predefined. Since real-world applications often lack predefined concepts, we further develop an end-to-end algorithm to learn interpretable, concise, and diverse parameterized concepts optimized for variance reduction. Our experiments with synthetic and real-world datasets show that both known and learned concept-based estimators significantly improve OPE performance. Crucially, we show that, unlike other OPE methods, concept-based estimators are easily interpretable and allow for targeted interventions on specific concepts, further enhancing the quality of these estimators.

## 1 INTRODUCTION

[1]. In domains like healthcare, education, and public policy, where interacting with the environment can be risky, prohibitively expensive, or unethical (Sutton & Barto, 2018; Murphy et al., 2001; Mandel et al., 2014), estimating policy value from batch data before deployment is essential for the practical application of RL. OPE aims to estimate the effectiveness of a specific policy, known as the evaluation or target policy, using data collected beforehand from a different policy, referred to as the behavior policy.

OPE has been widely studied to determine when an evaluation policy outperforms a behavior policy (e.g., Komorowski et al. (2018a); Precup et al. (2000); Thomas & Brunskill (2016); Jiang & Li (2016)). Importance sampling (IS) methods adjust for distributional mismatches between behavior and target policies by reweighting historical data, yielding generally unbiased and consistent estimates (Precup et al., 2000). Despite their desirable properties (Thomas & Brunskill, 2016; Jiang & Li, 2016; Farajtabar et al., 2018), IS methods often face high variance, especially with limited overlap between behavioral samples and evaluation targets or in data-scarce conditions. Evaluation policies may outperform behavior policies for specific individuals or subgroups (Keramati et al., 2021b), making it misleading to rely solely on aggregate policy value estimates. While causal inference approaches (e.g., Athey et al. (2019); Nie & Wager (2021)) explore individual-level outcome differences under evaluation versus behavior policies, they do not address sequential settings. Work such as Keramati et al. (2021b) has shown that with predefined groups, certain OPE estimators can yield more accurate evaluations. However, in practice, these groups are often unknown, prompting the need for methods to learn interpretable characterizations of the circumstances where the evaluation policy benefits certain individuals over others.

In this paper, we propose performing OPE using interpretable concepts (Koh et al., 2020; Madeira et al., 2023) instead of relying solely on state and action information. We demonstrate that this

---

[1]Preprint. Work in progress. The code for replicating the experiments of the paper can be found at: https://anonymous.4open.science/r/ConceptOPE/Readme.md

approach offers significant practical benefits for evaluation. These concepts can capture critical aspects in historical data, such as key transitions in a patient's treatment or features affecting short-term outcomes that serve as proxies for long-term results. By learning interpretable concepts from data, we introduce a new family of concept-based IS estimators that provide more accurate value estimates and stronger statistical guarantees. Additionally, these estimators allow us to identify which concepts contribute most to variance in evaluation. When the evaluation is unreliable, we can modify, intervene on, or remove these high-variance concepts to assess how the resulting evaluation improves (Marcinkevičs et al., 2024; Madeira et al., 2023).

Consider a physician treating two patients with similar disease dynamics. Although their blood counts and oxygen levels may differ, their overall disease profiles might be alike. Therefore, if one patient responds well to a particular treatment, the same treatment could potentially benefit the other. By learning meaningful concepts based on disease profiles rather than individual symptoms at each time point, we can more reliably evaluate which actions are likely to be effective. This is illustrated in Figure 1.

Our work makes the following key contributions: i) We introduce a new family of IS estimators based on interpretable concepts; ii) We derive theoretical conditions ensuring lower variance compared to existing IS estimators; iii) We propose an end-to-end algorithm for optimizing parameterized concepts when concepts are unknown, using OPE characteristics like variance; iv) We show, through synthetic and real experiments, that our estimators for both known and unknown concepts outperform existing ones; v) We interpret the learned concepts to explain OPE characteristics and suggest intervention strategies to further improve OPE estimates.

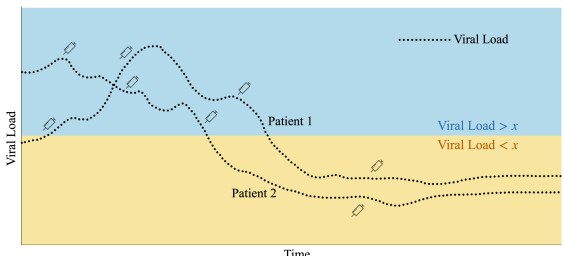

Figure 1: Simple example of a state vs concept. In this scenario, the state is the viral load in a patient's blood, whereas the concept is defined as the viral load being above or below a certain threshold $x$. The concept divides patients into two groups, in which different treatments are administered, indicated by the frequency of syringes. We do evaluation based on these two conceptual groups.

## 2    RELATED WORK

**Off-Policy Evaluation.**    There is a long history of methods for performing OPE, broadly categorized into model-based or model-free (Sutton & Barto, 2018). Model-based methods, such as the Direct Method (DM), learn a model of the environment to simulate trajectories and estimate the policy value (Paduraru, 2013; Chow et al., 2015; Hanna et al., 2017; Fonteneau et al., 2013; Liu et al., 2018b). These methods often rely on strong assumptions about the parametric model for statistical guarantees. Model-free methods, like IS, correct sampling bias in off-policy data through reweighting to obtain unbiased estimates (e.g., Precup et al. (2000); Horvitz & Thompson (1952); Thomas & Brunskill (2016)). Doubly robust (DR) estimators (e.g., Jiang & Li (2016); Farajtabar et al. (2018)) combine model-based DM and model-free IS for OPE but may fail to reduce variance when both DM and IS have high variance. Various methods have been developed to refine estimation accuracy in IS, such as truncating importance weights and estimating weights from steady-state visitation distributions (Liu et al., 2018a; Xie et al., 2019; Doroudi et al., 2017; Bossens & Thomas, 2024).

**Off-Policy Evaluation based on Subgroups.**    Keramati et al. (2021b) extend OPE to estimate treatment effects for subgroups and provide actionable insights on which subgroups may benefit from specific treatments, assuming subgroups are known or identified using regression trees. Unlike regression trees, which are limited in scalability, our approach employs CBMs to learn interpretable concepts that directly characterize individuals, enabling a new family of IS estimators based on these concepts. Similarly, Shen et al. (2021) propose reducing variance by omitting likelihood ratios for certain states. Our work complements this by summarizing relevant trajectory information using concepts, rather than omitting states irrelevant to the return. The advantage of using concepts as

opposed to states is that we can easily interpret and intervene on these concepts unlike the state information.

Marginalized Importance Sampling (MIS) estimators (Uehara et al., 2020; Liu et al., 2018a; Nachum et al., 2019; Zhang et al., 2020b;a) mitigate the high variance of traditional IS by reweighting data tuples using density ratios computed from state visitation at each time step. These estimators enhance robustness by focusing on states with high visitation density ratios, thereby marginalizing out less visited states. However, MIS has its challenges: computing density ratios can introduce high variance, particularly in complex state spaces, and it obscures which aspects of the state space contribute directly to variance. Some studies, such as Katdare et al. (2023) and Fujimoto et al. (2023), improve MIS by decomposing density ratio estimation into components like large density ratio mismatch and transition probability mismatch. Our work differs from MIS by categorizing states using interpretable concepts rather than solely relying on density ratios. This approach enables targeted interventions that enhance policy adjustments, leading to better returns and reduced variance in OPE. Unlike MIS, our method provides interpretability, which becomes increasingly important as problem complexity grows. Proposals for hybrid estimators, such as those in Pavse & Hanna (2022a), suggest using low-dimensional abstraction of state spaces with MIS to manage high-dimensional spaces more effectively. Our research provides a foundational framework for developing such hybrid estimators.

**Concept Bottleneck Models.** Concept Bottleneck Models (Koh et al., 2020) are a class of prediction models that first predict a set of human interpretable concepts, and subsequently use these concepts to predict a downstream label. Variations of these models include learning soft probabilistic concepts (Mahinpei et al., 2021), learning hierarchical concepts (Panousis et al., 2023) and learning concepts in a semi-supervised manner (Sawada & Nakamura, 2022). The key advantage of these models is they allow us to explicitly intervene on concepts and interpret what might happen to a downstream label if certain concepts were changed (Marcinkevičs et al., 2024). Unlike previous works, we leverage this idea to introduce a new class of estimators for *off-policy evaluation* where we group trajectories based on interpretable concepts which are relevant for the downstream evaluation task.

## 3 PRELIMINARIES

**Concept Bottleneck Models** Conventional CBMs learn a mapping from some input features $x \in \mathbb{R}^d$ to targets $y$ via some interpretable concepts $c \in \mathbb{R}^k$ based on training data of the form $\{x_n, c_n, y_n\}_{n=1}^N$. This mapping is a composition of a mapping from inputs to concepts, $f : \mathbb{R}^d \to \mathbb{R}^k$, and a mapping from concepts to targets, $g : \mathbb{R}^k \to \mathbb{R}$. These may be trained via independent, sequential or joint training (Marcinkevičs et al., 2024). Variations which consider learning concepts in a greedy fashion or in a semisupervised way include Wu et al. (2022); Havasi et al. (2022).

**Markov Decision Processes (MDP).** An MDP is defined by a tuple $\mathcal{M} = (\mathcal{S}, \mathcal{A}, P, R, \gamma, T)$. $\mathcal{S}$ and $\mathcal{A}$ are the state and action spaces, $P : \mathcal{S} \times \mathcal{A} \to \Delta(\mathcal{S})$ and $R : \mathcal{S} \times \mathcal{A} \to \Delta(\mathbb{R})$ are the transition and reward functions, $\gamma \in [0, 1]$ is the discount factor, $T \in \mathbb{Z}^+$ is the fixed time horizon. A policy $\pi : \mathcal{S} \to \Delta(\mathcal{A})$ is a mapping from each state to a probability distribution over actions in $\mathcal{A}$. A $T$-step trajectory following policy $\pi$ is denoted by $\tau = [(s_t, a_t, r_t, s_{t+1})]_{t=1}^T$ where $s_1 \sim d_1, a_t \sim \pi(s_t), r_t \sim r(s_t, a_t), s_{t+1} \sim p(s_t, a_t)$. The value function of policy $\pi$, denoted by $V_\pi : \mathcal{S} \to \mathbb{R}$, maps each state to the expected discounted sum of rewards starting from that state following policy $\pi$. That is, $V_\pi(s) = \mathbb{E}_\pi[\sum_{t=1}^T \gamma^{t-1} r_t | s_1 = s]$.

**Off-Policy Evaluation.** In OPE, we have a dataset of $T$-step trajectories $\mathcal{D} = \{\tau^{(n)}\}_{n=1}^N$ independently generated by a *behaviour policy* $\pi_b$. Our goal is to estimate the value function of another *evaluation policy*, $\pi_e$. We aim to use $\mathcal{D}$ to produce an estimator, $\hat{V}_{\pi_e}$, that has low mean squared error, $MSE(V_{\pi_e}, \hat{V}_{\pi_e}) = \mathbb{E}_{\mathcal{D} \sim P_{\pi_b}^\tau}[(V_{\pi_e} - \hat{V}_{\pi_e})^2]$. Here, $P_{\pi_b}^\tau$ denotes the distribution of trajectories $\tau$, under $\pi_b$, from which $\mathcal{D}$ is sampled.

## 4 CONCEPT-BASED OFF-POLICY EVALUATION

We now introduce Concept-driven Off-Policy Evaluation (Concept-OPE). In this section, we formally define the mathematical definition of the concept, outline their desiderata, and present the corre-

sponding OPE estimators. In the following sections, we divide our Concept-OPE studies into two parts. Section 5 covers scenarios where concepts are known from domain knowledge, while Section 6 addresses cases where concepts are unknown and must be learned by optimizing a parameterized representation.

### 4.1 Formal Definition of the Concept

Given a dataset $\mathcal{D} = \{\tau^{(n)}\}_{n=1}^N$ of $n$ $T$-step trajectories, let $\phi : \mathcal{S} \times \mathcal{A} \times R \times \mathcal{S} \rightarrow \mathcal{C} \in \mathbb{R}^d$ denote a function that maps trajectory histories $h_t$ to interpretable concepts in $d$-dimensional concept space $\mathcal{C}$. This mapping results in the concept vector $c_t = [c_t^1, c_t^2, ..., c_t^d]$ at time $t$, defined as $\phi(h_t)$. These concepts can capture various vital information in the history $h_t$, such as transition dynamics, short-term rewards, influential states, interdependencies in actions across timesteps, etc. Without loss of generality, in this work, we consider concepts $c_t$ to be just functions of current state $s_t$. This assumption considers the scenario where concepts capture important information based on the criticalness of the state. The concept function $\phi$ satisfy the following desiderata: explainability, conciseness, better trajectory coverage and diversity. A detailed description of desiderata is provided in Appendix A.

### 4.2 Concept-Based Estimators for OPE.

We introduce a new class of concept-based OPE estimators to formalize the application of concepts in OPE. These estimators are adapted versions of their original non-concept-based counterparts. Here, we present the results specifically for per-decision IS and standard IS estimators, as these serve as the foundation for several other estimators. We also demonstrate in Appendix C how these methods can be extended to other estimators.

**Definition 4.1** (Concept-Based Importance Sampling (CIS))**.**

$$\hat{V}_{\pi_e}^{CIS} = \frac{1}{N} \sum_{n=1}^N \rho_{0:T}^{(n)} \sum_{t=0}^T \gamma^t r_t^{(n)}; \quad \rho_{0:T}^{(n)} = \prod_{t'=0}^T \frac{\pi_e^c(a_{t'}^{(n)}|c_{t'}^{(n)})}{\pi_b^c(a_{t'}^{(n)}|c_{t'}^{(n)})}$$

**Definition 4.2** (Concept-based Per-Decision Importance Sampling, CPDIS)**.**

$$\hat{V}_{\pi_e}^{CPDIS} = \frac{1}{N} \sum_{n=1}^N \sum_{t=0}^T \gamma^t \rho_{0:t}^{(n)} r_t^{(n)}; \quad \rho_{0:t}^{(n)} = \prod_{t'=0}^t \frac{\pi_e^c(a_{t'}^{(n)}|c_{t'}^{(n)})}{\pi_b^c(a_{t'}^{(n)}|c_{t'}^{(n)})}$$

Concept-based variants of IS replace the traditional IS ratio with one that leverages the concept $c_t$ at time $t$ instead of the state $s_t$. This enables customized evaluations for various concept types, such as: 1) subgroups with similar short-term outcomes, 2) cases with comparable state-visitation densities, and 3) subjects with high-variance transitions. Details on selecting concept types are in Appendix B.

## 5 Concept-based OPE under Known Concepts

We first consider the scenario where the concepts are known apriori using domain knowledge and human expertise. These concepts automatically satisfy the desiderata defined in Appendix A.

### 5.1 Theoretical Analysis of Known Concepts

In this subsection, we discuss the theoretical guarantees of OPE under known concepts. We make the completeness assumption where every action of a particular state has a non-zero probability of appearing in the batch data. When this assumption is satisfied, we obtain unbiasedness and lower variance when compared with traditional estimators. Proofs follow in Appendix D.

**Assumption 5.1** (Completeness)**.** $\forall s \in S, a \in A, if \ \pi_b(a|s), \pi_b^c(a|c) > 0 \ then \ \pi_e(a|s), \pi_e^c(a|c) > 0$

This assumption states that if an action appears in the batch data with some probability, it also has a chance of being evaluated with some probability.

**Assumption 5.2.** $\forall s \in S, a \in A, |\pi_e^c(a|c) - \pi_e(a|s)| < \beta \ and \ |\pi_e^c(a|c) - \pi_e(a|s)| < \beta$. *This assumption states that for all states $s$, the policies conditioned on concepts are allowed to differ from the state policies by atmost $\beta$, which is defined by the practitioner.*

This assumption constrains concept-based policies to be close to state-based policies, with a maximum allowable difference of $\beta$, defined by the practitioner. This is to ensure that the evaluation policy $\pi_e^c$ under concepts is reflective of the original policy $\pi_e$. If the practitioner is confident in the state representation, they may set a lower $\beta$ to find concepts that align closely with state policies. Conversely, a higher $\beta$ allows for more deviation between concept and state policies.

**Theorem 5.3** (Bias). *Under known-concepts, when assumption 5.1 holds, both $\hat{V}_{\pi_e}^{CIS}$ and $\hat{V}_{\pi_e}^{CPDIS}$ are unbiased estimators of the true value function $V_{\pi_e}$. (Proof: See Appendix D for details.)*

**Theorem 5.4** (Variance comparison with traditional OPE estimators). *When $Cov(\rho_{0:t}^c r_t, \rho_{0:k}^c r_k) \leq Cov(\rho_{0:t} r_t, \rho_{0:k} r_k)$, the variance of known concept-based IS estimators is lower than traditional estimators, i.e. $\mathbb{V}_{\pi_b}[\hat{V}^{CIS}] \leq \mathbb{V}_{\pi_b}[\hat{V}^{IS}]$, $\mathbb{V}_{\pi_b}[\hat{V}^{CPDIS}] \leq \mathbb{V}_{\pi_b}[\hat{V}^{PDIS}]$. (Proof: See Appendix D)*

As noted in Jiang & Li (2016), the covariance assumption across timesteps is crucial yet challenging for OPE variance comparisons. Concepts being interpretable allows a user to design policies which align with this assumption, thereby reducing variance. We also compare concept-based estimators to the MIS estimator, the gold standard for minimizing variance via steady-state distribution ratios.

**Theorem 5.5** (Variance comparison with MIS estimator). *When $Cov(\rho_{0:t}^c r_t, \rho_{0:k}^c r_k) \leq Cov(\frac{d^{\pi_e}(s_t, a_t)}{d^{\pi_b}(s_t, a_t)} r_t, \frac{d^{\pi_e}(s_k, a_k)}{d^{\pi_b}(s_k, a_k)} r_k)$, the variance of known concept-based IS estimators is lower than the Variance of MIS estimator, i.e. $\mathbb{V}_{\pi_b}[\hat{V}^{CIS}] \leq \mathbb{V}_{\pi_b}[\hat{V}^{MIS}]$, $\mathbb{V}_{\pi_b}[\hat{V}^{CPDIS}] \leq \mathbb{V}_{\pi_b}[\hat{V}^{MIS}]$.*

Finally, we evaluate the CR-bounds on the MSE and quantify the tightness achieved using concepts.

**Theorem 5.6** (Confidence bounds for Concept-based estimators). *The Cramer-Rao bound on the Mean-Square Error of CIS and CPDIS estimator under known-concepts is tightened by a factor of $K^{2T}$, where $K$ is the ratio of the cardinality of the concept-space and state-space.*

High IS ratios arise from low behavior policy probabilities $\pi_b$ due to poor batch sampling, leading to worst-case bounds. Concepts address this by better characterizing poorly sampled states, increasing probabilities, and reducing skewed IS ratios, thus tightening the bounds.

## 5.2 Experimental Setup and Metrics

**Environments:** We consider a synthetic domain: WindyGridworld and the real world MIMIC-III dataset for acutely hypotensive ICU patients as our experiment domains for the rest of the paper.

*WindyGridworld:* We (as human experts) define the concept $c_t = \phi(\text{distance to target, wind})$ as a function of the distance to the target and the wind acting on the agent at a given state. This concept can take 25 unique values, ranging from 0 to 24. For example: $c_t = 0$ when distance to target $\in [15, 19] \times [15, 19]$ and wind $= [0, 0]$. The first and second co-ordinates represent the horizontal and vertical features respectively. Detailed description of known concepts in Appendix G.

*MIMIC:* The concept $c_t \in \mathbb{Z}^{15}$ represents a function of 15 different vital signs (interpretable features) of a patient at a given timestep. The vital signs considered are: Creatinine, $FiO_2$, Lactate, Partial Pressure of Oxygen ($PaO_2$), Partial Pressure of $CO_2$, Urine Output, GCS score, and electrolytes such as Calcium, Chloride, Glucose, $HCO_3$, Magnesium, Potassium, Sodium, and $SpO_2$. Each vital sign is binned into 10 discrete levels, ranging from 0 (very low) to 9 (very high).

For example, a patient with the concept representation $[0, 2, 1, 1, 2, 0, 9, 5, 2, 0, 6, 2, 1, 5, 9]$ shows the following conditions: acute kidney injury (very low creatinine), severe hypoxemia (very low $PaO_2$), metabolic alkalosis (very high $SpO_2$), and critical electrolyte imbalances (low potassium and magnesium), along with severe hypoglycemia. The normal GCS score indicates preserved neurological function, but over-oxygenation and potential respiratory failure are likely. The combination of anuria, AKI, and hypoglycemia points strongly toward hypotension or shock as underlying causes.

**Policy descriptions:** In the case of WindyGridworld, we run a PPO Schulman et al. (2017) algorithm for 10k epochs and consider the evaluation policy $\pi_e$ as the policy at epoch 10k, while the behavior policy $\pi_b$ is taken as the policy at epoch 5k. For the MIMIC case, we generate the behavior policy $\pi_b$ by running an Approximate Nearest Neighbors algorithm with 200 neighbors, using Manhattan distance as the distance metric. The evaluation policy $\pi_e$ involves a more aggressive use of vasopressors (10% more) compared to the behavior policy. See Appendix F for further details.

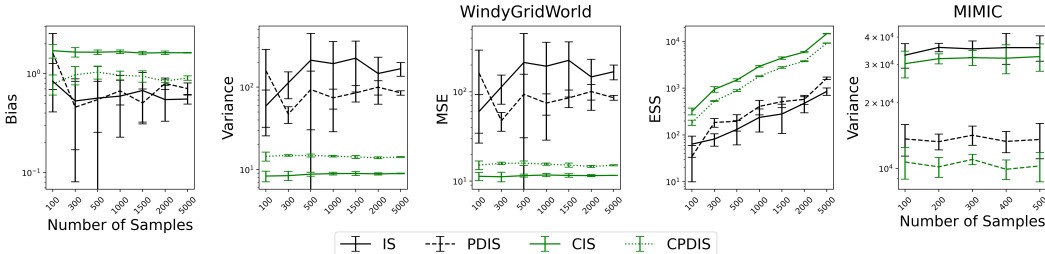

Figure 2: WindyGridworld: Known Concept-based estimators have lower variance, MSE, higher ESS compared to traditional OPE estimators, with a higher Bias. MIMIC: Known Concept-based estimators improve upon the variance.

**Metrics:** In the case of the synthetic domain, we measure bias, variance, mean squared error, and the effective sample size (ESS) to assess the quality of our concept-based OPE estimates. The ESS is defined as $N \times \frac{\mathbb{V}_{\pi_e}[\hat{V}_{\pi_e}^{on-policy}]}{\mathbb{V}_{\pi_b}[\hat{V}_{\pi_e}]}$, where $N$ is the number of trajectories in the off-policy data, and $\hat{V}_{\pi_e}^{on-policy}$ and $\hat{V}_{\pi_e}$ are the on-policy and OPE estimates of the value function, respectively. For MIMIC, where the true on-policy estimate is unknown due to the unknown transition dynamics and environment model, we only consider variance as the metric.

## 5.3 RESULTS AND DISCUSSION

**Known concept-based estimators demonstrate reduced variance, improved ESS, and lower MSE compared to traditional estimators, although they come with slightly higher bias.**

Figure 2 compares known-concept and traditional OPE estimators. We observe a consistent reduction in variance and an increase in ESS across all sample sizes for the concept-based estimators. Although our theoretical analysis suggests that known-concept estimators are unbiased, practical results indicate some bias. While unbiased estimates are generally preferred, they can lead to higher errors when the behavior policy does not cover all states. This issue is especially pronounced in limited data settings, which are common in medical applications. Despite this bias-variance trade-off, the MSE for concept-based OPE estimators shows a 1-2 order of magnitude improvement over traditional estimators due to significant variance reduction. In the real-world MIMIC example, concept-based estimators exhibit a variance reduction of one order of magnitude compared to traditional OPE estimators. This demonstrates that categorizing diverse states—such as varying gridworld positions or patient vital signs—into shared concepts based on common attributes improves OPE characterization.

## 6 CONCEPT-BASED OPE UNDER UNKNOWN CONCEPTS

While domain knowledge and predefined concepts can enhance OPE, real-world complexities and limited human expertise often make these concepts suboptimal, inaccurate, or unknown, with few interpretable features available. Here, we address cases where concepts are unknown and must be estimated. We use a parametric representation of concepts via CBMs, which initially may not meet the required desiderata. This section introduces a methodology to optimize parameterized concepts to meet these desiderata, alongside improving OPE metrics like variance.

### 6.1 METHODOLOGY

Algorithm 1 outlines the training methodology. We split the batch trajectories into training trajectories $\mathcal{T}_{\text{train}}$ and evaluation trajectories $\mathcal{T}_{\text{OPE}}$, with the evaluation policy $\pi_e$, the behavior policy $\pi_b$, and an OPE estimator (eg: CIS/CPDIS) known beforehand. We aim to learn our concepts using a CBM parameterized by $\theta$. The CBM maps states to outputs through an intermediary concept layer. In this work, the output $o$ is the next state, indicating that the bottleneck concepts capture transition dynamics. Other possible outputs could include short-term rewards, long-term returns, or any user-defined information of interest present in the batch data. In addition to learning concepts, we also learn parameterized concept policies $\tilde{\pi}^c$ which maps concepts to actions parameterized by $\theta_b, \theta_e$ for behavior and evaluation policy respectively.

---

**Algorithm 1** Parameterized Concept-based Off Policy Evaluation

---

**Require:** Trajectories $\{\mathcal{T}_{\text{train}}, \mathcal{T}_{\text{OPE}}\}$, Policies $\{\pi_e, \pi_b\}$, OPE Estimator.
**Ensure:** CBM $\theta$, concept policies $\tilde{\pi}^c$ $\{\theta_b, \theta_e\}$
        Loss terms: $\{L_{\text{output}}, L_{\text{interpretability}}, L_{\text{diversity}}, L_{\text{OPE-metric}}, L_{\text{policy}}\} = 0$
 1: **while** Not Converged **do**
 2:    **for** trajectory in $\mathcal{T}_{\text{train}}$ **do**
 3:        **for** $(s, a, r, s', o)$ in trajectory **do**           ▷ Choices for $o$: $s'$ (Next state) / $r$ (Next reward)
 4:            $c', o' \leftarrow \text{CBM}(s)$            ▷ CBM predicts concept $c'$ and output label $o'$
 5:            $L_{\text{output}} \mathrel{+}= C_{\text{output}}(o, o')$     ▷ Eg: MSE/Cross-entropy between true next state and predicted next state
 6:            $L_{\text{interpretability}} \mathrel{+}= C_{\text{interpretability}}(c')$         ▷ Eg: L1-loss over weights
 7:            $L_{\text{diversity}} \mathrel{+}= C_{\text{diversity}}(c')$        ▷ Eg: Cosine distance between sub-concepts
 8:            $L_{\text{policy}} \mathrel{+}= C_{\text{policy}}(c')$   ▷ Eg: MSE/Cross-entropy between predicted logits and true logits in Assn 5.2
 9:        **end for**
10:    **end for**
11:    Returns $\leftarrow$ Estimator$(\mathcal{T}_{train}, \pi_e, \pi_b, \text{CBM})$         ▷ Eg: CIS/CPDIS
12:    Loss$(\theta, \theta_b, \theta_e) = L_{\text{output}} + L_{\text{interpretability}} + L_{\text{diversity}} + C_{\text{OPE-metric}}(\text{Returns})$   ▷ Eg: Variance
13:    Gradient Descent on $\{\theta, \theta_b, \theta_e\}$ using Loss$(\theta, \theta_b, \theta_e)$
14: **end while**
15: **Return** Concept OPE Returns $\leftarrow$ Estimator$(\mathcal{T}_{\text{OPE}}, \pi_e, \pi_b, CBM)$

---

For each transition tuple $(s, a, r, s')$, the CBM computes a concept vector $c'$ and an output $o'$. Since the concepts are initially unknown, they do not inherently satisfy the concept desiderata and must be learned through constraints. Lines 5-7 impose soft constraints on the concepts to meet these desiderata using loss functions. The losses are updated based on output, interpretability, and diversity, with MSE used for $C_{\text{output}}$, L1 loss for $C_{\text{interpretability}}$, and cosine distance for $C_{\text{diversity}}$. In Line 8, we constrain the difference between the concept policies and the original policies to satisfy Assumption 5.2. For our experiments, we take $\beta = 0$, however a user can choose a different value to allow for more deviation in the concept policies $\tilde{\pi}^c$ and original policies $\pi$. In line 11, we evaluate the OPE estimator's returns based on the concepts at the current iteration with metrics like variance. The aggregate loss, Loss$(\theta)$, guides gradient descent on CBM parameters $\theta$. Finally, the OPE estimator is applied to $\mathcal{T}_{\text{OPE}}$ using learned concepts, yielding concept-based OPE returns. Integrating multiple competing loss components makes this problem complex, and, to our knowledge, this is the first approach that incorporates the OPE metric directly into the loss function.

## 6.2 THEORETICAL ANALYSIS OF UNKNOWN CONCEPTS

The theoretical implications mainly differ in the bias, consequently MSE and their Confidence bounds on moving from known to unknown concepts, as analyzed below. Proofs are listed in Appendix E.

**Theorem 6.1** (Bias). *Under unknown concepts, the concept-based estimators are biased.*

Unlike known-concepts, the concept policies are unknown and thus the change of measure theorem from probability distributions $\pi_b$ to $\pi_b^c$ is not applicable, leading to bias. In the special case where $\pi_b^c(.|c_t) = \pi_b(.|s_t)$, the estimator is unbiased.

**Theorem 6.2** (Variance comparison with traditional OPE estimators). *When $Cov(\rho_{0:t}^c r_t, \rho_{0:k}^c r_k) \le Cov(\rho_{0:t} r_t, \rho_{0:k} r_k)$, the variance of concept-based IS estimators is lower than the traditional estimators, i.e. $\mathbb{V}_{\pi_b}[\hat{V}^{CIS}] \le \mathbb{V}_{\pi_b}[\hat{V}^{IS}]$, $\mathbb{V}_{\pi_b}[\hat{V}^{CPDIS}] \le \mathbb{V}_{\pi_b}[\hat{V}^{PDIS}]$. (Proof: Appendix E)*

**Theorem 6.3** (Variance comparison with MIS estimator). *When $Cov(\rho_{0:t}^c r_t, \rho_{0:k}^c r_k) \le Cov(\frac{d^{\pi_e}(s_t, a_t)}{d^{\pi_b}(s_t, a_t)} r_t, \frac{d^{\pi_e}(s_k, a_k)}{d^{\pi_b}(s_k, a_k)} r_k)$, like known concepts, the variance is lower than the Variance of MIS estimator, i.e. $\mathbb{V}_{\pi_b}[\hat{V}^{CIS}] \le \mathbb{V}_{\pi_b}[\hat{V}^{MIS}]$, $\mathbb{V}_{\pi_b}[\hat{V}^{CPDIS}] \le \mathbb{V}_{\pi_b}[\hat{V}^{MIS}]$. (Proof: Appendix E)*

Similar to known concepts, when the covariance assumption is satisfied, even unknown concept-based estimators can provide lower variances than traditional and MIS estimators. In known concepts however, this assumption has to be satisfied by the practitioner, whereas in unknown concepts, this assumption can be used as a loss function in our methodology to implicitly reduce variance.

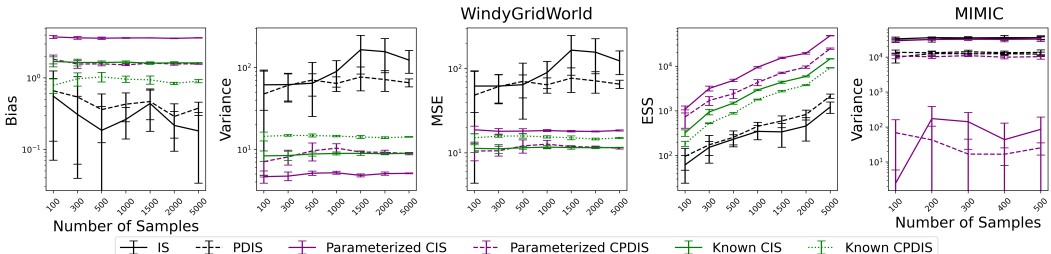

Figure 3: For both domains, unknown concept-based estimators show lower variance. In WindyGrid-world, they improve MSE and ESS but exhibit higher bias compared to traditional OPE estimators.

**Theorem 6.4** (Confidence bounds for Concept-based estimators). *The Cramer-Rao bound on the Mean-Square Error of CIS and CPDIS estimator loosen by $\epsilon(|\mathbb{E}_{\pi_e^c}[\hat{V}_{\pi_e}]|^2)$, under unknown concepts over known-concepts. Here, $\mathbb{E}_{\pi_e^c}[\hat{V}_{\pi_e}]$ is the on-policy estimate of concept-based IS (PDIS) estimator.*

The confidence bounds of unknown concepts mirror that of known-concepts, with the addition of the bias term whose maximum value is the true on-policy estimate of the estimator. This is typically unknown in real-world scenarios and requires additional domain knowledge to mitigate.

### 6.3 EXPERIMENTAL SETUP

**Environments, Policy descriptions, Metrics:** Same as those in known concepts section.

**Concept representation:** In both examples, we use a 4-dimensional concept $c_t \in \mathcal{R}^4$, where each sub-concept is a linear weighted function of human-interpretable features $f$, i.e., $c_t^i = w \cdot f(s_t)$, with $w$ optimized as previously discussed. Detailed descriptions of the features and optimized concepts after CBM training are provided in Appendix I. For MIMIC, features $f$ are normalized vital signs, as threshold information for discretization is unavailable. In brevity of space, we move the training and hyperparameter details to Appendix G.

### 6.4 RESULTS AND DISCUSSION

Figure 3 captures the OPE results from our unknown concepts, which we discuss further.

**Optimized concepts using Algorithm 1 yield improvements across all metrics except bias compared to traditional OPE estimators.** Significant improvements in variance, MSE, and ESS are observed for the WindyGridworld and MIMIC datasets, with gains of 1-2 and 2-3 orders of magnitude, respectively. This improvement is due to our algorithm's ability to identify concepts that satisfy the desiderata, including achieving variance reduction as specified in line 12 of the algorithm. However, like known concepts, optimized concepts show a higher bias than traditional estimators. This is because, unlike variance, bias cannot be optimized in the loss function without the true on-policy estimate, which is typically unavailable in real-world settings. As a result, external information may be essential for further bias reduction.

**Optimized concepts yield improvements across all metrics besides bias over known concept estimators.** Our methodology achieves 1-2 orders of magnitude improvement in variance, MSE, and ESS compared to known concepts. This suggests that our algorithm can learn concepts that surpass human-defined ones in improving OPE metrics. This is particularly valuable in cases with imperfect experts or highly complex real-world scenarios where perfect expertise is unfeasible. However, these optimized concepts introduce higher bias, primarily because the training algorithm prioritized variance reduction over bias minimization. This bias could be reduced by incorporating variance regularization into the training process.

**Optimized concepts are interpretable, show conciseness and diversity.** We list the optimized concepts in Appendix I. These concepts exhibit sparse weights, enhancing their conciseness, with significant variation in weights across different dimensions of the concepts, reflecting diversity. This work focuses on linearly varying concepts, but more complex concepts, such as symbolic representations Majumdar et al. (2023), could better model intricate environments.

# 7 Interventions on Concepts for Insights on Evaluation

Concepts provide interpretations, allowing practitioners to identify sources of variance—an advantage over traditional state abstractions like Pavse & Hanna (2022a). Concepts also clarify reasons behind OPE characteristics, such as high variance, enabling corrective interventions based on domain knowledge or human evaluation. We outline the details of performing interventions next.

## 7.1 Methodology

Given trajectory history $h_t$ and concept $c_t$, we define $c_t^{int}$ as the intervention (alternative) concept an expert proposes at time $t$. We define human criteria $h_c : (h_t, c_t) \to \{0, 1\}$ as a function constructed from domain expertise that takes in $(h_t, c_t)$ as input and outputs a boolean value. This human criteria function determines whether an intervention needs to be conducted over the current concept $c_t$. As an example, if a practitioner has access to true on-policy values, he/she can estimate which concepts suffer from bias. If a concept doesn't suffer from bias, the human criteria $h_c(h_t, c_t) = 1$ is satisfied and the concept is not intervened upon, else $h_c(h_t, c_t) = 0$ and the intervened concept $c_t^{\text{int}}$ is used instead. The final concept $\tilde{c}_t$ is then defined as: $\tilde{c}_t = h_c(h_t, c_t) \cdot c_t + (1 - h_c(h_t, c_t)) \cdot c_t^{\text{int}}$.

The human criteria $h_c$ for our experiments is described as follows. In Windygridworld, we assume access to oracle concepts, listed in Appendix G. When the learned concept $c_t$ matches the true concept, $h_c(h_t, c_t) = 1$, otherwise 0. In MIMIC, interventions are based on a patient's urine output at a specific timestep. We observe in the next subsection that patients with low urine output generally have higher variance, making urine output the human criteria. Thus, $h_c(h_t, c_t) = 1$ when urine output > 30 ml/hr, and 0 otherwise. In this work, we consider 3 possible intervention strategies, 2 being based on state representations and the last being qualitative concept interventions based on domain knowledge.

*Intervening with the state representation and policies.* We intervene on the concept with the state and use policies dependent on state to perform OPE, i.e $c_t^{\text{int}} = s_t$, $\pi_e^c(a_t|\tilde{c}_t) = \pi_e(a_t|s_t)$, $\pi_b^c(a_t|\tilde{c}_t) = \pi_b(a_t|s_t)$. This can be thought of as a comparative measure a practitioner can look for between the concept and the state representations.

*Intervening with the state representation and Maximum likelihood estimator of the policies.* We replace the errorneous concept with the corresponding state and use the MLE of the state conditioned policy to perform OPE, i.e $c_t^{\text{int}} = s_t$, $\pi_e^c(a_t|\tilde{c}_t) = MLE(\pi_e(a_t|s_t))$, $\pi_b^c(a_t|\tilde{c}_t) = MLE(\pi_b(a_t|s_t))$. This can be thought of as a comparative measure a practitioner can look for between the concept and the state representations, while priortising over the most confident action.

*Intervening with a qualitative concept while retaining concept-based policies.* In this approach, a human expert replaces the concept using external domain knowledge, and policies are adjusted to reflect the new concept values. This method aligns with Tang & Wiens (2023), where human-annotated counterfactual trajectories enhance semi-offline OPE. However, while Tang & Wiens focus on quantitative counterfactual annotations in the state representation, we employ human interventions to qualitatively adjust concepts. In case of WindyGridworld, we consider the oracle concepts as our qualitative concept, while for MIMIC, we consider the learnt C-PDIS estimator as qualitative concept while intervening on C-IS estimator.

## 7.2 Interpretation of learnt concepts

We interpret the optimized concepts in Fig. 4. In the WindyGridworld environment, we compare the ground-truth concepts with the optimized ones and observe two additional concepts predicted in the bottom-right region. This likely stems from overfitting to reduce variance in the OPE loss, suggesting a need for inspection and possible intervention. For MIMIC, prior studies indicate that patients with urine output above 30 ml/hr are less susceptible to hypotension than those with lower output Kellum & Prowle (2018); Singer et al. (2016); Vincent & De Backer (2013). Using this knowledge, we analyze patient trajectories and find that lower urine output correlates with higher variance, while higher output corresponds to lower variance. This insight helps identify patients who may benefit from targeted interventions.

## 7.3 Results and Analysis From Interventions on Concepts

**Interpretable concepts allow for targeted interventions that significantly enhance OPE estimates by reducing Bias and MSE in the synthetic domain and reducing Variance in MIMIC.**

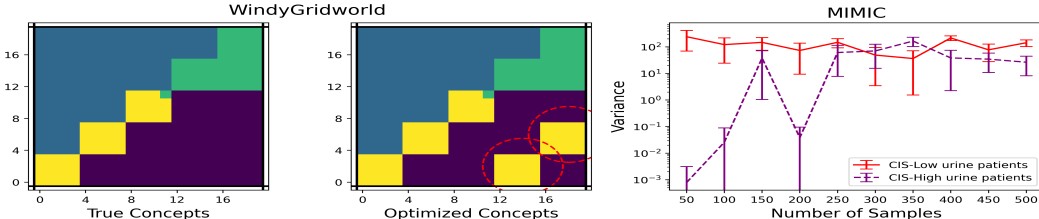

Figure 4: Interpretations of optimized concepts. WindyGridworld: Left panel: Ground-truth concept regions defined by domain knowledge. Right panel: Concept-regions identified by our algorithm. Our algorithm uncovers two additional regions circled in red on the bottom right, highlighting the concepts requiring intervention. MIMIC: Domain knowledge interpretation reveals that patients with low urine output typically exhibit higher variance compared to those with high urine output over the learned concepts, suggesting potential areas for intervention.

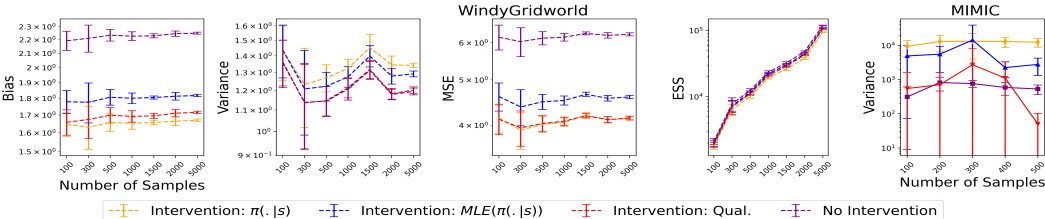

Figure 5: Interventions: Qualitative interventions reduce Bias and MSE for unknown estimators in WindyGridworld and lower variance in MIMIC. Behavior-policy-based interventions improve over non-intervened concepts but are outperformed by qualitative interventions.

In the WindyGridworld environment, we observe a reduction in bias. This occurs because replacing erroneous concepts with oracle concepts introduces information about the on-policy estimates that was previously missing during the optimization of unknown concepts, all while maintaining the same order of variance and ESS estimates. Similarly, in MIMIC, applying qualitative interventions to states with low urine output further reduces variance by 1-2 orders of magnitude.

**Not all interventions improve Concept OPE characteristics and should be used at the practitioner's discretion.** In WindyGridworld, interventions based on state representations increase bias and MSE compared to qualitative interventions, while in MIMIC, they lead to higher variance. This occurs because traditional state policies $\pi_b$ and $\pi_e$ do not address the lack of on-policy information and diminish the benefits of using concept policies $\pi_b^c$ and $\pi_e^c$, rendering these interventions ineffective. In contrast, qualitative interventions—such as oracle concepts in WindyGridworld and urine output thresholds in MIMIC retain the advantages of using concept-based policies and address specific issues, making the intervention more impactful. Importantly, this framework allows practitioners to inspect and choose among alternative interventions as needed.

## 8 Conclusions, Limitations and Future Work

We introduced a new family of concept-based OPE estimators, demonstrating that known-concept estimators can outperform traditional ones with greater accuracy and theoretical guarantees. For unknown concepts, we proposed an algorithm to learn interpretable concepts that improve OPE evaluations by identifying performance issues and enabling targeted interventions to reduce variance. These advancements benefit safety-critical fields like healthcare, education, and public policy by supporting reliable, interpretable policy evaluations. By reducing variance and providing policy insights, this approach enhances informed decision-making, facilitates personalized interventions, and refines policies before deployment for greater real-world effectiveness. A limitation of our work is trajectory distribution mismatch when learning unknown concepts, particularly in low-sample settings, which can lead to high-variance OPE. Targeted interventions help mitigate this issue. We also did not address hidden confounding variables or potential CBM concept leakage, focusing instead on evaluation. Future work will address these challenges and extend our approach to more general, partially observable environments.

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

## A  CONCEPT DESIDERATA

**Explainability:** Explainability ensures that the concept function $\phi$ is composed of human-interpretable functions $f_1, f_2, \ldots, f_n$. Each interpretable function $f_i$ depends on the current state, past actions, rewards, and states, i.e., $s_t, a_{0:t-1}, r_{0:t-1}, s_{0:t-1}$. Mathematically:

$$c_t = \phi(s_t, a_{0:t-1}, r_{0:t-1}, s_{0:t-1}) = \psi(f_1(s_t, a_{0:t-1}, r_{0:t-1}, s_{0:t-1}), \ldots, f_n(s_t, a_{0:t-1}, r_{0:t-1}, s_{0:t-1})) \tag{1}$$

Here, $\psi$ maps the human-interpretable functions $f_i$ to the concept $c_t$, and both $\phi$ and $\psi$ share the same co-domain space $\mathcal{C}$. In essence, $\phi$ can be defined using a single interpretable function or a combination of multiple interpretable functions.

As a running example in this paper (applicable across domains), the concept function $\phi(s_t)$ for diagnosing hypertension can be expressed using human-interpretable features:

$$
\begin{aligned}
c_t &= \phi(s_t) \\
&= \phi(\text{SBP}, \text{DBP}, \text{HR}, \text{Glucose levels}, \text{GCS}, \text{Age}, \text{Weight}) \\
&= \psi(f_1(\text{SBP}), f_2(\text{DBP}), f_3(\text{HR}), f_4(\text{Glucose levels}), f_5(\text{GCS}), f_6(\text{Age}, \text{Weight}))
\end{aligned}
$$

Where:

- $f_1(\text{SBP})$ maps Systolic Blood Pressure to a category (e.g., Low, Normal, High).
- $f_2(\text{DBP})$ maps Diastolic Blood Pressure to a category (e.g., Low, Normal, High).
- $f_3(\text{HR})$ maps Heart Rate to a category (e.g., Low, Normal, High).
- $f_4(\text{Glucose levels})$ maps blood glucose levels to a category (e.g., Low, Normal, High).
- $f_5(\text{GCS})$ maps GCS scores to a category.
- $f_6(\text{Age}, \text{Weight})$ maps age and weight to Body Mass Index (BMI).

This ensures that the concept $\phi(s_t)$ for diagnosing hypertension is built from human-interpretable features, making the diagnostic process explainable. Each function $f_i$ translates raw medical data into intuitive categories that are meaningful to medical practitioners.

**Conciseness:** Conciseness ensures that the concept function $\phi$ represents the minimal mapping of interpretable functions $f_1, f_2, \ldots, f_n$ to the concept $c_t$. If multiple mappings $\psi_1, \psi_2, \ldots, \psi_m$ satisfy $\phi$, we choose the mapping $\psi$ that provides the simplest composition of $f_i$ to describe $c_t$.

E.g. Obesity can be represented by different combinations of human-interpretable functions. We select the least complex representation that remains interpretable. The two possible representations are:

$$c_t = \psi_1(f_1(\text{height}), f_2(\text{weight}), f_3(\text{SBP}), f_4(\text{DBP}))$$
$$c_t = \psi_2(f_5(\text{BMI}), f_3(\text{SBP}))$$

Since BMI encapsulates both height and weight, and either SBP or DBP accurately summarizes blood pressure pertinent to Obesity, the concept $c_t = \psi_2(f_5(\text{BMI}), f_3(\text{SBP}))$ is more concise.

**Better Trajectory Coverage:** Concept-based policies have a higher coverage than traditional state policies. Mathematically:

$$\sum_{\tau \in \mathcal{T}_1} \sum_{t=0}^{T} \pi^c(a_t|c_t) \geq \sum_{\tau \in \mathcal{T}_2} \sum_{t=0}^{T} \pi(a_t|s_t) \tag{2}$$

Here, $\pi^c, \pi$ represent policies conditioned on concepts and states respectively, $\mathcal{T}_1, \mathcal{T}_2$ is the set of all possible trajectories under $\pi^c, \pi$ and $T$ is the total number of timesteps.

**Diversity:** The diversity property ensures that each dimension of the concept at a given timestep captures distinct and independent aspects of the state space, minimizing overlap.

As an example, the concept function $\phi(s_t)$ for a comprehensive patient health assessment can be represented as:

$$\begin{aligned}
\phi(s_t) &= [c_t^1, c_t^2, \ldots, c_t^d] \\
&= [c_t^1(\text{Cardiovascular Health}), c_t^2(\text{Metabolic Health}), c_t^3(\text{Respiratory Health})] \\
&= [\psi_1(f_1(\text{blood pressure}), f_2(\text{cholesterol levels}), f_3(\text{heart rate variability})), \\
&\quad \psi_2(f_1(\text{blood glucose levels}), f_2(\text{BMI}), f_3(\text{metabolic history})), \\
&\quad \psi_3(f_1(\text{lung function}), f_2(\text{oxygen saturation}), f_3(\text{respiratory history}))]
\end{aligned}$$

Each dimension of the concept $c_t^i$ captures unique information, contributing to a holistic assessment of the patient's health without redundancy.

## B  CHOICE OF CONCEPT TYPES

*Concepts capturing subgroups with short-term benefits.* If $\phi$ maps state $s_t$ and action $a_t$ to immediate reward $r_t$, the resulting concepts can identify subgroups with similar short-term benefits, facilitating more personalized OPE, as seen in Keramati et al. (2021a). Unlike Keramati et al. (2021b), we do not limit $\phi$ to a regression tree.

*Concepts capturing high-variance transitions.* If $\phi$ highlights changes in state $s_t$ and action $a_t$ that cause significant shifts in value estimates, it can capture influential transitions or dynamics from historical data, similar to Gottesman et al. (2020).

*Concepts capturing least influential states.* If $\phi$ identifies the least (or most) influential states $s_t$, it can help focus more on critical states, reducing variance by only applying IS ratios to those states Bossens & Thomas (2024).

*Concepts capturing state-density information.* If $\phi$ extracts information from histories to predict state-action visitation counts, concept-based OPE with $\phi$ functions similarly to Marginalized OPE estimators, like Xie et al. (2019), which reweight trajectories based on state-visitation distributions. However, density-based concepts may be less interpretable and harder to intervene in the context of OPE.

## C  GENERALIZED CONCEPT-BASED OPE ESTIMATORS

Building on the OPE estimators discussed in the main paper, we extend the integration of concepts into other popular OPE estimators. Without making any additional assumptions about the estimators' definitions, concepts can be seamlessly incorporated into the original formulations of these estimators.

**Definition C.1** (Concept-based Weighted Importance Sampling, CWIS)**.**

$$\hat{V}_{\pi_e}^{CWIS} = \frac{\sum_{n=1}^{N} \rho_{0:T}^{(n)} \sum_{t=0}^{T} \gamma^t r_t^{(n)}}{\sum_{n=1}^{N} \rho_{0:T}^{(n)}}; \quad \rho_{0:T}^{(n)} = \prod_{t'=0}^{T} \frac{\pi_e(a_{t'}^{(n)}|c_{t'}^{(n)})}{\pi_b(a_{t'}^{(n)}|c_{t'}^{(n)})}$$

**Definition C.2** (Concept-based Per-Decision Weighted Importance Sampling, CPDWIS)**.**

$$\hat{V}_{\pi_e}^{CPDWIS} = \frac{\sum_{n=1}^{N} \sum_{t=0}^{T} \rho_{0:t}^{(n)} \gamma^t r_t^{(n)}}{\sum_{n=1}^{N} \sum_{t=0}^{T} \rho_{0:t}^{(n)}}; \quad \rho_{0:t}^{(n)} = \prod_{t'=0}^{t} \frac{\pi_e(a_{t'}^{(n)}|c_{t'}^{(n)})}{\pi_b(a_{t'}^{(n)}|c_{t'}^{(n)})}$$

**Definition C.3** (Concept-based Doubly Robust Estimator, CDR)**.**

$$\hat{V}_{CDR} = \frac{1}{N} \sum_{i=1}^{N} \sum_{t=0}^{T} \prod_{k=0}^{t} \frac{\pi_e(a_k^{(i)} \mid c_k^{(i)})}{\pi_b(a_k^{(i)} \mid c_k^{(i)})} \left( r_t^{(i)} - \hat{Q}(s_t^{(i)}, a_t^{(i)}) \right) + \hat{V}(s_t^{(i)})$$

Assuming good model-based estimates $\hat{V}(s_t), \hat{Q}(s_t^{(i)}, a_t^{(i)})$, all the advantages seen in the traditional DR estimator translate over to the concept-space representation. It's important to note, the concepts are only used to reweight the Importance Sampling ratios and are not incorporated in the model-based estimates. This allows concepts to have a general form and are not under any markovian assumption, thus satisfying the Bellman equation.

**Definition C.4** (Concept-based Marginalized Importance Sampling Estimator, CMIS)**.**

$$\hat{V}_{CMIS} = \sum_{n=1}^{N} \sum_{t=0}^{T} \frac{d_{\pi_e^c}(c_t)}{d_{\pi_b^c}(c_t)} \gamma^t r_t$$

Different algorithms from the DICE family attempt to estimate the state-distribution ratio $\frac{d_{\pi_e}(s_t)}{d_{\pi_b}(s_t)}$. MIS in the concept representation accounts for concept-visitation counts. These counts retain all the statistical guarantees of the state representation. However, a drawback is that concept-visitation counts are less intuitive than the original concept definition. This makes it harder to assess the quality of the OPE.

## D  KNOWN CONCEPT-BASED OPE ESTIMATORS: THEORETICAL PROOFS

In this section, we provide the detailed proofs for the known concept scenario.

**Theorem.** *For any arbitary function $f$, $\mathbb{E}_{c \sim d_{\pi^c}} f(c) = \mathbb{E}_{s \sim d_\pi} f(\phi(s))$*

*Proof: See Pavse & Hanna (2022b).*

## D.1 IS

### D.1.1 BIAS

$$Bias = |\mathbb{E}_{\pi_b^c}[\hat{V}_{\pi_e^c}^{CIS}] - \mathbb{E}_{\pi_e^c}[\hat{V}_{\pi_e^c}^{CIS}]| \tag{a}$$

$$= |\mathbb{E}_{\pi_b^c}\left[\rho_{0:T}^{(n)}\sum_{t=0}^{T}\gamma^t r_t^{(n)}\right] - \mathbb{E}_{\pi_e^c}[\hat{V}_{\pi_e}^{CIS}]| \tag{b}$$

$$= |\sum_{n=1}^{N}\left(\prod_{t=0}^{T}\pi_b^c(a_t^{(n)}|c_t^{(n)})\right)\rho_{0:T}^{(n)}\sum_{t=0}^{T}\gamma^t r_t^{(n)} - \mathbb{E}_{\pi_e}[\hat{V}_{\pi_e^c}^{CIS}]| \tag{c}$$

$$= |\sum_{n=1}^{N}\prod_{t=0}^{T}\left(\pi_b^c(a_t^{(n)}|c_t^{(n)})\frac{\pi_e^c(a_t^{(n)}|c_t^{(n)})}{\pi_b^c(a_t^{(n)}|c_t^{(n)})}\right)\sum_{t=0}^{T}\gamma^t r_t^{(n)} - \mathbb{E}_{\pi_e^c}[\hat{V}_{\pi_e}^{CIS}]| \tag{d}$$

$$= |\sum_{n=1}^{N}\prod_{t=0}^{T}\pi_e^c(a_t^{(n)}|c_t^{(n)})\sum_{t=0}^{T}\gamma^t r_t^{(n)} - \mathbb{E}_{\pi_e^c}[\hat{V}_{\pi_e^c}^{CIS}]| = 0 \tag{e}$$

**Explanation of steps:**

(a) We start by expressing the definition of Bias as the difference between expected values of the value function sampled under the behavior policy $\pi_b^c$ and the concept-based evaluation policy $\pi_e^c(a|c)$.

(b) We expand the respective definitions.

(c) Each term is expanded to represent the probability of the trajectories, factoring in the importance sampling ratio.

(d) Grouping similar terms. This change of measure is possible as the concepts are known and can be modify the trajectory probabilities.

(e) The denominator of the IS term cancels with the probability of the trajectory under $\pi_b^c$. Using the definition of $\mathbb{E}_{\pi_e^c}[\hat{V}_{\pi_e^c}^{CIS}] = \sum_{n=1}^{N}\prod_{t=0}^{T}\pi_e^c(a_t^{(n)}|c_t^{(n)})\sum_{t=0}^{T}\gamma^t r_t^{(n)}$.

### D.1.2 VARIANCE

$$\mathbb{V}[\hat{V}_{\pi_e^c}^{CIS}] = \mathbb{E}_{\pi_b^c}[(\hat{V}_{\pi_e^c}^{CIS})^2] - (\mathbb{E}_{\pi_b^c}[\hat{V}_{\pi_e^c}^{CIS}])^2 \tag{a}$$

We first evaluate the expectation of the square of the estimator:

$$\mathbb{E}_{\pi_b^c}[(\hat{V}_{\pi_e}^{CIS})^2] = \mathbb{E}_{\pi_b^c}\left[\left(\rho_{0:T}^{(n)}\sum_{t=0}^{T}\gamma^t r_t^{(n)}\right)^2\right] \tag{b}$$

$$= \mathbb{E}_{\pi_b^c}\left[\sum_{t=0}^{T}\sum_{t'=0}^{T}\rho_{0:T}^2\gamma^{(t+t')}r_t^{(n)}r_{t'}^{(n')}\right] \tag{c}$$

$$= \sum_{n=1}^{N}\prod_{t=0}^{T}\frac{(\pi_e^c(a_t^{(n)}|c_t^{(n)}))^2}{\pi_b^c(a_t^{(n)}|c_t^{(n)})}\sum_{t=0}^{T}\sum_{t'=0}^{T}\gamma^{(t+t')}r_t r_{t'} \tag{d}$$

Evaluating the second term in the variance expression:

$$(\mathbb{E}_{\pi_b^c}[\hat{V}_{\pi_e^c}^{CIS}])^2 = \left(\mathbb{E}_{\pi_b^c}[\rho_{0:T}^{(n)}\sum_{t=0}^{T}\gamma^t r_t^{(n)}]\right)^2 \tag{e}$$

$$= \sum_{n=1}^{N}\prod_{t=0}^{T}\left(\pi_b^c(a_t^{(n)}|c_t^{(n)})(\frac{\pi_e^c(a_t^{(n)}|c_t^{(n)})}{\pi_b^c(a_t^{(n)}|c_t^{(n)})})\right)^2\sum_{t=0}^{T}\sum_{t'=0}^{T}\gamma^{(t+t')}r_t r_{t'} \tag{f}$$

$$= \sum_{n=1}^{N}\prod_{t=0}^{T}\left(\pi_e^c(a_t^{(n)}|c_t^{(n)})\right)^2\sum_{t=0}^{T}\sum_{t'=0}^{T}\gamma^{(t+t')}r_t r_{t'} \tag{g}$$

Subtracting the squared expectation from the expectation of the squared estimator:

$$\mathbb{V}[\hat{V}_{\pi_e^c}^{CIS}] = \sum_{n=1}^{N}\prod_{t=0}^{T}\left((\pi_e^c(a_t^{(n)}|c_t^{(n)})^2(\frac{1}{\pi_b^c(a_t^{(n)}|c_t^{(n)})}-1)\right)\sum_{t=0}^{T}\sum_{t'=0}^{T}\gamma^{(t+t')}r_t r_{t'} \qquad \text{(h)}$$

**Explanation of steps:**

(a) We begin with the definition of variance for our estimator.

(b) We evaluate the first term of the Variance.

(c),(d) We expand the square of the estimator as the square of a sum of weighted returns.

(e) We calculate the square of the expectation of the estimator.

(f) We expand this squared expectation.

(g) The denominator of the IS ratio cancels with the probability of the trajectory.

### D.1.3 VARIANCE COMPARISON BETWEEN CIS RATIOS AND IS RATIOS

**Theorem.** $\mathbb{V}[\prod_{t=0}^{T}\frac{\pi_e^c(a_t|c_t)}{\pi_b^c(a_t|c_t)}] \leq \mathbb{V}[\prod_{t=0}^{T}\frac{\pi_e(a_t|s_t)}{\pi_b(a_t|s_t)}]$

**Proof:** The proof is similar to Pavse & Hanna (2022b), where we generalize to concepts from state abstractions. Using Lemma D and Assumption 5.1, we can say that:

$$\mathbb{E}_{c\sim d_{\pi^c}}\prod_{t=0}^{T}\frac{\pi_e^c(a_t|c_t)}{\pi_b^c(a_t|c_t)} = \mathbb{E}_{s\sim d_\pi}\prod_{t=0}^{T}\frac{\pi_e(a_t|s_t)}{\pi_b(a_t|s_t)} = 1 \qquad \text{(a)}$$

Denoting the difference between the two variances as $D$:

$$D = \mathbb{V}[\prod_{t=0}^{T}\frac{\pi_e(a_t|s_t)}{\pi_b(a_t|s_t)}] - \mathbb{V}[\prod_{t=0}^{T}\frac{\pi_e^c(a_t|c_t)}{\pi_b^c(a_t|c_t)}] \qquad \text{(b)}$$

$$= \mathbb{E}_{\pi_b}[\prod_{t=0}^{T}\left(\frac{\pi_e(a_t|s_t)}{\pi_b(a_t|s_t)}\right)^2] - [\mathbb{E}_{\pi_b}\prod_{t=0}^{T}\left(\frac{\pi_e(a_t|s_t)}{\pi_b(a_t|s_t)}\right)]^2 - \mathbb{E}_{\pi_b^c}[\prod_{t=0}^{T}\left(\frac{\pi_e^c(a_t|c_t)}{\pi_b^c(a_t|c_t)}\right)^2] + [\mathbb{E}_{\pi_b^c}\prod_{t=0}^{T}\left(\frac{\pi_e^c(a_t|c_t)}{\pi_b^c(a_t|c_t)}\right)]^2 \qquad \text{(c)}$$

$$= \mathbb{E}_{\pi_b}[\prod_{t=0}^{T}\left(\frac{\pi_e(a_t|s_t)}{\pi_b(a_t|s_t)}\right)^2] - \mathbb{E}_{\pi_b^c}[\prod_{t=0}^{T}\left(\frac{\pi_e^c(a_t|c_t)}{\pi_b^c(a_t|c_t)}\right)^2] \qquad \text{(d)}$$

$$= \sum_{s}\prod_{t=0}^{T}\pi_b(a_t|s_t)[\prod_{t=0}^{T}\left(\frac{\pi_e(a_t|s_t)}{\pi_b(a_t|s_t)}\right)^2] - \sum_{c}\prod_{t=0}^{T}\pi_b^c(a_t|c_t)[\prod_{t=0}^{T}\left(\frac{\pi_e^c(a_t|c_t)}{\pi_b^c(a_t|c_t)}\right)^2] \qquad \text{(e)}$$

$$= \sum_{c}\left(\sum_{s}\prod_{t=0}^{T}\pi_b(a_t|s_t)[\prod_{t=0}^{T}\left(\frac{\pi_e(a_t|s_t)}{\pi_b(a_t|s_t)}\right)^2] - \prod_{t=0}^{T}\pi_b^c(a_t|c_t)[\prod_{t=0}^{T}\left(\frac{\pi_e^c(a_t|c_t)}{\pi_b^c(a_t|c_t)}\right)^2]\right) \qquad \text{(f)}$$

$$= \sum_{c}\left(\sum_{s}[\prod_{t=0}^{T}\left(\frac{\pi_e(a_t|s_t)^2}{\pi_b(a_t|s_t)}\right)] - [\prod_{t=0}^{T}\left(\frac{\pi_e^c(a_t|c_t)^2}{\pi_b^c(a_t|c_t)}\right)]\right) \qquad \text{(g)}$$

We will analyse the difference of variance for 1 fixed concept and denote it as D':

$$D' = [\prod_{t=0}^{T}\left(\frac{\pi_e^c(a_t|c_t)^2}{\pi_b^c(a_t|c_t)}\right)] - \left(\sum_{s}[\prod_{t=0}^{T}\left(\frac{\pi_e(a_t|s_t)^2}{\pi_b(a_t|s_t)}\right)]\right) \qquad \text{(h)}$$

Now, if we can show $D' \geq 0$ for $|c|$, where $|c|$ is the cardinality of the concept representation, then the difference will always be positive, thus completing our proof. We will use induction to prove $D' \geq 0$ on the total number of concepts from 1 to $|c| = n < |S|$. Now, our induction statement $T(n)$ to prove is, $D' \geq 0$ where $n = |c'|$. For $n = 1$, the statement is trivially true where every concept can be represented as the traditional representation of the state. Our inductive hypothesis states that

$$D' = \left([\prod_{t=0}^{T}\left(\frac{\pi_e^c(a_t|c_t)^2}{\pi_b^c(a_t|c_t)}\right)] - \sum_{s}[\prod_{t=0}^{T}\left(\frac{\pi_e(a_t|s_t)^2}{\pi_b(a_t|s_t)}\right)]\right) \geq 0 \qquad \text{(i)}$$

Now, we define $S = \sum_s [\prod_{t=0}^{T} \left( \frac{\pi_e(a_t|s_t)^2}{\pi_b(a_t|s_t)} \right)]$, $C = \prod_{t=0}^{T} \pi_e^c(a_t|c_t)^2$, $C' = \prod_{t=0}^{T} \pi_b^c(a_t|c_t)$. After making the substitutions, we obtain

$$C^2 \leq SC' \tag{j}$$

This result holds true for $|c| = n$ as per the induction. Now, we add a new state $s_{n+1}$ to the concept as part of the induction, and obtain the following difference:

$$D' = S \times \frac{\pi_e(a|s_{n+1})^2}{\pi_b(a|s_{n+1})} - \frac{C}{C'} \times \frac{\pi_e(a|s_{n+1})^2}{\pi_b(a|s_{n+1})} \tag{k}$$

Let $\pi_e(a|s_{n+1}) = X$ and $\pi_b(a|s_{n+1}) = Y$. Substituting, we get:

$$D' = S\frac{X^2}{Y} - \frac{C}{C'}\frac{X^2}{Y} = \frac{(SC' - C)X^2}{C'Y} \tag{l}$$

D' is minimum when C is maximized, hence we substitute $C \leq \sqrt{SC'}$ from the induction hypothesis in the expression

$$D' \leq \frac{(SC' - \sqrt{SC'})X^2}{C'Y} \tag{m}$$

As $SC' \geq 0$, the term $SC' - \sqrt{SC'}$ is never negative, leading to $D' \leq 0$, since the remaining quantities are always positive. Thus, the induction hypothesis holds, and that concludes the proof.

### D.1.4 VARIANCE COMPARISON BETWEEN CIS AND IS ESTIMATORS

**Theorem.** *When* $Cov(\prod_{t=0}^{t} \frac{\pi_e^c(a_t|c_t)}{\pi_b^c(a_t|c_t)} r_t, \prod_{t=0}^{k} \frac{\pi_e^c(a_t|c_t)}{\pi_e^c(a_t|c_t)} r_k) \leq$ $Cov(\prod_{t=0}^{k} \frac{\pi_e(a_t|s_t)}{\pi_b(a_t|s_t)} r_t, \prod_{t=0}^{k} \frac{\pi_e(a_t|s_t)}{\pi_b(a_t|s_t)} r_k)$, *the variance of known concept-based IS estimators is lower than traditional estimators, i.e.* $\mathbb{V}_{\pi_b}[\hat{V}^{CIS}] \leq \mathbb{V}_{\pi_b}[\hat{V}^{IS}]$, $\mathbb{V}_{\pi_b}[\hat{V}^{CPDIS}] \leq \mathbb{V}_{\pi_b}[\hat{V}^{PDIS}]$.

**Proof:** Using Lemma D and Assumption 5.1, we can say that:

$$\mathbb{E}_{c \sim d_{\pi^c}} \left[ \sum_{t=0}^{T} \prod_{t=0}^{T} \frac{\pi_e^c(a_t|c_t)}{\pi_b^c(a_t|c_t)} r_t(c_t, a_t) \right] = \mathbb{E}_{s \sim d_\pi} \left[ \sum_{t=0}^{T} \prod_{t=0}^{T} \frac{\pi_e(a_t|s_t)}{\pi_b(a_t|s_t)} r_t(s_t, a_t) \right] \tag{a}$$

The Variance for a single example of a CIS estimator is given by

$$\mathbb{V}[\hat{V}^{CIS}] = \frac{1}{T^2} \left( \sum_{t=0}^{T} \mathbb{V}[\prod_{t=0}^{T} \frac{\pi_e^c(a_t|c_t)}{\pi_b^c(a_t|c_t)} r_t] + 2 \sum_{t<k} Cov(\prod_{t=0}^{T} \frac{\pi_e^c(a_t|c_t)}{\pi_b^c(a_t|c_t)} r_t, \prod_{t=0}^{T} \frac{\pi_e^c(a_t|c_t)}{\pi_b^c(a_t|c_t)} r_k) \right) \tag{b}$$

The Variance for a single example of a IS estimator is given by

$$\mathbb{V}[\hat{V}^{IS}] = \frac{1}{T^2} \left( \sum_{t=0}^{T} \mathbb{V}[\prod_{t=0}^{T} \frac{\pi_e(a_t|s_t)}{\pi_b(a_t|s_t)} r_t] + 2 \sum_{t<k} Cov(\prod_{t=0}^{T} \frac{\pi_e(a_t|s_t)}{\pi_b(a_t|s_t)} r_t, \prod_{t=0}^{T} \frac{\pi_e(a_t|s_t)}{\pi_b(a_t|s_t)} r_k) \right) \tag{c}$$

We take the difference between the variances, and note the difference of the covariances is not positive as per the assumption. Hence, if we show the differences of variances per timestep is negative, we

complete our proof.

$$D = \mathbb{V}[\prod_{t=0}^{T} \frac{\pi_e^c(a_t|c_t)}{\pi_b^c(a_t|c_t)} r_t] - \mathbb{V}[\prod_{t=0}^{T} \frac{\pi_e(a_t|s_t)}{\pi_b(a_t|s_t)} r_t] \tag{d}$$

$$= \mathbb{E}_{\pi_b^c}[\left(\prod_{t=0}^{T} \frac{\pi_e^c(a_t|c_t)}{\pi_b^c(a_t|c_t)} r_t\right)^2] - [\mathbb{E}_{\pi_b^c}\left(\prod_{t=0}^{T} \frac{\pi_e^c(a_t|c_t)}{\pi_b^c(a_t|c_t)} r_t\right)]^2 - \mathbb{E}_{\pi_b}[\left(\prod_{t=0}^{T} \frac{\pi_e(a_t|s_t)}{\pi_b(a_t|s_t)} r_t\right)^2] + [\mathbb{E}_{\pi_b}\left(\prod_{t=0}^{T} \frac{\pi_e(a_t|s_t)}{\pi_b(a_t|s_t)} r_t\right)]^2 \tag{e}$$

$$= \mathbb{E}_{\pi_b^c}[\left(\prod_{t=0}^{T} \frac{\pi_e^c(a_t|c_t)}{\pi_b^c(a_t|c_t)} r_t\right)^2] - \mathbb{E}_{\pi_b}[\left(\prod_{t=0}^{T} \frac{\pi_e(a_t|s_t)}{\pi_b(a_t|s_t)} r_t\right)^2] \tag{e}$$

$$= \sum_c \prod_{t=0}^{T} (\pi_b^c(a_t|c_t)) [\left(\prod_{t=0}^{T} \frac{\pi_e^c(a_t|c_t)}{\pi_b^c(a_t|c_t)} r_t\right)^2] - \sum_s \prod_{t=0}^{T} (\pi_b(a_t|s_t)) [\left(\prod_{t=0}^{T} \frac{\pi_e(a_t|s_t)}{\pi_b(a_t|s_t)} r_t\right)^2] \tag{f}$$

$$= \sum_c \sum_{s \in \phi^{-1}(c)} \prod_{t=0}^{T} (\pi_b^c(a_t|c_t)) [\left(\prod_{t=0}^{T} \frac{\pi_e^c(a_t|c_t)}{\pi_b^c(a_t|c_t)} r_t\right)^2] - \sum_s \prod_{t=0}^{T} (\pi_b(a_t|s_t)) [\left(\prod_{t=0}^{T} \frac{\pi_e(a_t|s_t)}{\pi_b(a_t|s_t)} r_t\right)^2] \tag{g}$$

$$\leq R_{max}^2 \left( \sum_c \sum_{s \in \phi^{-1}(c)} \prod_{t=0}^{T} (\pi_b^c(a_t|c_t)) [\left(\prod_{t=0}^{T} \frac{\pi_e^c(a_t|c_t)}{\pi_b^c(a_t|c_t)}\right)^2] - \sum_s \prod_{t=0}^{T} (\pi_b(a_t|s_t)) [\left(\prod_{t=0}^{T} \frac{\pi_e(a_t|s_t)}{\pi_b(a_t|s_t)}\right)^2] \right) \tag{h}$$

The rest of the proof is identical to the previous subsection, wherein we perform induction on the cardinality of the concept for and the term inside the bracket is never positive, thus completing the proof.

### D.1.5 UPPER BOUND ON THE VARIANCE

$$\mathbb{V}[\hat{V}_{\pi_b^c}^{CIS}] = \mathbb{E}_{\pi_b^c}(\left( \sum_{t=0}^{T} \gamma^t r_t \prod_{t'=0}^{T} \frac{\pi_e(a_{t'}|c_{t'})}{\pi_b(a_{t'}|c_{t'})} \right)^2) - \mathbb{E}_{\pi_b^c}\left( \sum_{t=0}^{T} \gamma^t r_t \prod_{t'=0}^{T} \frac{\pi_e(a_{t'}|c_{t'})}{\pi_b(a_{t'}|c_{t'})} \right)^2 \tag{a}$$

$$\leq \mathbb{E}_{\pi_b^c}(\left( \sum_{t=0}^{T} \gamma^t r_t \prod_{t'=0}^{T} \frac{\pi_e(a_{t'}|c_{t'})}{\pi_b(a_{t'}|c_{t'})} \right)^2) \tag{b}$$

$$\leq \frac{1}{N} \sum_{n=1}^{N}(\left( \sum_{t=0}^{T} \gamma^t r_t \prod_{t'=0}^{T} \frac{\pi_e(a_{t'}|c_{t'})}{\pi_b(a_{t'}|c_{t'})} \right)^2) + \frac{7T^2 R_{max}^2 U_c^{2T} ln(\frac{2}{\delta})}{3(N-1)} + \sqrt{\frac{ln(\frac{2}{\delta})}{N^3 - N^2} \sum_{i<j}^{N}(X_i^2 - X_j^2)^2} \tag{c}$$

$$\leq T^2 R_{max}^2 U_c^{2T}(\frac{1}{N} + \frac{ln\frac{2}{\delta}}{3(N-1)}) + \sqrt{\frac{ln(\frac{2}{\delta})}{N^3 - N^2} \sum_{i<j}^{N}(X_i^2 - X_j^2)^2} \tag{d}$$

**Explanation of steps:**

(a) We begin with the definition of variance.

(b) The second term is always greater than 0

(c) Applying Bernstein inequality with probability 1-$\delta$. $X_i$ refers to the CIS estimate for 1 sample.

(d) Grouping terms 1 and 2 together, where $U_c = max \frac{\pi_e^c(a|c)}{\pi_b^c(a|c)}$.

The first term of the variance dominates the second with increase in number of samples. Thus, Variance is of the complexity $\mathcal{O}(\frac{T^2 R_{max}^2 U_c^{2T}}{N})$

### D.1.6 Upper Bound on the MSE

$$MSE = Bias^2 + Variance = Variance \sim \mathcal{O}(\frac{T^2 R_{max}^2 U_c^{2T}}{N}) \tag{a}$$

The Upper Bound on the MSE of Concept-based IS estimator is of the same form as the Cramer-Rao bounds of the traditional IS estimator as stated in Jiang & Li (2016). We investigate when the MSE bounds can be tightened in the concept representation. We first say,

$$U_c = max\frac{\pi_e^c(a|c)}{\pi_b^c(a|c)} = U_s\frac{K_1}{K_2} \tag{b}$$

Here, $U_s = max\frac{\pi_e(a|s)}{\pi_b(a|s)}$, $K_1$ is the cardinality of the states which have the same concept $c$ under evaluation policy $\pi_e$, while $K_2$ refers to the same quantity under the behavior policy $\pi_b$. Typically, the maximum value of the IS ratio occurs when $\pi_e(a|s) >> \pi_b(a|s)$, i.e. the action taken is very likely under the evaluation policy $\pi_e$ while it's unlikely under the behavior policy $\pi_b$. This typically happens when that particular state has less coverage, or doesn't appear in the data generated by the behavior policy $\pi_b$. Under concepts however, similar states are visited and categorized, which improves the information on the state $s$ through $c$, leading to $K_2 > 1$. On the other hand, as both $\pi_e^c(a|s)$ and $\pi^e(a|s)$ are close to 1, $K_1 = 1$. Thus, $K = \frac{K_1}{K_2} < 1$ and Hence,

$$\mathcal{O}(\frac{T^2 R_{max}^2 U_c^{2T}}{N}) \sim \mathcal{O}(\frac{T^2 R_{max}^2 (U_s K)^{2T}}{N}) \sim \mathcal{O}(\frac{T^2 R_{max}^2 U_s^{2T}}{N})K^{2T} \tag{3}$$

Thus, the Concept-based MSE bounds are tightened by a factor of $K^{2T}$.

### D.1.7 Variance comparison with MIS estimator

**Theorem.** *Let $\rho$ be the product of the Importance Sampling ratio in the state space, and $d^{\pi_e}, d^{\pi_b}$ be the stationary density ratios. Then,*

$$\mathbb{E}(\rho_{0:T}|s_t, a_t) = \frac{d^{\pi_e}(s_t, a_t)}{d^{\pi_b}(s_t, a_t)}$$

*Proof: See Liu et al. (2020)*

**Theorem.** *Let $X_t$ and $Y_t$ be two sequences of random variables. Then*

$$\mathbb{V}(\sum_t Y_t) - \mathbb{V}(\sum_t \mathbb{E}[Y_t|X_t]) \geq 2\sum_{t<k} \mathbb{E}[Y_t Y_k] - 2\sum_{t<k} \mathbb{E}[\mathbb{E}[Y_t|X_t]\mathbb{E}[Y_k|X_k]]$$

*Proof: See Liu et al. (2020)*

**Theorem.** *When $Cov(\prod_{t=0}^t \frac{\pi_e^c(a_t|c_t)}{\pi_b^c(a_t|c_t)}r_t, \prod_{t=0}^k \frac{\pi_e^c(a_t|c_t)}{\pi_b^c(a_t|c_t)}r_k) \leq Cov(\frac{d^{\pi_e}(s_t,a_t)}{d^{\pi_b}(s_t,a_t)}r_t, \frac{d^{\pi_e}(s_k,a_k)}{d^{\pi_b}(s_k,a_k)}r_k)$, the variance of known CIS estimators is lower than the Variance of MIS estimator, i.e. $\mathbb{V}_{\pi_b}[\hat{V}^{CIS}] \leq \mathbb{V}_{\pi_b}[\hat{V}^{MIS}]$.*

**Proof:** *We start from the assumption:*

$$Cov(\prod_{t=0}^T \frac{\pi_e^c(a_t|c_t)}{\pi_b^c(a_t|c_t)}r_t, \prod_{t=0}^T \frac{\pi_e^c(a_t|c_t)}{\pi_b^c(a_t|c_t)}r_k) = \mathbb{E}[\prod_{t=0}^T \frac{\pi_e^c(a_t|c_t)}{\pi_b^c(a_t|c_t)} \prod_{t=0}^T \frac{\pi_e^c(a_t|c_t)}{\pi_b^c(a_t|c_t)}r_t r_k] - \mathbb{E}[\prod_{t=0}^T \frac{\pi_e^c(a_t|c_t)}{\pi_b^c(a_t|c_t)}r_t]\mathbb{E}[\prod_{t=0}^T \frac{\pi_e^c(a_t|c_t)}{\pi_b^c(a_t|c_t)}r_k] \tag{a}$$

$$= \mathbb{E}[\prod_{t=0}^T \frac{\pi_e(a_t|s_t)}{\pi_b(a_t|s_t)} \prod_{t=0}^T \frac{\pi_e(a_k|s_k)}{\pi_b(a_k|s_k)} K^2 r_t r_k] - \mathbb{E}[\prod_{t=0}^T \frac{\pi_e(a_t|s_t)}{\pi_b(a_t|s_t)} K r_t]\mathbb{E}[\prod_{t=0}^T \frac{\pi_e(a_t|s_t)}{\pi_b(a_t|s_t)} K r_k] \tag{b}$$

$$\leq \mathbb{E}[\prod_{t=0}^T \frac{\pi_e(a_t|s_t)}{\pi_b(a_t|s_t)} \prod_{t=0}^T \frac{\pi_e(a_k|s_k)}{\pi_b(a_k|s_k)} r_t r_k] - \mathbb{E}[\prod_{t=0}^T \frac{\pi_e(a_t|s_t)}{\pi_b(a_t|s_t)} r_t]\mathbb{E}[\prod_{t=0}^T \frac{\pi_e(a_t|s_t)}{\pi_b(a_t|s_t)} r_k] \tag{c}$$

$$\leq \mathbb{E}[(\frac{d^{\pi_e}(s_t,a_t)}{d^{\pi_b}(s_t,a_t)})(\frac{d^{\pi_e}(s_k,a_k)}{d^{\pi_b}(s_k,a_k)})r_t r_k] - \mathbb{E}[\frac{d^{\pi_e}(s_t,a_t)}{d^{\pi_b}(s_t,a_t)}r_t]\mathbb{E}[\frac{d^{\pi_e}(s_k,a_k)}{d^{\pi_b}(s_k,a_k)}r_k] \tag{d}$$

**Explanation of steps:**

*(a) We begin with the definition of covariance.*

*(b),(c) Using the definition of $\pi^c$, with K (the ratio of state-space distribution ratio) $< 1$.*

*(d) Applying Lemma D.1.7 to both the terms.*

*Finally, using Lemma D.1.7, substituting $Y_t = \prod_{t=0}^{T} \frac{\pi_e(a_t|s_t)}{\pi_b(a_t|s_t)} r_t$ and $X_t = s_t, a_t, r_t$ completes our proof.*

## D.2   PDIS

### D.2.1   BIAS

$$Bias = |\mathbb{E}_{\pi_b^c}[\hat{V}_{\pi_e^c}^{CPDIS}] - \mathbb{E}_{\pi_e^c}[\hat{V}_{\pi_e^c}^{CPDIS}]| \tag{a}$$

$$= |\mathbb{E}_{\pi_e^c}\left[\sum_{t=0}^{T} \gamma^t \rho_{0:t}^{(n)} r_t^{(n)}\right] - \mathbb{E}_{\pi_e^c}[\hat{V}_{\pi_e^c}^{CPDIS}]| \tag{b}$$

$$= |\sum_{n=1}^{N} \left(\prod_{t=0}^{T} \pi_b^c(a_t^{(n)}|c_t^{(n)})\right) \sum_{t=0}^{T} \gamma^t \rho_{0:t}^{(n)} r_t^{(n)} - \mathbb{E}_{\pi_e^c}[\hat{V}_{\pi_e^c}^{CPDIS}]| \tag{c}$$

$$= |\sum_{n=1}^{N} \sum_{t=0}^{T} \gamma^t \left(\prod_{t'=0}^{t} \pi_b^c(a_{t'}^{(n)}|c_{t'}^{(n)})(\frac{\pi_e^c(a_{t'}^{(n)}|c_{t'}^{(n)})}{\pi_b^c(a_{t'}^{(n)}|c_{t'}^{(n)})})\right) r_t^{(n)} - \mathbb{E}_{\pi_e^c}[\hat{V}_{\pi_e}^{CPDIS}]| = 0 \tag{d}$$

**Explanation of steps:**   Similar to CIS.

### D.2.2   VARIANCE

Following the process similar to CIS estimator:

$$\mathbb{V}[\hat{V}_{\pi_b^c}^{CPDIS}] = \mathbb{E}_{\pi_b^c}[(\hat{V}_{\pi_b^c}^{CPDIS})^2] - (\mathbb{E}_{\pi_b^c}[\hat{V}_{\pi_b^c}^{CPDIS}])^2 \tag{a}$$

We first evaluate the expectation of the square of the estimator:

$$\mathbb{E}_{\pi_b^c}[(\hat{V}_{\pi_b^c}^{CPDIS})^2] = \mathbb{E}_{\pi_b^c}\left[\left(\sum_{t=0}^{T} \gamma^t \rho_{0:t} r_t\right)^2\right] \tag{b}$$

$$= \mathbb{E}_{\pi_b^c}\left[\sum_{t=0}^{T} \sum_{t'=0}^{T} \rho_{0:t} \rho_{0:t'} \gamma^{(t+t')} r_t r_{t'}\right] \tag{c}$$

$$= \sum_{n=1}^{N} \sum_{t=0}^{T} \sum_{t'=0}^{T} \left(\prod_{t''=0}^{t} \pi_b^c(a_t|c_t)(\frac{\pi_e^c(a_t|c_t)}{\pi_b^c(a_t|c_t)})\right) \left(\prod_{t'''=0}^{t'} (\frac{\pi_e^c(a_t|c_t)}{\pi_b^c(a_t|c_t)})\right) \gamma^{(t+t')} r_t r_{t'} \tag{d}$$

$$= \sum_{n=1}^{N} \sum_{t=0}^{T} \sum_{t'=0}^{T} \left(\prod_{t''=0}^{t} \pi_e^c(a_t|c_t)\right) \left(\prod_{t'''=0}^{t'} (\frac{\pi_e^c(a_t|c_t)}{\pi_b^c(a_t|c_t)})\right) \gamma^{(t+t')} r_t r_{t'} \tag{e}$$

Evaluating the second term in the variance expression:

$$(\mathbb{E}_{\pi_b^c}[\hat{V}_{\pi_b^c}^{CPDIS}])^2 = \left(\sum_{n=1}^{N}\sum_{t=0}^{T}\mathbb{E}_{\pi_b^c}[\gamma^t\rho_{0:t}r_t]\right)^2 \tag{f}$$

$$= \sum_{n=1}^{N}\sum_{t=0}^{T}\sum_{t'=0}^{T}\left(\prod_{t''=0}^{t}\pi_b^c(a_{t''}|c_{t''})(\frac{\pi_e^c(a_{t''}|c_{t''})}{\pi_b^c(a_{t''}|c_{t''})})\right)\left(\prod_{t'''=0}^{t'}\pi_b^c(a_{t'''}|s_{t'''})(\frac{\pi_e^c(a_{t'''}|c_{t'''})}{\pi_b^c(a_{t'''}|c_{t'''})})\right)\gamma^{(t+t')}r_tr_{t'} \tag{g}$$

$$= \sum_{n=1}^{N}\sum_{t=0}^{T}\sum_{t'=0}^{T}\left(\prod_{t''=0}^{t}\pi_e^c(a_{t''}|c_{t''})\right)\left(\prod_{t'''=0}^{t'}\pi_e^c(a_{t'''}|s_{t'''})\right)\gamma^{(t+t')}r_tr_{t'} \tag{h}$$

Subtracting the squared expectation from the expectation of the squared estimator:

$$\mathbb{V}[\hat{V}_{\pi_b^c}^{CPDIS}] = \sum_{n=1}^{N}\sum_{t=0}^{T}\sum_{t'=0}^{T}\left(\prod_{t'''=0}^{t}\pi_e^c(a_{t'''}|c_{t'''})\right)\left(\prod_{t''=0}^{t'}\pi_e^c(a_{t''}|c_{t''})(\frac{1}{\pi_b^c(a_{t''}|c_{t''})}-1)\right)\gamma^{(t+t')}r_tr_{t'} \tag{i}$$

**Explanation of steps:** Similar to CIS.

### D.2.3 VARIANCE COMPARISON BETWEEN CPDIS RATIOS AND PDIS RATIOS

**Theorem.** $\mathbb{V}[\sum_{t=0}^{T}\prod_{t'=0}^{t}\frac{\pi_e^c(a_{t'}|c_{t'})}{\pi_b^c(a_{t'}|c_{t'})}] \leq \mathbb{V}[\sum_{t=0}^{T}\prod_{t'=0}^{t}\frac{\pi_e(a_{t'}|s_{t'})}{\pi_b(a_{t'}|s_{t'})}]$

**Proof:** Similar to CIS estimator.

### D.2.4 VARIANCE COMPARISON BETWEEN CPDIS AND PDIS ESTIMATORS

**Theorem.** *If for any fixed $0 \leq t \leq k < T$, if*

$$Cov(\prod_{t'=0}^{t}\frac{\pi_e^c(a_{t'}|c_{t'})}{\pi_b^c(a_{t'}|c_{t'})}r_t, \prod_{t'=0}^{k}\frac{\pi_e^c(a_{t'}|c_{t'})}{\pi_b^c(a_{t'}|c_{t'})}r_k) \leq Cov(\prod_{t'=0}^{t}\frac{\pi_e(a_{t'}|s_{t'})}{\pi_b(a_{t'}|s_{t'})}r_t, \prod_{t=0}^{T}\frac{\pi_e(a_{t'}|s_{t'})}{\pi_b(a_{t'}|s_{t'})}r_k)$$

*then* $\mathbb{V}[\hat{V}^{CPDIS}] \leq \mathbb{V}[\hat{V}^{PDIS}]$.

**Proof:** Similar to CIS estimator.

### D.2.5 UPPER BOUND ON THE VARIANCE

$$\mathbb{V}[\hat{V}_{\pi_b^c}^{CPDIS}] = \mathbb{E}_{\pi_b^c}(\left(\sum_{t=0}^{T}\gamma^t r_t\prod_{t'=0}^{t}\frac{\pi_e^c(a_{t'}|c_{t'})}{\pi_b^c(a_{t'}|c_{t'})}\right)^2) - \mathbb{E}_{\pi_b^c}\left(\sum_{t=0}^{T}\gamma^t r_t\prod_{t'=0}^{t}\frac{\pi_e^c(a_{t'}|c_{t'})}{\pi_b^c(a_{t'}|c_{t'})}\right)^2 \tag{a}$$

$$\leq \mathbb{E}_{\pi_b^c}(\left(\sum_{t=0}^{T}\gamma^t r_t\prod_{t'=0}^{t}\frac{\pi_e^c(a_{t'}|c_{t'})}{\pi_b^c(a_{t'}|c_{t'})}\right)^2) \tag{b}$$

$$\leq \frac{1}{N}\sum_{n=1}^{N}(\left(\sum_{t=0}^{T}\gamma^t r_t\prod_{t'=0}^{t}\frac{\pi_e^c(a_{t'}|c_{t'})}{\pi_b^c(a_{t'}|c_{t'})}\right)^2) + \frac{7T^2R_{max}^2U_c^{2T}ln(\frac{2}{\delta})}{3(N-1)} + \sqrt{\frac{ln(\frac{2}{\delta})}{N^3-N^2}\sum_{i<j}^{N}(X_i^2-X_j^2)^2} \tag{c}$$

$$\leq T^2R_{max}^2U_c^{2T}(\frac{1}{N}+\frac{ln\frac{2}{\delta}}{3(N-1)}) + \sqrt{\frac{ln(\frac{2}{\delta})}{N^3-N^2}\sum_{i<j}^{N}(X_i^2-X_j^2)^2} \tag{d}$$

**Explanation of steps:** Similar to CIS.

### D.2.6 UPPER BOUND ON THE MSE

$$MSE = Bias^2 + Variance = Variance \sim \mathcal{O}(\frac{T^2 R_{max}^2 U_c^{2T}}{N}) \sim \mathcal{O}(\frac{T^2 R_{max}^2 U_s^{2T}}{N}) K^{2T} \qquad (4)$$

**Proof:** Similar to CIS estimator.

### D.2.7 VARIANCE COMPARISON WITH MIS ESTIMATOR

**Theorem.** *When* $Cov(\prod_{t=0}^t \frac{\pi_e^c(a_t|c_t)}{\pi_b^c(a_t|c_t)} r_t, \prod_{t=0}^k \frac{\pi_e^c(a_t|c_t)}{\pi_b^c(a_t|c_t)} r_k) \leq Cov(\frac{d^{\pi_e}(s_t,a_t)}{d^{\pi_b}(s_t,a_t)} r_t, \frac{d^{\pi_e}(s_k,a_k)}{d^{\pi_b}(s_k,a_k)} r_k),$ *the variance of known CPDIS estimators is lower than the Variance of MIS estimator, i.e.* $\mathbb{V}_{\pi_b}[\hat{V}^{CPDIS}] \leq \mathbb{V}_{\pi_b}[\hat{V}^{MIS}].$

**Proof:** Similar to CIS estimator.

## E   UNKNOWN CONCEPT-BASED OPE ESTIMATORS: THEORETICAL PROOFS

In this section, we provide the theoretical proofs of the unknown concept scenarios.

### E.1   IS

#### E.1.1   BIAS

We begin by stating the expression for the expected value of the CIS estimator under $\pi_b$:

$$Bias = |\mathbb{E}_{\pi_b}[\hat{V}_{\pi_e}^{CIS}] - \mathbb{E}_{\pi_e}[\hat{V}_{\pi_e^c}^{CIS}]| \qquad (a)$$

$$= |\mathbb{E}_{\pi_b}\left[\rho_{0:T}^{(n)} \sum_{t=0}^T \gamma^t r_t^{(n)}\right] - \mathbb{E}_{\pi_e}[\hat{V}_{\pi_e^c}^{CIS}]| \qquad (b)$$

$$= |\sum_{n=1}^N \left(\prod_{t=0}^T \pi_b(a_t^{(n)}|s_t^{(n)})\right) \rho_{0:T}^{(n)} \sum_{t=0}^T \gamma^t r_t^{(n)} - \mathbb{E}_{\pi_e}[\hat{V}_{\pi_e^c}^{CIS}]| \qquad (c)$$

$$= |\sum_{n=1}^N \prod_{t=0}^T \left(\pi_b(a_t^{(n)}|s_t^{(n)}) \frac{\pi_e^c(a_t^{(n)}|\tilde{c}_t^{(n)})}{\pi_b^c(a_t^{(n)}|\tilde{c}_t^{(n)})}\right) \sum_{t=0}^T \gamma^t r_t^{(n)} - \mathbb{E}_{\pi_e}[\hat{V}_{\pi_e^c}^{CIS}]| \qquad (d)$$

**Explanation of steps:**

(a) We start by expressing the definition of Bias as the difference between expected values of the value function sampled under the behavior policy $\pi_b$ and the concept-based evaluation policy $\pi_e(a|c)$.

(b) We expand the respective definitions.

(c) Each term is expanded to represent the probability of the trajectories, factoring in the importance sampling ratio.

(d) Similar terms are grouped together to concisely represent the impact of the importance sampling ratios.

The bias of the CIS estimator is mimimum when the concepts $\tilde{c}_t$ equals the traditional state representations $s_t$, thus, implying imperfect concept-based sampling induces bias. As the concepts are unknown, the reparameterization of the probabilities of the behavior trajectories isn't possible, thus leading to a finite bias as opposed to Known-concept representations.

#### E.1.2   VARIANCE

We start with the definition of variance for the CIS estimator:

$$\mathbb{V}[\hat{V}_{\pi_e}^{CIS}] = \mathbb{E}_{\pi_b}[(\hat{V}_{\pi_e}^{CIS})^2] - (\mathbb{E}_{\pi_b}[\hat{V}_{\pi_e}^{CIS}])^2 \qquad (a)$$

We first evaluate the expectation of the square of the estimator:

$$\mathbb{E}_{\pi_b}[(\hat{V}_{\pi_e}^{CIS})^2] = \mathbb{E}_{\pi_b}\left[\left(\rho_{0:T}^{(n)}\sum_{t=0}^{T}\gamma^t r_t^{(n)}\right)^2\right] \tag{b}$$

$$= \mathbb{E}_{\pi_b}\left[\sum_{t=0}^{T}\sum_{t'=0}^{T}\rho_{0:T}^2 \gamma^{(t+t')}r_t^{(n)}r_{t'}^{(n')}\right] \tag{c}$$

$$= \sum_{n=1}^{N}\prod_{t=0}^{T}\left(\pi_b(a_t^{(n)}|s_t^{(n)})(\frac{\pi_e^c(a_t^{(n)}|\tilde{c}_t^{(n)})}{\pi_b^c(a_t^{(n)}|\tilde{c}_t^{(n)})})^2\right)\sum_{t=0}^{T}\sum_{t'=0}^{T}\gamma^{(t+t')}r_t r_{t'} \tag{d}$$

Evaluating the second term in the variance expression:

$$(\mathbb{E}_{\pi_b}[\hat{V}_{\pi_e}^{CIS}])^2 = \left(\mathbb{E}_{\pi_b}[\rho_{0:T}^{(n)}\sum_{t=0}^{T}\gamma^t r_t^{(n)}]\right)^2 \tag{e}$$

$$= \sum_{n=1}^{N}\prod_{t=0}^{T}\left(\pi_b(a_t^{(n)}|s_t^{(n)})(\frac{\pi_e^c(a_t^{(n)}|\tilde{c}_t^{(n)})}{\pi_b^c(a_t^{(n)}|\tilde{c}_t^{(n)})})\right)^2\sum_{t=0}^{T}\sum_{t'=0}^{T}\gamma^{(t+t')}r_t r_{t'} \tag{f}$$

Subtracting the squared expectation from the expectation of the squared estimator:

$$\mathbb{V}[\hat{V}_{\pi_e}^{CIS}] = \sum_{n=1}^{N}\prod_{t=0}^{T}\left((\pi_b(a_t^{(n)}|s_t^{(n)}) - \pi_b(a_t^{(n)}|s_t^{(n)})^2)(\frac{\pi_e^c(a_t^{(n)}|\tilde{c}_t^{(n)})}{\pi_b^c(a_t^{(n)}|\tilde{c}_t^{(n)})})^2\right)\sum_{t=0}^{T}\sum_{t'=0}^{T}\gamma^{(t+t')}r_t r_{t'} \tag{g}$$

**Explanation of steps:**

(a) We begin with the definition of variance for our estimator.

(b) We expand the square of the estimator as the square of a sum of weighted returns.

(c),(d) We further expand the expected value of this squared sum and evaluate the expected values under the assumption that trajectories are sampled independently.

(e) We calculate the square of the expectation of the estimator.

(f) We expand this squared expectation.

(g) We obtain the final expression for variance by subtracting the squared expectation from the expectation of the squared estimator, simplifying to consider the covariance terms.

### E.1.3 Variance comparison between Concept IS ratios and Traditional IS ratios

**Theorem.** $\mathbb{V}[\prod_{t=0}^{T}\frac{\pi_e^c(a_t|\tilde{c}_t)}{\pi_b^c(a_t|\tilde{c}_t)}] \leq \mathbb{V}[\prod_{t=0}^{T}\frac{\pi_e(a_t|s_t)}{\pi_b(a_t|s_t)}]$

**Proof:** The proof is similar to Pavse & Hanna (2022b) and the ones we used in known concepts, where we generalize to parameterized concepts from state abstractions. The proof remains intact because we make no assumptions on how the concepts are derived, as long as they satisfy the desiderata. Using Lemma D and Assumption 5.1, we can say that:

$$\mathbb{E}_{c\sim d_{\pi^c}}\prod_{t=0}^{T}\frac{\pi_e^c(a_t|\tilde{c}_t)}{\pi_b^c(a_t|\tilde{c}_t)} = \mathbb{E}_{s\sim d_\pi}\prod_{t=0}^{T}\frac{\pi_e(a_t|s_t)}{\pi_b(a_t|s_t)} = 1 \tag{a}$$

Denoting the difference between the two variances as $D$:

$$D = Var[\prod_{t=0}^{T} \frac{\pi_e(a_t|s_t)}{\pi_b(a_t|s_t)}] - Var[\prod_{t=0}^{T} \frac{\pi_e^c(a_t|\tilde{c}_t)}{\pi_b^c(a_t|\tilde{c}_t)}] \tag{b}$$

$$= \mathbb{E}_{\pi_b}[\prod_{t=0}^{T} \left( \frac{\pi_e(a_t|s_t)}{\pi_b(a_t|s_t)} \right)^2] - \mathbb{E}_{\pi_b}[\prod_{t=0}^{T} \left( \frac{\pi_e^c(a_t|\tilde{c}_t)}{\pi_b^c(a_t|\tilde{c}_t)} \right)^2] \tag{c}$$

$$= \sum_s \prod_{t=0}^{T} \pi_b(a_t|s_t)[\prod_{t=0}^{T} \left( \frac{\pi_e(a_t|s_t)}{\pi_b(a_t|s_t)} \right)^2] - \sum_c \prod_{t=0}^{T} \pi_b^c(a_t|\tilde{c}_t)[\prod_{t=0}^{T} \left( \frac{\pi_e^c(a_t|\tilde{c}_t)}{\pi_b^c(a_t|\tilde{c}_t)} \right)^2] \tag{d}$$

$$= \sum_c \left( \sum_s \prod_{t=0}^{T} \pi_b(a_t|s_t)[\prod_{t=0}^{T} \left( \frac{\pi_e(a_t|s_t)}{\pi_b(a_t|s_t)} \right)^2] - \prod_{t=0}^{T} \pi_b^c(a_t|\tilde{c}_t)[\prod_{t=0}^{T} \left( \frac{\pi_e^c(a_t|\tilde{c}_t)}{\pi_b^c(a_t|\tilde{c}_t)} \right)^2] \right) \tag{e}$$

$$= \sum_c \left( \sum_s [\prod_{t=0}^{T} \left( \frac{\pi_e(a_t|s_t)^2}{\pi_b(a_t|s_t)} \right)] - [\prod_{t=0}^{T} \left( \frac{\pi_e^c(a_t|\tilde{c}_t)^2}{\pi_b^c(a_t|\tilde{c}_t)} \right)] \right) \tag{f}$$

We will analyse the difference of variance for 1 fixed concept and denote it as D':

$$D' = [\prod_{t=0}^{T} \left( \frac{\pi_e^c(a_t|\tilde{c}_t)^2}{\pi_b^c(a_t|\tilde{c}_t)} \right)] - \left( \sum_s [\prod_{t=0}^{T} \left( \frac{\pi_e(a_t|s_t)^2}{\pi_b(a_t|s_t)} \right)] \right) \tag{g}$$

Now, if we can show $D' \geq 0$ for $|c|$, where $|c|$ is the cardinality of concept representation, then the difference will always be positive, thus completing our proof. We will use induction to prove $D' \geq 0$ on the total number of concepts from 1 to $|c| = n < |S|$. Now, our induction statement $T(n)$ to prove is, $D' \geq 0$ where $n = |c'|$. For $n = 1$, the statement is trivially true where every concept can be represented as the traditional representation of the state. Our inductive hypothesis states that

$$D' = \left( [\prod_{t=0}^{T} \left( \frac{\pi_e^c(a_t|\tilde{c}_t)^2}{\pi_b^c(a_t|\tilde{c}_t)} \right)] - \sum_s [\prod_{t=0}^{T} \left( \frac{\pi_e(a_t|s_t)^2}{\pi_b(a_t|s_t)} \right)] \right) \geq 0 \tag{h}$$

Now, we define $S = \sum_s [\prod_{t=0}^{T} \left( \frac{\pi_e(a_t|s_t)^2}{\pi_b(a_t|s_t)} \right)]$, $C = \prod_{t=0}^{T} \pi_e^c(a_t|\tilde{c}_t)^2$, $C' = \prod_{t=0}^{T} \pi_b^c(a_t|\tilde{c}_t)$. After making the substitutions, we obtain

$$C^2 \leq SC' \tag{i}$$

This result holds true for $|c| = n$ as per the induction. Now, we add a new state $s_{n+1}$ to the concept as part of the induction, and obtain the following difference:

$$D' = S \times \frac{\pi_e(a|s_{n+1})^2}{\pi_b(a|s_{n+1})} - \frac{C}{C'} \times \frac{\pi_e(a|s_{n+1})^2}{\pi_b(a|s_{n+1})} \tag{j}$$

Let $\pi_e(a|s_{n+1}) = X$ and $\pi_b(a|s_{n+1}) = Y$. Substituting, we get:

$$D' = S\frac{X^2}{Y} - \frac{C}{C'}\frac{X^2}{Y} = \frac{(SC' - C)X^2}{C'Y} \tag{k}$$

D' is minimum when C is maximized, hence we substitute $C \leq \sqrt{SC'}$ from the induction hypothesis in the expression

$$D' \leq \frac{(SC' - \sqrt{SC'})X^2}{C'Y} \tag{l}$$

As $SC' \geq 0$, the term $SC' - \sqrt{SC'}$ is never negative, leading to $D' \leq 0$, since the remaining quantities are always positive. Thus, the induction hypothesis holds, and that concludes the proof.

### E.1.4 VARIANCE COMPARISON BETWEEN UNKNOWN CIS AND IS ESTIMATORS

**Theorem.** *When* $Cov(\prod_{t=0}^{t} \frac{\pi_e^c(a_t|c_t)}{\pi_b^c(a_t|c_t)} r_t, \prod_{t=0}^{k} \frac{\pi_e^c(a_t|c_t)}{\pi_b^c(a_t|c_t)} r_k) \leq$
$Cov(\prod_{t=0}^{k} \frac{\pi_e(a_t|s_t)}{\pi_b(a_t|s_t)} r_t, \prod_{t=0}^{k} \frac{\pi_e(a_t|s_t)}{\pi_b(a_t|s_t)} r_k)$, *the variance of unknown CIS estimator is lower than IS estimator, i.e.* $\mathbb{V}_{\pi_b}[\hat{V}^{CIS}] \leq \mathbb{V}_{\pi_b}[\hat{V}^{IS}]$.

**Proof:** Using Lemma D and Assumption 5.1, we can say that:

$$\mathbb{E}_{c \sim d_{\pi^c}} \left[ \sum_{t=0}^{T} \prod_{t=0}^{T} \frac{\pi_e^c(a_t|\tilde{c}_t)}{\pi_b^c(a_t|\tilde{c}_t)} r_t(c_t, a_t) \right] = \mathbb{E}_{s \sim d_{\pi}} \left[ \sum_{t=0}^{T} \prod_{t=0}^{T} \frac{\pi_e(a_t|s_t)}{\pi_b(a_t|s_t)} r_t(s_t, a_t) \right] \tag{a}$$

The Variance for a single example of a CIS estimator is given by

$$\mathbb{V}[\hat{V}^{CIS}] = \frac{1}{T^2} \left( \sum_{t=0}^{T} \mathbb{V}[\prod_{t=0}^{T} \frac{\pi_e^c(a_t|\tilde{c}_t)}{\pi_b^c(a_t|\tilde{c}_t)} r_t] + 2 \sum_{t<k} Cov(\prod_{t=0}^{T} \frac{\pi_e^c(a_t|\tilde{c}_t)}{\pi_b^c(a_t|\tilde{c}_t)} r_t, \prod_{t=0}^{T} \frac{\pi_e^c(a_t|\tilde{c}_t)}{\pi_b^c(a_t|\tilde{c}_t)} r_k) \right) \tag{b}$$

The Variance for a single example of a IS estimator is given by

$$\mathbb{V}[\hat{V}^{IS}] = \frac{1}{T^2} \left( \sum_{t=0}^{T} \mathbb{V}[\prod_{t=0}^{T} \frac{\pi_e(a_t|s_t)}{\pi_b(a_t|s_t)} r_t] + 2 \sum_{t<k} Cov(\prod_{t=0}^{T} \frac{\pi_e(a_t|s_t)}{\pi_b(a_t|s_t)} r_t, \prod_{t=0}^{T} \frac{\pi_e(a_t|s_t)}{\pi_b(a_t|s_t)} r_k) \right) \tag{c}$$

We take the difference between the variances, and note the difference of the covariances is not positive as per the assumption. Hence, if we show the differences of variances per timestep is negative, we complete our proof.

$$D = \mathbb{V}[\prod_{t=0}^{T} \frac{\pi_e^c(a_t|\tilde{c}_t)}{\pi_b^c(a_t|\tilde{c}_t)} r_t] - \mathbb{V}[\prod_{t=0}^{T} \frac{\pi_e(a_t|s_t)}{\pi_b(a_t|s_t)} r_t] \tag{d}$$

$$= \mathbb{E}_{\pi_b^c}[\left( \prod_{t=0}^{T} \frac{\pi_e^c(a_t|\tilde{c}_t)}{\pi_b^c(a_t|\tilde{c}_t)} r_t \right)^2] - \mathbb{E}_{\pi_b}[\left( \prod_{t=0}^{T} \frac{\pi_e(a_t|s_t)}{\pi_b(a_t|s_t)} r_t \right)^2] \tag{e}$$

$$= \sum_{c} \prod_{t=0}^{T} (\pi_b^c(a_t|\tilde{c}_t)) [\left( \prod_{t=0}^{T} \frac{\pi_e^c(a_t|\tilde{c}_t)}{\pi_b^c(a_t|\tilde{c}_t)} r_t \right)^2] - \sum_{s} \prod_{t=0}^{T} (\pi_b(a_t|s_t)) [\left( \prod_{t=0}^{T} \frac{\pi_e(a_t|s_t)}{\pi_b(a_t|s_t)} r_t \right)^2] \tag{f}$$

$$= \sum_{c} \sum_{s \in \phi^{-1}(c)} \prod_{t=0}^{T} (\pi_b^c(a_t|\tilde{c}_t)) [\left( \prod_{t=0}^{T} \frac{\pi_e^c(a_t|\tilde{c}_t)}{\pi_b^c(a_t|\tilde{c}_t)} r_t \right)^2] - \sum_{s} \prod_{t=0}^{T} (\pi_b(a_t|s_t)) [\left( \prod_{t=0}^{T} \frac{\pi_e(a_t|s_t)}{\pi_b(a_t|s_t)} r_t \right)^2]$$

$$\tag{g}$$

$$\leq R_{max}^2 \left( \sum_{c} \sum_{s \in \phi^{-1}(c)} \prod_{t=0}^{T} (\pi_b^c(a_t|\tilde{c}_t)) [\left( \prod_{t=0}^{T} \frac{\pi_e^c(a_t|\tilde{c}_t)}{\pi_b^c(a_t|\tilde{c}_t)} \right)^2] - \sum_{s} \prod_{t=0}^{T} (\pi_b(a_t|s_t)) [\left( \prod_{t=0}^{T} \frac{\pi_e(a_t|s_t)}{\pi_b(a_t|s_t)} \right)^2] \right)$$

$$\tag{h}$$

The rest of the proof is identical to the previous subsection of known concepts, wherein we apply induction over the cardinality of the concepts and show the term inside the bracket is never positive, thus completing the proof.

### E.1.5 UPPER BOUND ON THE BIAS

Unlike known concepts, there exists a finite bias in case of unknown concepts, and the finite bounds need to be analyzed.

$$Bias = |\sum_{n=1}^{N} \prod_{t=0}^{T} \left( \pi_b(a_t^{(n)}|s_t^{(n)}) \frac{\pi_e^c(a_t^{(n)}|\tilde{c}_t^{(n)})}{\pi_b^c(a_t^{(n)}|\tilde{c}_t^{(n)})} \right) \sum_{t=0}^{T} \gamma^t r_t^{(n)} - \mathbb{E}_{\pi_e^c}[\hat{V}_{\pi_e^c}^{CIS}]| \tag{a}$$

$$\leq |\sum_{n=1}^{N} \prod_{t=0}^{T} \left( \pi_e^c(a_t^{(n)}|\tilde{c}_t^{(n)}) (\frac{\pi_b(a_t^{(n)}|s_t^{(n)})}{\pi_b(a_t^{(n)}|\tilde{c}_t^{(n)})}) \right) \sum_{t=0}^{T} \gamma^t r_t^{(n)}| + |\mathbb{E}_{\pi_e^c}[\hat{V}_{\pi_e^c}^{CIS}]| \tag{b}$$

$$\leq \frac{1}{N} |\sum_{n=1}^{N} \prod_{t=0}^{T} \frac{\pi_e^c(a_t^{(n)}|\tilde{c}_t^{(n)})}{\pi_b^c(a_t^{(n)}|\tilde{c}_t^{(n)})} \sum_{t=0}^{T} \gamma^t r_t^{(n)}| + \frac{7TR_{max}U_c^T ln(\frac{2}{\delta})}{3(N-1)} + \sqrt{\frac{ln(\frac{2}{\delta})}{N^3-N^2} \sum_{i<j}^{N} (X_i - X_j)^2} + |\mathbb{E}_{\pi_e^c}[\hat{V}_{\pi_e^c}^{CIS}]| \tag{c}$$

$$\leq TR_{max}U_c^T (\frac{1}{N} + \frac{ln\frac{2}{\delta}}{3(N-1)}) + \sqrt{\frac{ln(\frac{2}{\delta})}{N^3-N^2} \sum_{i<j}^{N} (X_i - X_j)^2} + |\mathbb{E}_{\pi_e^c}[\hat{V}_{\pi_e^c}^{CIS}]| \tag{d}$$

**Explanation of steps:**

(a) We begin with the evaluated Bias expression.

(b) Applying triangle inequality.

(c) Applying Bernstein inequality with probability 1-$\delta$. $X_i$ refers to the CIS estimate for 1 sample.

(d) Grouping terms 1 and 2 together, where $U_c = max \frac{\pi_e^c(a|\tilde{c})}{\pi_b^c(a|\tilde{c})}$.

The first term of the bias dominates the second in terms of the number of samples, with the true expectation of the CIS estimator being unknown in general cases. Generally, the maximum possible reward is known, which leads to the first term dominating the Bias expression. Thus, Bias is of the complexity $\mathcal{O}(\frac{TR_{max}U_c^T}{N})$

### E.1.6 Upper Bound on the Variance

$$\mathbb{V}[\hat{V}_{\pi_b}^{CPDIS}] = \mathbb{E}_{\pi_b}\left(\left(\sum_{t=0}^{T}\gamma^t r_t \prod_{t'=0}^{T}\frac{\pi_e^c(a_{t'}|\tilde{c}_{t'})}{\pi_b^c(a_{t'}|\tilde{c}_{t'})}\right)^2\right) - \mathbb{E}_{\pi_b}\left(\sum_{t=0}^{T}\gamma^t r_t \prod_{t'=0}^{T}\frac{\pi_e^c(a_{t'}|\tilde{c}_{t'})}{\pi_b^c(a_{t'}|\tilde{c}_{t'})}\right)^2 \qquad (a)$$

$$\leq \mathbb{E}_{\pi_b}\left(\left(\sum_{t=0}^{T}\gamma^t r_t \prod_{t'=0}^{T}\frac{\pi_e^c(a_{t'}|\tilde{c}_{t'})}{\pi_b^c(a_{t'}|\tilde{c}_{t'})}\right)^2\right) \qquad (b)$$

$$\leq \frac{1}{N}\sum_{n=1}^{N}\left(\left(\sum_{t=0}^{T}\gamma^t r_t \prod_{t'=0}^{T}\frac{\pi_e^c(a_{t'}|\tilde{c}_{t'})}{\pi_b^c(a_{t'}|\tilde{c}_{t'})}\right)^2\right) + \frac{7T^2 R_{max}^2 U_c^{2T}ln(\frac{2}{\delta})}{3(N-1)} + \sqrt{\frac{ln(\frac{2}{\delta})}{N^3-N^2}\sum_{i<j}^{N}(X_i^2-X_j^2)^2} \qquad (c)$$

$$\leq T^2 R_{max}^2 U_c^{2T}\left(\frac{1}{N} + \frac{ln\frac{2}{\delta}}{3(N-1)}\right) + \sqrt{\frac{ln(\frac{2}{\delta})}{N^3-N^2}\sum_{i<j}^{N}(X_i^2-X_j^2)^2} \qquad (d)$$

**Explanation of steps:**

(a) We begin with the definition of variance.

(b) The second term is always greater than 0

(c) Applying Bernstein inequality with probability 1-$\delta$. $X_i$ refers to the CIS estimate for 1 sample.

(d) Grouping terms 1 and 2 together, where $U_c = max \frac{\pi_e(a|\tilde{c})}{\pi_b(a|\tilde{c})}$.

The first term of the variance dominates the second with increase in number of samples. Thus, Variance is of the complexity $\mathcal{O}(\frac{T^2 R_{max}^2 U_c^{2T}}{N})$

### E.1.7 Upper Bound on the MSE

$$MSE = Bias^2 + Variance \qquad (a)$$

$$\sim \mathcal{O}(\frac{TR_{max}U_c^T}{N})^2 + \epsilon(|\mathbb{E}_{\pi_e^c}[\hat{V}_{\pi_e}^{CIS}]|^2) + \mathcal{O}(\frac{T^2 R_{max}^2 U_c^{2T}}{N}) \qquad (b)$$

$$\sim \mathcal{O}(\frac{T^2 R_{max}^2 U_c^{2T}}{N}) + \epsilon(|\mathbb{E}_{\pi_e^c}[\hat{V}_{\pi_e}^{CIS}]|^2) \qquad (c)$$

$$\sim \mathcal{O}(\frac{T^2 R_{max}^2 U_s^{2T}}{N})K^{2T} + \epsilon(|\mathbb{E}_{\pi_e^c}[\hat{V}_{\pi_e}^{CIS}]|^2) \qquad (d)$$

The arguments are similar to the known-concept bounds of the MSE, with the difference being the expressions for $U_c, U_s, K$ are over approximations of concepts instead of true concepts and an irreducible error over $\mathbb{E}_{\pi_e^c}[\hat{V}_{\pi_e}^{CIS}]$ as the distribution is sampled in the concept representations instead of state representations.

### E.1.8 VARIANCE COMPARISON WITH MIS ESTIMATOR

**Theorem.** *When* $Cov(\prod_{t=0}^t \frac{\pi_e^c(a_t|c_t)}{\pi_b^c(a_t|c_t)}r_t, \prod_{t=0}^k \frac{\pi_e^c(a_t|c_t)}{\pi_b^c(a_t|c_t)}r_k) \leq Cov(\frac{d^{\pi_e}(s_t,a_t)}{d^{\pi_b}(s_t,a_t)}r_t, \frac{d^{\pi_e}(s_k,a_k)}{d^{\pi_b}(s_k,a_k)}r_k)$, *the variance is lower than the Variance of MIS estimator, i.e.* $\mathbb{V}_{\pi_b}[\hat{V}^{CIS}] \leq \mathbb{V}_{\pi_b}[\hat{V}^{MIS}]$.

***Proof:*** *We start from the assumption:*

$$Cov(\prod_{t=0}^T \frac{\pi_e^c(a_t|\tilde{c}_t)}{\pi_b^c(a_t|\tilde{c}_t)}r_t, \prod_{t=0}^T \frac{\pi_e^c(a_t|\tilde{c}_t)}{\pi_b^c(a_t|\tilde{c}_t)}r_k) = \mathbb{E}[\prod_{t=0}^T \frac{\pi_e^c(a_t|\tilde{c}_t)}{\pi_b^c(a_t|\tilde{c}_t)} \prod_{t=0}^T \frac{\pi_e^c(a_t|\tilde{c}_t)}{\pi_b^c(a_t|\tilde{c}_t)}r_t r_k] - \mathbb{E}[\prod_{t=0}^T \frac{\pi_e^c(a_t|\tilde{c}_t)}{\pi_b^c(a_t|\tilde{c}_t)}r_t]\mathbb{E}[\prod_{t=0}^T \frac{\pi_e^c(a_t|\tilde{c}_t)}{\pi_b^c(a_t|\tilde{c}_t)}r_k] \tag{a}$$

$$= \mathbb{E}[\prod_{t=0}^T \frac{\pi_e(a_t|s_t)}{\pi_b(a_t|s_t)} \prod_{t=0}^T \frac{\pi_e(a_t|s_t)}{\pi_b(a_t|s_t)}K^2 r_t r_k] - \mathbb{E}[\prod_{t=0}^T \frac{\pi_e(a_t|s_t)}{\pi_b(a_t|s_t)}K r_t]\mathbb{E}[\prod_{t=0}^T \frac{\pi_e(a_t|s_t)}{\pi_b(a_t|s_t)}K r_k] \tag{b}$$

$$\leq \mathbb{E}[\prod_{t=0}^T \frac{\pi_e(a_t|s_t)}{\pi_b(a_t|s_t)} \prod_{t=0}^T \frac{\pi_e(a_t|s_t)}{\pi_b(a_t|s_t)}r_t r_k] - \mathbb{E}[\prod_{t=0}^T \frac{\pi_e(a_t|s_t)}{\pi_b(a_t|s_t)}r_t]\mathbb{E}[\prod_{t=0}^T \frac{\pi_e(a_t|s_t)}{\pi_b(a_t|s_t)}r_k] \tag{c}$$

$$\leq \mathbb{E}[(\frac{d^{\pi_e}(s_t,a_t)}{d^{\pi_b}(s_t,a_t)})(\frac{d^{\pi_e}(s_k,a_k)}{d^{\pi_b}(s_k,a_k)})r_t r_k] - \mathbb{E}[\frac{d^{\pi_e}(s_t,a_t)}{d^{\pi_b}(s_t,a_t)}r_t]\mathbb{E}[\frac{d^{\pi_e}(s_k,a_k)}{d^{\pi_b}(s_k,a_k)}r_k] \tag{d}$$

**Explanation of steps:**

*(a) We begin with the definition of covariance.*

*(b),(c) Using the definition of $\pi^c$, with K (the ratio of state-space distribution ratio) < 1.*

*(c) Applying Lemma D.1.7 to both the terms.*

*Finally, using Lemma D.1.7, substituting $Y_t = \prod_{t'=0}^T \left(\frac{\pi_e(a_{t'}|s_{t'})}{\pi_b(a_{t'}|s_{t'})}\right) r_t$ and $X_t = s_t, a_t, r_t$ completes our proof.*

### E.2 PDIS

#### E.2.1 BIAS

$$Bias = |\mathbb{E}_{\pi_b}[\hat{V}_{\pi_e}^{CPDIS}] - \mathbb{E}_{\pi_e^c}[\hat{V}_{\pi_e^c}^{CPDIS}]| \tag{a}$$

$$= |\mathbb{E}_{\pi_b}\left[\sum_{t=0}^T \gamma^t \rho_{0:t}^{(n)} r_t^{(n)}\right] - \mathbb{E}_{\pi_e^c}[\hat{V}_{\pi_e^c}^{CPDIS}]| \tag{b}$$

$$= |\sum_{n=1}^N \left(\prod_{t=0}^T \pi_b(a_t^{(n)}|s_t^{(n)})\right) \sum_{t=0}^T \gamma^t \rho_{0:t}^{(n)} r_t^{(n)} - \mathbb{E}_{\pi_e^c}[\hat{V}_{\pi_e^c}^{CPDIS}]| \tag{c}$$

$$= |\sum_{n=1}^N \sum_{t=0}^T \gamma^t \left(\prod_{t'=0}^t \pi_e^c(a_{t'}^{(n)}|\tilde{c}_{t'}^{(n)})(\frac{\pi_b(a_{t'}^{(n)}|s_{t'}^{(n)})}{\pi_b(a_{t'}^{(n)}|\tilde{c}_{t'}^{(n)})})\right) r_t^{(n)} - \mathbb{E}_{\pi_e^c}[\hat{V}_{\pi_e^c}^{CPDIS}]| \tag{d}$$

**Explanation of steps:** Similar to CIS.

#### E.2.2 VARIANCE

Following the process similar to CIS estimator:

$$\mathbb{V}[\hat{V}_{\pi_b}^{CPDIS}] = \mathbb{E}_{\pi_b}[(\hat{V}_{\pi_b}^{CPDIS})^2] - (\mathbb{E}_{\pi_b}[\hat{V}_{\pi_b}^{CPDIS}])^2 \tag{a}$$

We first evaluate the expectation of the square of the estimator:

$$\mathbb{E}_{\pi_b}[(\hat{V}_{\pi_b}^{CPDIS})^2] = \mathbb{E}_{\pi_b}\left[\left(\sum_{t=0}^{T}\gamma^t\rho_{0:t}r_t\right)^2\right] \tag{b}$$

$$= \mathbb{E}_{\pi_b}\left[\sum_{t=0}^{T}\sum_{t'=0}^{T}\rho_{0:t}\rho_{0:t'}\gamma^{(t+t')}r_t r_{t'}\right] \tag{c}$$

$$= \sum_{n=1}^{N}\sum_{t=0}^{T}\sum_{t'=0}^{T}\left(\prod_{t''=0}^{t}\pi_b(a_t|s_t)(\frac{\pi_e^c(a_t|\tilde{c}_t)}{\pi_b^c(a_t|\tilde{c}_t)})\right)\left(\prod_{t'''=0}^{t'}(\frac{\pi_e^c(a_t|\tilde{c}_t)}{\pi_b^c(a_t|\tilde{c}_t)})\right)\gamma^{(t+t')}r_t r_{t'} \tag{d}$$

Evaluating the second term in the variance expression:

$$(\mathbb{E}_{\pi_b}[\hat{V}_{\pi_b}^{CPDIS}])^2 = \left(\sum_{n=1}^{N}\sum_{t=0}^{T}\mathbb{E}_{\pi_b}[\gamma^t\rho_{0:t}r_t]\right)^2 \tag{e}$$

$$= \sum_{n=1}^{N}\sum_{t=0}^{T}\sum_{t'=0}^{T}\left(\prod_{t''=0}^{t}\pi_b(a_{t''}|s_{t''})(\frac{\pi_e^c(a_{t''}|\tilde{c}_{t''})}{\pi_b^c(a_{t''}|\tilde{c}_{t''})})\right)\left(\prod_{t'''=0}^{t'}\pi_b(a_{t'''}|s_{t'''})(\frac{\pi_e^c(a_{t'''}|\tilde{c}_{t'''})}{\pi_b^c(a_{t'''}|\tilde{c}_{t'''})})\right)\gamma^{(t+t')}r_t r_{t'} \tag{f}$$

Subtracting the squared expectation from the expectation of the squared estimator:

$$\mathbb{V}[\hat{V}_{\pi_b}^{CPDIS}] = \sum_{n=1}^{N}\sum_{t=0}^{T}\sum_{t'=0}^{T}\left(\prod_{t'''=0}^{t}\pi_b(a_{t'''}|s_{t'''})(\frac{\pi_e^c(a_{t'''}|\tilde{c}_{t'''})}{\pi_b^c(a_{t'''}|\tilde{c}_{t'''})})\right)\left(\prod_{t''=0}^{t'}(1-\pi_b(a_{t''}|s_{t''}))(\frac{\pi_e^c(a_{t''}|\tilde{c}_{t''})}{\pi_b^c(a_{t''}|\tilde{c}_{t''})})\right)\gamma^{(t+t')}r_t r_{t'} \tag{g}$$

**Explanation of steps: Similar to CIS**

### E.2.3 VARIANCE COMPARISON BETWEEN UNKNOWN CPDIS AND PDIS ESTIMATORS

**Theorem E.1.** *When* $Cov(\prod_{t=0}^{t}\frac{\pi_e^c(a_t|c_t)}{\pi_b^c(a_t|c_t)}r_t, \prod_{t=0}^{k}\frac{\pi_e^c(a_t|c_t)}{\pi_b^c(a_t|c_t)}r_k) \leq$ $Cov(\prod_{t=0}^{k}\frac{\pi_e(a_t|s_t)}{\pi_b(a_t|s_t)}r_t, \prod_{t=0}^{k}\frac{\pi_e(a_t|s_t)}{\pi_b(a_t|s_t)}r_k)$, *the variance of parameterized CPDIS estimators is lower than PDIS estimator, i.e.* $\mathbb{V}_{\pi_b}[\hat{V}^{CPDIS}] \leq \mathbb{V}_{\pi_b}[\hat{V}^{PDIS}]$.

**Proof:** Similar to CIS estimator.

### E.2.4 UPPER BOUND ON THE BIAS

Unlike known concepts, there exists a finite bias in case of unknown concepts, and the bounds need to be analyzed.

$$Bias = |\sum_{n=1}^{N}\sum_{t=0}^{T}\gamma^t\left(\prod_{t'=0}^{t}\pi_e^c(a_{t'}^{(n)}|\tilde{c}_{t'}^{(n)})(\frac{\pi_b(a_{t'}^{(n)}|s_{t'}^{(n)})}{\pi_b^c(a_{t'}^{(n)}|\tilde{c}_{t'}^{(n)})})\right)r_t^{(n)} - \mathbb{E}_{\pi_e^c}[\hat{V}_{\pi_e}^{CPDIS}]| \tag{a}$$

$$\leq |\sum_{n=1}^{N}\sum_{t=0}^{T}\gamma^t\left(\prod_{t'=0}^{t}\pi_e^c(a_{t'}^{(n)}|\tilde{c}_{t'}^{(n)})(\frac{\pi_b(a_{t'}^{(n)}|s_{t'}^{(n)})}{\pi_b^c(a_{t'}^{(n)}|\tilde{c}_{t'}^{(n)})})\right)r_t^{(n)} + |\mathbb{E}_{\pi_e^c}[\hat{V}_{\pi_e}^{CPDIS}]| \tag{b}$$

$$\leq \frac{1}{N}\sum_{n=1}^{N}\sum_{t=0}^{T}\gamma^t\prod_{t'=0}^{t}(\frac{\pi_e^c(a_{t'}^{(n)}|\tilde{c}_{t'}^{(n)})}{\pi_b^c(a_{t'}^{(n)}|\tilde{c}_{t'}^{(n)})})r_t^{(n)} + \frac{7TR_{max}U_c^T ln(\frac{2}{\delta})}{3(N-1)} + \sqrt{\frac{ln(\frac{2}{\delta})}{N^3-N^2}\sum_{i<j}^{N}(X_i-X_j)^2} + \mathbb{E}_{\pi_e^c}[\hat{V}_{\pi_e}^{CPDIS}] \tag{c}$$

$$\leq TR_{max}U_c^T(\frac{1}{N} + \frac{ln\frac{2}{\delta}}{3(N-1)}) + \sqrt{\frac{ln(\frac{2}{\delta})}{N^3-N^2}\sum_{i<j}^{N}(X_i-X_j)^2} + |\mathbb{E}_{\pi_e^c}[\hat{V}_{\pi_e}^{CPDIS}]| \tag{d}$$

**Explanation of steps:** Similar to CIS.

### E.2.5 UPPER BOUND ON THE VARIANCE

$$\mathbb{V}[\hat{V}_{\pi_b}^{CPDIS}] = \mathbb{E}_{\pi_b}\left(\left(\sum_{t=0}^{T}\gamma^t r_t \prod_{t'=0}^{t}\frac{\pi_e^c(a_{t'}|\tilde{c}_{t'})}{\pi_b^c(a_{t'}|\tilde{c}_{t'})}\right)^2\right) - \mathbb{E}_{\pi_b}\left(\sum_{t=0}^{T}\gamma^t r_t \prod_{t'=0}^{t}\frac{\pi_e^c(a_{t'}|\tilde{c}_{t'})}{\pi_b^c(a_{t'}|\tilde{c}_{t'})}\right)^2 \qquad \text{(a)}$$

$$\leq \mathbb{E}_{\pi_b}\left(\left(\sum_{t=0}^{T}\gamma^t r_t \prod_{t'=0}^{t}\frac{\pi_e^c(a_{t'}|\tilde{c}_{t'})}{\pi_b^c(a_{t'}|\tilde{c}_{t'})}\right)^2\right) \qquad \text{(b)}$$

$$\leq \frac{1}{N}\sum_{n=1}^{N}\left(\left(\sum_{t=0}^{T}\gamma^t r_t \prod_{t'=0}^{t}\frac{\pi_e^c(a_{t'}|\tilde{c}_{t'})}{\pi_b^c(a_{t'}|\tilde{c}_{t'})}\right)^2\right) + \frac{7T^2 R_{max}^2 U_c^{2T}ln(\frac{2}{\delta})}{3(N-1)} + \sqrt{\frac{ln(\frac{2}{\delta})}{N^3-N^2}\sum_{i<j}^{N}(X_i^2-X_j^2)^2} \qquad \text{(c)}$$

$$\leq T^2 R_{max}^2 U_c^{2T}\left(\frac{1}{N}+\frac{ln\frac{2}{\delta}}{3(N-1)}\right) + \sqrt{\frac{ln(\frac{2}{\delta})}{N^3-N^2}\sum_{i<j}^{N}(X_i^2-X_j^2)^2} \qquad \text{(d)}$$

**Explanation of steps:** Similar to CIS.

### E.2.6 UPPER BOUND ON THE MSE

$$MSE = Bias^2 + Variance \qquad \text{(a)}$$

$$\sim \mathcal{O}(\frac{TR_{max}U_c^T}{N})^2 + \epsilon(|\mathbb{E}_{\pi_e^c}[\hat{V}_{\pi_e}^{CPDIS}]|^2) + \mathcal{O}(\frac{T^2 R_{max}^2 U_c^{2T}}{N}) \qquad \text{(b)}$$

$$\sim \mathcal{O}(\frac{T^2 R_{max}^2 U_c^{2T}}{N}) + \epsilon(|\mathbb{E}_{\pi_e^c}[\hat{V}_{\pi_e}^{CPDIS}]|^2) \qquad \text{(c)}$$

$$\sim \mathcal{O}(\frac{T^2 R_{max}^2 U_s^{2T}}{N})K^{2T} + \epsilon(|\mathbb{E}_{\pi_e^c}[\hat{V}_{\pi_e}^{CPDIS}]|^2) \qquad \text{(d)}$$

The arguments are similar to the known-concept bounds of the MSE, with the difference being the expressions for $U_c, U_s, K$ are over approximations of concepts instead of true concepts and an irreducible error over $\mathbb{E}_{\pi_e^c}[\hat{V}_{\pi_e}^{CPDIS}]$ as the distribution is sampled in the concept representations instead of state representations.

### E.2.7 VARIANCE COMPARISON WITH MIS ESTIMATOR

**Theorem.** *When* $Cov(\prod_{t=0}^{t}\frac{\pi_e^c(a_t|c_t)}{\pi_b^c(a_t|c_t)}r_t, \prod_{t=0}^{k}\frac{\pi_e^c(a_t|c_t)}{\pi_b^c(a_t|c_t)}r_k) \leq Cov(\frac{d^{\pi_e}(s_t,a_t)}{d^{\pi_b}(s_t,a_t)}r_t, \frac{d^{\pi_e}(s_k,a_k)}{d^{\pi_b}(s_k,a_k)}r_k)$, *the variance is lower than the Variance of MIS estimator, i.e.* $\mathbb{V}_{\pi_b}[\hat{V}^{CPDIS}] \leq \mathbb{V}_{\pi_b}[\hat{V}^{MIS}]$.

***Explanation of Steps:** Similar to CIS*

## F ENVIRONMENTS

**WindyGridworld** Figure 6 illustrates the Windy Gridworld environment, a 20x20 grid divided into regions with varying wind directions and penalties. The agent's goal is to navigate from a randomly chosen starting point to a fixed goal in the top-right corner. Off-diagonal winds increase in strength near non-windy regions, affecting the agent's movement. Each of the four available actions moves the agent four steps in the chosen direction. Reaching the goal earns a +5 reward while moving away results in a -0.2 penalty. Additional negative rewards are based on regional penalties within the grid. Each episode ends after 200 steps.

The grid is split into 25 blocks, each measuring 4x4 units with each region having a penalty based on the wind-strength. Blocks affected by wind display the direction and strength (e.g., '←↑ (-2,+2)' indicates northward and westward winds with a strength of 2 units each). This setup encourages the agent to navigate through non-penalty areas for optimal rewards.

Figure 6: Schematic of windy-gridworld environment. The top-right corner refers to the goal target of the agent. The wind direction and reward penalty is indicated in each region.

**MIMIC-III** We use the publicly available MIMIC-III database (Johnson et al., 2016) from PhysioNet (Goldberger et al., 2000), which records the treatment and progression of ICU patients at the Beth Israel Deaconess Medical Center in Boston, Massachusetts. We focus on the task of managing acutely hypotensive patients in the ICU. Our preprocessing follows the original MIMIC-III steps detailed in Komorowski et al. (2018c) and used in subsequent works (Keramati et al., 2021b; Matsson & Johansson, 2021). After processing the data in Excel, we group patients by 'ICU-stayID' to form distinct trajectories.

The state space includes 15 features: Creatinine, $FiO_2$, Lactate, Partial Pressure of Oxygen, Partial Pressure of $CO_2$, Urine Output, GCS score, and electrolytes such as Calcium, Chloride, Glucose, $HCO_3$, Magnesium, Potassium, Sodium, and $SpO_2$. Each feature is binned into 10 levels from 0 (very low) to 9 (very high).

Treatments for hypotension include IV fluid bolus administration and vasopressor initiation, with doses categorized into four levels: "none," "low," "medium," and "high," forming a total action space of 16 discrete actions. The reward function depends on the next mean arterial pressure (MAP) and ranges from -1 to 0, linearly distributed between 20 and 65. A MAP above 65 indicates that the patient is not experiencing hypotension.

## G  ADDITIONAL EXPERIMENTAL DETAILS

### G.1  UNKNOWN CONCEPTS EXPERIMENTAL SETUP

**Environments, Policy descriptions, Metrics:** Same as those in known concepts section.

**Training and Hyperparameter Details:** We use 400 training, 50 validation, and 50 test trajectories sampled from the behavior policy to train the CBMs, which predict the next state transitions from the current state. The model architecture includes an input layer, a bottleneck, two 256-neuron layers, and an output layer, all with ReLU activations. Training is performed using the Adam Optimizer with a learning rate of 1e-3 on an Nvidia P100 GPU (16 GB) within the Pytorch framework.

Training targets multiple loss components: the OPE metric, interpretability, diversity, and CBM output. The non-convex nature of the loss landscape can lead to issues such as non-convergence and NaN values. To address this, we employ a three-stage training strategy.

In the first stage, we optimize all losses except the OPE metric to stabilize the initial training process. In the second stage, the OPE metric is gradually incorporated into the optimization until convergence is achieved. Finally, in the third stage, we freeze the CBM weights to refine the remaining losses while controlling variations in the OPE metric.

This strategy balances the learning of critical on-policy features with maintaining relevant OPE metrics, thereby enhancing concept learning and policy generalization. Despite these efforts, managing the complexity of the loss landscape remains a significant challenge, particularly in dynamic environments, and represents an important direction for future research.

### G.2  KNOWN, ORACLE AND INTERVENED CONCEPTS FOR WINDYGRIDWORLD

| X | Y | Known Concept | Oracle Concept | Optimized Concept |
|---|---|---|---|---|
| (0,4) | (0,4) | 0 | 0 | 0 |
| (4,8) | (0,4) | 1 | 1 | 1 |
| (8,12) | (0,4) | 2 | 1 | 1 |
| (12,16) | (0,4) | 3 | 1 | 0 |
| (16,20) | (0,4) | 4 | 1 | 1 |
| (0,4) | (4,8) | 5 | 2 | 2 |
| (4,8) | (4,8) | 6 | 0 | 0 |
| (8,12) | (4,8) | 7 | 1 | 1 |
| (12,16) | (4,8) | 8 | 1 | 1 |
| (16,20) | (4,8) | 9 | 1 | 0 |
| (0,4) | (8,12) | 10 | 2 | 2 |
| (4,8) | (8,12) | 11 | 2 | 2 |
| (8,12) | (8,12) | 12 | 0 | 0 |
| (12,16) | (8,12) | 13 | 1 | 1 |
| (16,20) | (8,12) | 14 | 1 | 1 |
| (0,4) | (12,16) | 15 | 2 | 2 |
| (4,8) | (12,16) | 16 | 2 | 2 |
| (8,12) | (12,16) | 17 | 2 | 2 |
| (12,16) | (12,16) | 18 | 3 | 3 |
| (16,20) | (12,16) | 19 | 3 | 3 |
| (0,4) | (16,20) | 20 | 2 | 2 |
| (4,8) | (16,20) | 21 | 2 | 2 |
| (8,12) | (16,20) | 22 | 2 | 2 |
| (12,16) | (16,20) | 23 | 2 | 2 |
| (16,20) | (16,20) | 24 | 3 | 3 |

Table 1: WindyGridworld Concept Information

### G.3 ADDITIONAL DESCRIPTION ON POLICES FOR MIMIC-III

For the MIMIC-III dataset, it is common to generate behavior trajectories using K-nearest neighbors (KNN) as the true on-policy trajectories are unavailable. Examples of works that generate behavior trajectories or policies using KNNs include (Gottesman et al., 2020; Böck et al., 2022; Liu et al., 2022; Keramati et al., 2021b; Komorowski et al., 2018b; Peine et al., 2021). In this paper, we employ a popular variant of KNN, known as approximate nearest neighbors (ANN) search.

The advantages of ANN over traditional KNN include scalability, reduced computational cost, efficient indexing, and support for dynamic data. These benefits allow us to generate behavior and evaluation policies with a larger number of neighbors (200 in this study, which is double that used in prior works employing KNN) while achieving faster inference times. Examples of papers that use approximate nearest neighbors in medical applications include (Anagnostou et al., 2020; Gupta et al., 2022). For readers interested in the foundational work outlining the benefits of ANN over KNN, we refer to the seminal paper (Indyk & Motwani, 1998).

## H ABLATION EXPERIMENTS

### H.1 STATE ABSTRACTION CLUSTERING BASELINE

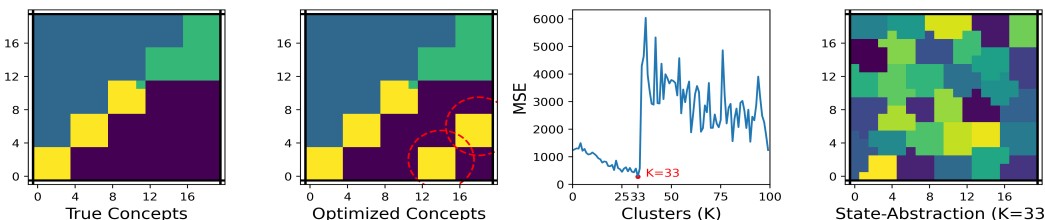

Figure 7: Comparison of learned concepts with state abstraction clusters: The first two subplots in Figure 7 show the true oracle concepts and the optimized concepts obtained using the methodology described in the main paper. The third subplot illustrates the OPE performance as the number of state clusters varies, showing improvement up to $K = 33$ clusters, followed by a spike in MSE and then a gradual improvement. The final subplot visualizes the state clusters for $K = 33$, the best-performing state abstraction for OPE. These clusters lack correspondence with the true oracle concepts or the optimized concepts, highlighting that learned concepts capture more meaningful and useful information compared to state abstractions.

In this subsection, we present an ablation study to compare the performance of OPE under concept-based representations versus state abstractions. This experiment is conducted in the Windy Gridworld environment. For state abstractions, we apply K-means clustering on the state representations (coordinates $(x, y)$) with varying values of $K$. The results are summarized in Figure 7.

We plot the mean squared error (MSE) of the OPE across different numbers of state abstraction clusters. Initially, the MSE decreases as the number of clusters increases, but it eventually exhibits a sudden rise followed by a downward trend as $K$ grows further. The minimum MSE is observed at $K = 33$. Upon inspecting the clusters for $K = 33$, we find that they primarily correspond to local geographical regions, showing no alignment with meaningful features such as the distance from the goal, wind penalty, etc.

These clusters differ significantly from the learned concepts shown in Figure 7. Moreover, they are neither readily interpretable nor easily amenable to intervention, highlighting the importance of using concept-based representations for OPE.

### H.2 IMPERFECT CONCEPTS BASELINE

In this ablation study, we evaluate the performance of concept-based OPE when the quality of concepts is poor or imperfect. Using the Windy Gridworld environment as an example, we define concepts as functions solely of the horizontal distance to the target. This approach neglects critical

information such as vertical distance to the target, wind effects, and region penalties. As a result, these concepts violate one of the primary desiderata: diversity. By capturing only one important concept dimension while disregarding others, these poor concepts fail to represent the full complexity of the environment.

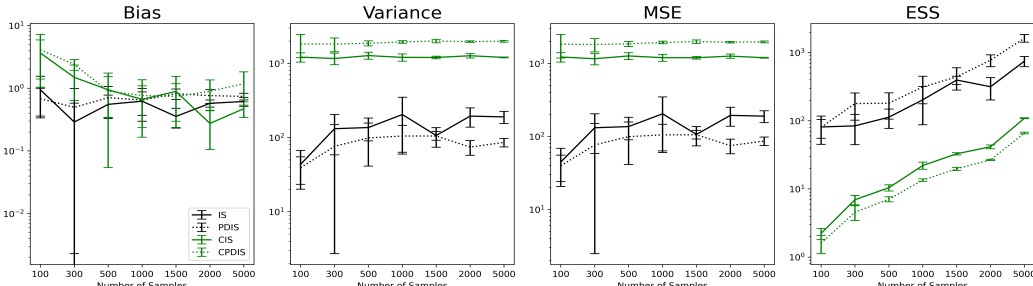

Figure 8: Imperfect concepts baseline. We consider a scenario where the concepts are just function of the horizontal distance to target, thus ignoring vital information like vertical distance to target, wind regions, thus lacking diversity, one of the important desiderata. We observe the OPE performance to be poor compared to traditional estimators, with higher bias, variance, MSE and lower ESS.

Figure 8 presents our results with suboptimal concepts. We observe that suboptimal concepts exhibit inferior OPE characteristics, including higher bias, variance, and MSE, as well as lower ESS, compared to traditional OPE estimators. This demonstrates that not all concept-based estimators lead to improved performance; the quality of the concepts plays a crucial role, which is closely tied to the desiderata they satisfy. Furthermore, this highlights the importance of having an algorithm capable of learning concepts with favorable OPE characteristics, especially in scenarios involving imperfect experts or highly complex domains where obtaining expertise is challenging. Nevertheless, poor concepts still allow for potential interventions, as the root cause of the poor OPE characteristics can be readily identified.

### H.3 INVERSE PROPENSITY SCORES COMPARISON BETWEEN CONCEPTS AND STATE REPRESENTATIONS

In this ablation study, we compare the inverse propensity scores (IS ratios) for concepts and states in the Windy Gridworld environment, focusing on known concepts. While the analysis is specific to this environment, the insights generalize to other domains. From Figure 9, we observe that the IPS scores under concepts are skewed more towards the left compared to those under states. Quantitatively, there is a reduction of nearly 1–2 orders of magnitude in the IPS scores. This highlights that the variance reduction achieved with concepts is directly linked to lower IPS scores, demonstrating a better characterization under concepts compared to states.

## I    OPTIMIZED PARAMETERIZED CONCEPTS

Table 2: WindyGridworld: Coefficients of the human interpretable features learnt while optimizing parameterized concepts. Here, the concept $c_t$ is a 4-dimensional vector $[c_1, c_2, c_3, c_4]$, where $c_i = w_i^T f_i$, with $f_i$ being the human interpretable features.

| Feature | CIS | | | | CPDIS | | | |
|---|---|---|---|---|---|---|---|---|
| | $c_1$ | $c_2$ | $c_3$ | $c_4$ | $c_1$ | $c_2$ | $c_3$ | $c_4$ |
| $f_1$: X-coordinate | 0.15 | -0.07 | 0.05 | 0.19 | -0.23 | 0.33 | -0.03 | 0.03 |
| $f_2$: Y-coordinate | -0.02 | -0.23 | 0.07 | -0.12 | -0.22 | 0.25 | 0.02 | -0.06 |
| $f_3$: Horizontal distance from target | -0.02 | 0.07 | -0.10 | 0.00 | -0.15 | -0.30 | 0.02 | -0.11 |
| $f_4$: Vertical distance from target | 0.06 | -0.26 | -0.09 | 0.06 | -0.11 | 0.10 | -0.04 | -0.21 |
| $f_5$: Horizontal Wind | 0.05 | 0.12 | -0.12 | 0.00 | -0.15 | 0.20 | 0.29 | -0.14 |
| $f_6$: Vertical Wind | 0.26 | 0.01 | -0.02 | 0.00 | -0.18 | 0.06 | -0.17 | 0.19 |
| $f_7$: Region penalty | 0.24 | 0.18 | -0.25 | 0.15 | 0.23 | 0.01 | -0.11 | 0.22 |
| $f_8$: Distance to left wall | -0.14 | -0.25 | 0.01 | 0.05 | -0.13 | 0.24 | 0.16 | 0.14 |
| $f_9$: Distance to right wall | 0.02 | 0.00 | 0.01 | 0.19 | -0.12 | -0.28 | 0.06 | 0.16 |
| $f_{10}$: Distance to top wall | -0.01 | -0.20 | -0.21 | 0.07 | -0.33 | -0.05 | -0.04 | -0.01 |
| $f_{11}$: Distance to bottom wall | -0.16 | 0.07 | 0.22 | -0.22 | 0.06 | -0.13 | 0.13 | -0.22 |
| $f_{12}$: Penalty of left subregion | -0.06 | 0.08 | -0.08 | -0.22 | -0.07 | -0.01 | 0.03 | -0.16 |
| $f_{13}$: Penalty of right subregion | -0.03 | 0.02 | -0.20 | -0.20 | -0.07 | -0.18 | -0.34 | -0.21 |
| $f_{14}$: Penalty of top subregion | 0.16 | 0.19 | -0.08 | -0.17 | 0.00 | 0.04 | -0.07 | 0.21 |
| $f_{15}$: Penalty of bottom subregion | 0.08 | 0.24 | 0.05 | -0.19 | 0.17 | -0.07 | -0.12 | 0.21 |
| $f_{16}$: Distance to left subregion | -0.11 | 0.05 | 0.00 | 0.26 | 0.10 | -0.07 | 0.22 | 0.04 |
| $f_{17}$: Distance to right subregion | 0.00 | -0.17 | 0.04 | 0.13 | 0.05 | -0.13 | 0.06 | 0.11 |
| $f_{18}$: Distance to top subregion | 0.07 | -0.03 | 0.13 | 0.08 | -0.12 | 0.01 | 0.06 | 0.00 |
| $f_{19}$: Distance to bottom subregion | -0.06 | -0.09 | -0.06 | -0.01 | -0.19 | -0.01 | 0.06 | 0.13 |
| Constant | -0.06 | -0.16 | 0.14 | -0.01 | -0.08 | -0.01 | 0.00 | 0.13 |

Table 3: MIMIC: Coefficients of the human interpretable features learnt while optimizing parameterized concepts.

| Feature | CIS | | | | CPDIS | | | |
|---|---|---|---|---|---|---|---|---|
| | $c_1$ | $c_2$ | $c_3$ | $c_4$ | $c_1$ | $c_2$ | $c_3$ | $c_4$ |
| $f_1$: Creatinine | -0.08 | -0.24 | 0.19 | -0.18 | -0.08 | -0.24 | 0.19 | -0.18 |
| $f_2$: FiO$_2$ | -0.13 | 0.00 | 0.04 | -0.06 | -0.13 | 0.00 | 0.04 | -0.06 |
| $f_3$: Lactate | -0.24 | -0.02 | -0.23 | 0.21 | -0.24 | -0.02 | -0.23 | 0.21 |
| $f_4$: Partial Pressure of O$_2$ | 0.09 | -0.07 | -0.06 | -0.12 | 0.09 | -0.07 | -0.06 | -0.12 |
| $f_5$: Partial Pressure of CO$_2$ | -0.21 | 0.16 | 0.19 | -0.03 | -0.21 | 0.16 | 0.19 | -0.03 |
| $f_6$: Urine Output | 0.06 | 0.07 | 0.06 | 0.22 | 0.06 | 0.07 | 0.06 | 0.22 |
| $f_7$: GCS Score | 0.11 | -0.05 | -0.01 | 0.15 | 0.11 | -0.05 | -0.01 | 0.15 |
| $f_8$: Calcium | 0.16 | -0.20 | 0.06 | 0.16 | 0.16 | -0.20 | 0.06 | 0.16 |
| $f_9$: Chloride | 0.02 | -0.11 | -0.04 | 0.14 | 0.02 | -0.11 | -0.04 | 0.14 |
| $f_{10}$: Glucose | 0.06 | -0.10 | -0.10 | -0.08 | 0.05 | -0.10 | -0.10 | -0.08 |
| $f_{11}$: HCO$_2$ | 0.21 | 0.14 | -0.20 | -0.22 | 0.20 | 0.14 | -0.20 | -0.22 |
| $f_{12}$: Magnesium | -0.15 | -0.02 | -0.20 | 0.01 | -0.15 | -0.02 | -0.20 | 0.01 |
| $f_{13}$: Potassium | 0.04 | 0.08 | 0.15 | -0.26 | 0.04 | 0.08 | 0.15 | -0.26 |
| $f_{14}$: Sodium | 0.00 | -0.02 | 0.24 | 0.19 | 0.00 | -0.02 | 0.24 | 0.19 |
| $f_{15}$: SpO$_2$ | -0.17 | -0.20 | -0.06 | -0.23 | -0.17 | -0.20 | -0.06 | -0.23 |

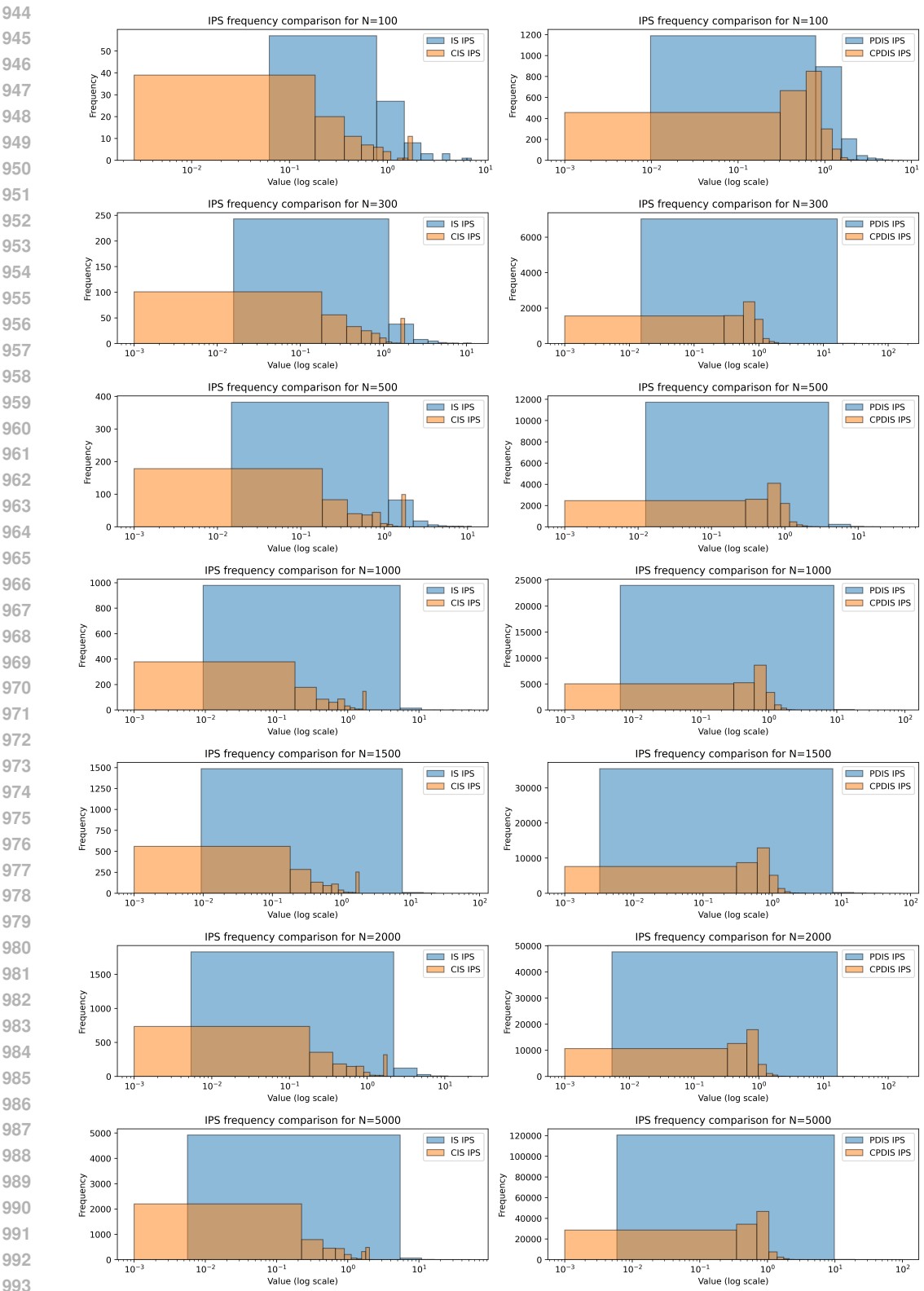

Figure 9: Inverse propensity score comparisons under concepts and states. Column 1 represents IPS score comparison between CIS and IS, while column 2 is the IPS score comparison between CPDIS and PDIS. Rows indicate varying number of trajectories. We observe, across all trajectory samples, the frequency of the lower IPS scores are left skewed in case of concepts over states. This indicates the source of variance reduction in concepts infact lies in the lowered IPS scores.

