# OpenReview forum: "Concept-driven Off Policy Evaluation"
_ICLR.cc/2025/Conference — Submitted to ICLR 2025_

### Official Review · Reviewer_JqPf · 2024-11-03

**Soundness:** 2
**Presentation:** 2
**Contribution:** 2
**Rating:** 5
**Confidence:** 2

**Summary:**

The paper introduces concept-driven off-policy evaluation (OPE). This approach mirrors the structure of concept bottleneck models, where the prediction task is split into two stages: first, predicting a set of human-interpretable concepts, and then using these concepts to complete the prediction task. The paper applies this concept-driven approach to OPE. Specifically, it maps trajectories to a concept space, derives a policy for each concept value, and uses the concept equivalent behavior and evaluation policies to calculate the importance sampling ratio. When concepts are not known in advance, the paper presents an algorithm to learn them. The evaluation is performed for specific policies on both synthetic and real datasets.

**Strengths:**

Importance smapling methods for OPE can suffer from high variance and using a concept bottelenck can reduce the variance. The concepts can also improve the interpretability of the evaluation task. The proposed idea is novel to the best of my knowledge.

**Weaknesses:**

I found this paper challenging to read due to numerous typos and mathematical inconsistencies. For example, in Section 4.1, $\phi$ is first introduced as a function of state, action, reward, and next state. Later, it’s interpreted as a function of the full trajectory. However, in Equation 1, $\phi^{-1}$ appears to return a state, creating confusion. Additionally, there’s a typo in the definition of w in Equation 1.
Theoretical results are presented with virtually no explanation. In both Sections 5.1 and 6.2, four theorems are listed in row without any discussion, making it difficult to follow or interpret the results and understand their limitations (for instance, Theorem 5.2 suggests that concept-driven evaluation is biased, which doesn’t align with the evaluations). It’s generally known that IS estimators can exhibit high variance, and various methods have been proposed to reduce this variance at the cost of extra bias. This paper’s proposed method falls into this category, though this is not clearly acknowledged.

There are also choices in the paper that lack justification. For instance, it uses an approximate nearest-neighbor policy with the MIMIC-III dataset. Having worked extensively with this dataset and reviewed recent studies on it across different tasks, I am not aware of any example where such a policy is applied. Additionally, this experiment employs a concept space of size $10^{15}$, which does not seem to offer interpretability and appears far removed from the motivating example in Figure 1.

**Questions:**

Please refer to weaknesses.

---

> ### Author Response · Authors · 2024-11-16
> **Authors Response to weaknesses**
>
> We sincerely thank the reviewer JqPf for the review and comments. We will sequentially address the questions.
>
> Q: I found this paper challenging to read due to numerous typos and mathematical inconsistencies. For example, in Section 4.1, ϕ is first introduced as a function of state, action, reward, and next state. Later, it’s interpreted as a function of the full trajectory. However, in Equation 1, ϕ−1appears to return a state, creating confusion. Additionally, there’s a typo in the definition of w in Equation 1.
>
> Response: We acknowledge there is an easier way to define a concept-based policy, without making it as complicated as Eqn 1 and rectifying notation. We make the changes in the latest version of the draft, section 4.1.
>
> To summarize, the concept function $\phi$ maps trajectory histories $h_t$ to concepts $c_t$. This function $\phi$ can capture various vital information in the history such as transition dynamics, short-term rewards, influential states, interdependencies in actions across timesteps, etc. Without loss of generality, in this work, we consider concepts $c_t$ to be just functions of current state $s_t$. This assumption considers the scenario where concepts capture important information based on the criticalness of the state. We provide this clarification in the latest draft of the paper, section 4.1.
>
> Concept-based policies $\pi^c_e, \pi^c_b$ are policies that are conditioned on concepts instead of states, where the concepts satisfy the desiderata. We make an additional assumption (5.2) on these concept-based policies. This assumption states that given a state, the concept-based policies $\pi^c_e, \pi^c_b$ are allowed to differ from the traditional policies $\pi_e,\pi_b$ by at most a quantity of $\beta$, i.e. $|\pi^c_e(a|c)-\pi_e(a|s)|<\beta$ and $|\pi^c_b(a|c)-\pi_b(a|s)|<\beta$. This is to ensure that the evaluation policy $π_e$ under concepts is reflective of the original policy while allowing to satisfy the concept desiderata depending on the flexibility of $\beta$. The quantity $\beta$ is defined at the discretion of the practitioner.
>
> Hopefully, this provides more clarity on the objective of the paper, and we are open to more suggestions on where we can improve.
>
> Q: Theoretical results are presented with virtually no explanation. In both Sections 5.1 and 6.2, four theorems are listed in row without any discussion, making it difficult to follow or interpret the results and understand their limitations (for instance, Theorem 5.2 suggests that concept-driven evaluation is biased, which doesn’t align with the evaluations).
>
> Response: We provide more detailed explanations to each of the Assumptions and Theorems in Sections 5 and 6, in the revised draft. These also cater to insights from theory and help design concepts and their corresponding policies practically. As a summary,
>
> 5.1 (Lines 210-213): This talks about the absolute continuity assumption, that every state-action pair that appears in the batch policy is capable of being evaluated with a positive probability.
>
> 5.2 (Lines 214-220): The state and concept policies differ by most $\beta$, to ensure that the evaluation policy $π^c_e$ under concepts is reflective of the original policy $\pi_e$.
>
> 5.3 (Lines 221-222): Known concepts are unbiased -> This is because the change of measure theorem from $\pi_b$ to $\pi^c_b$ is possible. However, this holds for infinite trajectories, and bias creeps in when there are finite trajectories.
>
> 5.4 (Lines 224-230): Variance under known concepts is lower than traditional IS, under the covariance assumption. We discuss why the covariance assumption can be realized better under concept representations as opposed to the state representations, in the revised draft of the paper.
>
> 5.5 (Lines 232-234): Variance Comparison with the MIS Estimator: We discuss how the covariance assumption under concept IS-product ratios can result in lower variance, even compared to the more efficient state-distribution ratios, which don’t suffer from the curse of the horizon with increasing trajectory length.
>
> 5.6 (Lines 236-243): Cramer-Rao bounds of our estimators: We explicitly discuss the scenario when the worst-case bounds tighten when moving from states to concepts. The crux is in the ratio $\frac{\pi_e(a|s)}{\pi_b(a|s)}$ where $\pi_b(a|s)$ is very low due to some states being poorly sampled. Under concepts, this state is better characterized which leads to an improved value of $\pi^c_b(a|c)$, which improves the denominator and reduces the IS ratio, which in turn improves the bounds. (We provide a short explanation right after stating the theorem in the main text and a detailed explanation in Appendix D.1.7.
>
> 6.1 (Lines 362-366): Unknown concepts are biased. We discuss why unknown concepts are biased, as change of measure theorem is not applicable due to unknown probability $\pi^c_b$.

---

> ### Author Response · Authors · 2024-11-16
> **Author response to weaknesses (continued)**
>
> 6.2, 6.3 (Lines 367-377): We discuss when variance under unknown concept OPEs are lower than traditional OPEs. More specifically, we discuss the covariance assumption can be used as a loss function in the algorithm section of the paper, unlike known concepts where the practitioner has to explicitly satisfy the assumption while designing the concept policies.
>
> 6.4 (Lines 390-397): We discuss how the confidence bounds loosen under unknown concepts as there is additional bias, and quantify the worst-case additional bias to be the value function $\mathbb{E}{\pi^c_e}[\hat{V}{\pi_e}]$ of the unknown concept-based estimator.
>
> Q: It’s generally known that IS estimators can exhibit high variance, and various methods have been proposed to reduce this variance at the cost of extra bias. This paper’s proposed method falls into this category, though this is not clearly acknowledged.
>
> Response: In addition to IS and PDIS, other common methods such as Weighted-IS [1], Per-Decision Weighted-IS [1], and truncating states that do not contribute to variance [2], aim to reduce variance at the cost of increased bias. While our approach reduces variance for bias, it achieves this by characterizing key state information and adhering to specific desiderata such as conciseness, diversity, and interpretability. This interpretability allows the sources of bias to be analyzed and addressed through interventions, as discussed in Section 7. Following your recommendation, we have added these examples to the related work section for completeness.
>
> [1] Eligibility traces for OPE (Precup et al.) 2000
> [2] Low Variance OPE with State-based IS (David Bossn, Philip Thomas) 2023
>
> Q: There are also choices in the paper that lack justification. For instance, it uses an approximate nearest-neighbor policy with the MIMIC-III. Having worked extensively with this dataset and reviewed recent studies on it across different tasks, I am not aware of any example where such a policy is applied.
>
> Response: For MIMIC-III, it’s very common to generate behavior trajectories based on K-nearest neighbors as the true on-policy trajectories aren’t available. Some references where behavior trajectories (or policies in general for MIMIC-III) are generated using KNNs are
>
> [1] Interpretable OPE in RL by Highlighting Influential Transitions: Here the evaluation policy is generated using 50 NNs.
>
> [2] Superhuman performance on sepsis MIMIC-III data by distributional RL.
>
> [3] Offline Policy Optimization with Eligible Actions (K=100 NNs)
>
> [4] Identification of Subgroups With Similar Benefits in OPE (K=100 NNs)
>
> [5] The AI Clinician learns optimal treatment strategies for sepsis in intensive care.
>
> [6] Development and validation of a RL algorithm to dynamically optimize mechanical ventilation in critical care.
>
> In this paper, we consider a popular variant of KNNs, called the approximate nearest neighbors search. The advantages of Approximate-NNs over KNNs are scalability, reduced computational cost, efficient indexing and support for dynamic data. This allows us to generate behavior and evaluation policies with a larger number of neighbors (200 in our paper, which is double than all known papers using KNN), with a faster inference time. Some examples of papers that use approximate nearest neighbors in medical examples are:
>
> [7] Approximate kNN Classification for Biomedical Data
>
> [8] Medical image retrieval via nearest neighbor search on pre-trained image features (This paper compares ANN as a baseline over their main algorithm called DenseLinkSearch).
>
> [9] Approximate nearest neighbors: towards removing the curse of dimensionality. (This is the seminal paper on ANNs which discusses advantages over KNNs).
>
> We add this complete analysis in Appendix G.3 in the revised version of the paper.
>
> Q: Additionally, this experiment employs a concept space of size 10^15, which does not seem to offer interpretability and appears far removed from the motivating example in Figure 1.
>
> Response: While the concept space may be higher, the interpretability lies in the concept representation and not the concept-space itself. As an example: a patient with the concept representation [0, 2, 1, 1, 2, 0, 9, 5, 2, 0, 6, 2, 1, 5, 9] shows the
> following conditions: acute kidney injury (very low creatinine), severe hypoxemia (very low PaO2), metabolic alkalosis (very high SpO2), and critical electrolyte imbalances (low potassium and magnesium), along with severe hypoglycemia. The normal GCS score indicates preserved neurological function, but over-oxygenation and potential respiratory failure are likely. The combination of anuria, AKI, and hypoglycemia points strongly toward hypotension or shock as the underlying cause. This example is listed in lines 259-264 of our paper.
>
> The design choice could be further reduced however to lower discretizations, which would reduce the space to M^15 where M is the user choice. We consider 10 to be thorough, while a lower number would be an easier condition.

---

> ### Comment · Reviewer_JqPf · 2024-11-20
>
> Thank you for your response. I want to acknowledge that I have read your rebuttal, and it has clarified some of the questions I had. However, I noticed that the paper has undergone significant changes during the rebuttal phase. Using a pdfdiff tool, I estimate that about 30–50% of the writing has been revised (it looks like a new paper!).
>
> At first glance, the changes have made the paper clearer and better written. For instance, the theorems and assumptions now have some extra space and explanations.
>
> That said, I am unsure whether such extensive revisions are typically acceptable during the rebuttal phase. Regardless, I want to inform the authors that I am carefully reviewing the revised version of the paper along with their response, but it may take some time.

---

> > ### Author Response · Authors · 2024-11-20
> >
> > Dear Reviewer JqPf,
> >
> > Thank you for your prompt response. Aside from grammatical mistakes and fluency in certain sentences, these are the main changes we have made in the paper which caters to all questions and suggestions by the reviewers. For instance,
> >
> > Section 4: Clarifies the definition of concepts and concept-based policies (This was requested by all the reviewers)
> >
> > Section 5: Theoretical section: This section was modified to connect (the assumptions and theorems) to (experiments and practical scenarios), (This was requested by Reviewer JqPf)
> >
> > Section 6: Theoretical section: This section was modified to connect (the assumptions and theorems) to (experiments and practical scenarios),  (This was requested by Reviewer JqPf). Methodology section: The explanation of the algorithm was revised to detail the nuances of each step, linking them to concepts desiderata, which underscores the significance of previously unexplored concepts. (This was requested by Reviewer 795h).
> >
> > Section 7: The methodology section of the interventions was slightly modified to better explain the definitions. We elaborated more upon human criteria, and the intervention strategies and removed some redundant definitions to improve the flow of the section. (This was requested by Reviewer n4XH).
> >
> > Appendix Section H (Ablations). We provided some additional analysis in the form of ablation experiments, like
> > a. Adding a state-abstraction baseline
> > b. IPS score comparison between concept and states
> > c. Quantitative analysis of OPE under imperfect (poor) concepts.  These ablation experiments further support the original claims in our paper and add better clarity to our original claims and observations. (These ablation experiments were requested by Reviewer 795h).
> >
> > Appendix Section G (Additional training and hyper-parameter details).
> > G2. Known, Oracle and Intervened concepts. We added some missing details on the concept policies in windygridworld. (This was requested by reviewer  795h).
> > G3. We added additional details on the background on using KNN-based policies in MIMIC-III, and justify our experiment choice. (This was requested by reviewer JqPf).
> >
> > All of these modifications are consistent with the original claims of the paper and don't change the experimental results, or theoretical proofs. These modifications further support the claims of the paper and provide additional clarity which helps the readers. Additionally,  please don't hesitate to ask for additional clarifications which help with the understanding of the paper!

---

> > > ### Comment · Reviewer_JqPf · 2024-11-26
> > >
> > > Thank you for your further clarification.
> > >
> > > The improvements in the writing are clear, specifically in the presentation of the results.
> > >
> > > The authors have also addressed most of my questions.
> > >
> > > Due to these improvements, I increase my rating.
> > >
> > > But still, I don't support the acceptance of this paper. While the paper's main claims have not changed, there are a lot of changes from the original submission. These changes are indeed in a good direction, but I don't think it's an accepted practice in the community to submit a paper with poor presentation and later fix it during rebuttal.

---

> ### Author Response · Authors · 2024-11-25
> **Possible further improvements to the paper?**
>
> Dear Reviewer JqPf,
>
> We once again thank you for your comments. We have responded to all your questions sequentially and incorporated the additional analysis in the revised version of the paper. Is there anything else you would like to see in the revised version or answer via rebuttals that can help further enhance the paper and increase the score?

---

### Official Review · Reviewer_tRrE · 2024-11-03

**Soundness:** 2
**Presentation:** 3
**Contribution:** 2
**Rating:** 5
**Confidence:** 4

**Summary:**

The goal of off-policy evaluation is to estimate the performance of an evaluation policy using data that is collected under a different behavior policy. Standard approaches to OPE can suffer from high variance. This work proposes a family of concept-based OPE estimators that reduce variance relative to standard importance-weighting estimators. They also develop an end-to-end algorithm for learning parametrized concepts, which are interpretable and can be used for off policy evaluation.

**Strengths:**

The authors study an important problem – it is well-known that off-policy evaluation estimators can suffer from high variance, so the idea of using concepts as a form of dimensionality reduction of the state-action space can yield empirical benefits.

**Weaknesses:**

The paper has some clarity / conceptual issues: The proposal of this paper is to use a concept bottleneck model to learn interpretable concepts and derive importance-weighting estimators based on these concepts. The idea of simplifying the state-action space using concepts is a promising one, but there are many technical details that are not clear from the paper.

- The authors posit that a concept at time-step $t$ can be obtained from a function $\phi$ that takes the entire trajectory from time-step $0$ to time-step $t$ as input, i.e. one can pass $(s_{t}, a_{0:t-1}, r_{0:t-1}, s_{0:t-1})$ to $\phi$. However, later in the paragraph the authors write that “In this work, we consider concepts $c_{t}$ to be just functions of current state $s_{t}$, and thus…” This is a bit confusing because $\phi$ is initially introduced as taking entire trajectories as input. Furthermore, if concepts $c_{t}$ are just functions of the current state $s_{t}$, I wonder if it would be reasonable to interpret concepts as a way of allowing us to do dimensionality reduction on the states?
- Suppose that we have a standard policy $\pi(a |s)$ and we compute a concept policy $\pi^{c}(a|c)$. The policy value function $V_{\pi}(s)$ is a function of state $s$, it is not clear to me from the paper how we can compute the analogous quantity for $V_{\pi^{c}}(s)$ or if such a quantity is even well-defined.
- Assumption 5.1 seems to require that any action-state pair that is possible under the evaluation policy must also be possible under the behavior policy. However, the written interpretation of the assumption is the  opposite.

**Questions:**

How can $\phi$ adapt to input trajectories of different length? Furthermore, in Equation (1), the authors appear to be able to invert the map $\phi$. How is this possible?

---

> ### Author Response · Authors · 2024-11-16
> **Author Response to Weaknesses 1,2**
>
> We sincerely thank the reviewer tRrE for the review and comments. We will sequentially address the questions.
>
> The paper has some clarity / conceptual issues: The authors posit that a concept at time-step t can be obtained from a function ϕ that takes the entire trajectory as input to ϕ. However, later in the paragraph the authors write that “In this work, we consider concepts ct to be just functions of current state st, and thus…” This is a bit confusing because ϕ is initially introduced as taking entire trajectories as input.
>
> Response: We acknowledge that section 4.1 could be better clarified to describe the notion of a concept function $\phi$. We make the changes in the latest version of the draft.
>
> To summarize, the concept function $\phi$ maps trajectory histories $h_t$ to concepts $c_t$. This function $\phi$ can capture various vital information in the history such as transition dynamics, short-term rewards, influential states, interdependencies in actions across timesteps, etc. Without loss of generality, in this work, we consider concepts $c_t$ to be just functions of the current state $s_t$. This assumption considers the scenario where concepts capture important information based on the criticalness of the state. Furthermore, the concept function ϕ satisfies the following desiderata: explainability, conciseness, better trajectory coverage and diversity.
> We provide this improved interpretation of a concept function in the latest draft of the paper, section 4.1. A detailed description of desiderata is provided in Appendix A.
>
> Furthermore, if concepts ct are just functions of the current state st, I wonder if it would be reasonable to interpret concepts as a way of allowing us to do dimensionality reduction on the states?
>
> Response: Certainly! One of the interpretations of concepts which are just functions of states can be viewed as dimensionality reduction, with concepts summarizing states that have similar properties like transition dynamics, short-term rewards, etc. However, it is to be noted that concepts can do much more than just dimensionality reduction. Concept-based representations allow for interventions, inspect reasons and isolate sources of variance into why a particular concept is contributing to high variance in an OPE, as discussed in the Interventions section of the paper. (Section 7).
>
> Furthermore, we conduct an ablation study (Appendix H.1, line 1799) where we perform K-means clustering over the state space. (This is our baseline for the state abstractions). First, we plot the MSE for our OPE performance against state clusters with varying numbers of clusters and then plot the clusters for the value of K(33) with minimum MSE. We observe a large number of clusters (As K=33), spread locally across, which is quite different from the true concepts. These clusters lack correspondence with the true oracle concepts or the optimized concepts, highlighting that learned concepts capture more meaningful and useful information compared to state abstractions. Furthermore, these aren’t readily interpretable and thus intervenable, underscoring the importance of performing OPE under concept representations.
>
> Q: Suppose that we have a standard policy π(a|s) and we compute a concept policy πc(a|c). The policy value function Vπ(s) is a function of state s, it is not clear to me from the paper how we can compute the analogous quantity for Vπc(s) or if such a quantity is even well-defined.
>
> Notably, the concept representation is only present in the Importance Sampling ratios, and thus in concept policy πc(a|c)  and not in the value functions $V_\pi$. This is important as we make no assumptions on concepts being Markovian, and Vπc(c) would imply there is a constraint on the concept to satisfy the Markovian property. This design choice makes the concept function \phi flexible in capturing other important aspects of the environment and offloading the burden of satisfying markovianness to traditional states. Let’s understand this through a Model-based OPE estimator to make this even clearer. Let’s take the Doubly-robust estimator as an example.
>
> $\hat{V}_{\text{DR}} $
>
> $ = \frac{1}{N} \sum_{i=1}^N \sum_{t=0}^T \prod_{k=0}^t \frac{\pi_e(a_k^{(i)} \mid s_k^{(i)})}{\pi_b(a_k^{(i)} \mid s_k^{(i)})} \left( r_t^{(i)} - \hat{Q}(s_t^{(i)}, a_t^{(i)}) \right) + \hat{V}(s_t^{(i)}) $
>
> $\hat{V}_{\text{CDR}} $
>
> $$
> = \frac{1}{N} \sum_{i=1}^N \sum_{t=0}^T \prod_{k=0}^t \frac{\pi_e(a_k^{(i)} \mid c_k^{(i)})}{\pi_b(a_k^{(i)} \mid c_k^{(i)})} \left( r_t^{(i)} - \hat{Q}(s_t^{(i)}, a_t^{(i)}) \right) + \hat{V}(s_t^{(i)})
> $$
>
> It’s important to note the concept representation only appears in the Importance Sampling ratios and not in the actual model-based estimates. The model-based estimates are still a function of states. As the concepts only appear in the IS ratios, they don’t have to be necessarily Markovian, and the Bellman equation is still satisfied as the burden of satisfying Markovianity still lies on the states.

---

> ### Author Response · Authors · 2024-11-16
> **Author Response to Weakness 3 and Question 1**
>
> Q: Assumption 5.1 seems to require that any action-state pair that is possible under the evaluation policy must also be possible under the behavior policy. However, the written interpretation of the assumption is the opposite.
>
> Response: Thanks for pointing it out! We have made the corrections in the revised draft of the paper. (Lines 210-212)
>
> Q: How can ϕ adapt to input trajectories of different lengths?
>
> Response: In this work, we consider concepts as functions of states alone, however, there are multiple ways to deal with input trajectories of different lengths. 1. We can define a window-length k beforehand, and consider k-latest (s,a,r,s’) transitions as input to the concept function $\phi$. 2. We can pad trajectories to the highest timestep, similar to word tokenization in NLP literature. On lines of implementation, an RNN or a transformer seems fit to handle input trajectories of different lengths with appropriate paddings.
>
> Q: Furthermore, in Equation (1), the authors appear to be able to invert the map ϕ. How is this possible?
>
> Response: Point taken. It is impractical to compute the inverse $\phi$ for large state spaces as defined in Equation 1, unless making additional assumptions like $\phi$ being a Bijective and invertible function, which is quite restrictive. To address this, we redefine concept-based policies as follows: (This is rewritten in section 4.1 of the paper):
> Concept-based policies $\pi^c_e, \pi^c_b$ are policies that are conditioned on concepts instead of states, where the concepts satisfy the desiderata. We make an additional assumption (5.2) on these concept-based policies. This assumption states that given a state, the concept-based policies $\pi^c_e, \pi^c_b$ are allowed to differ from the traditional policies $\pi_e,\pi_b$ by at most a quantity of $\beta$, i.e. $|\pi^c_e(a|c)-\pi_e(a|s)|<\beta$ and $|\pi^c_b(a|c)-\pi_b(a|s)|<\beta$. This is to ensure that the evaluation policy
> $π^c_e$ under concepts is reflective of the original policy $π_e$, while allowing to satisfy additional desiderata depending on the value of $\beta$. The quantity $\beta$ is defined at the discretion of the practitioner.
>
> This additional assumption allows for computational feasibility without having to evaluate the inverse of the function and instead use a soft constraint over the loss function in Line 8 of the instead. We provide all the additional details related to training in Appendix G of our paper. This new definition of the concept-policy function and the additional assumption 5.2 doesn’t change any theoretical proofs or practical experiments, as we made no assumptions on the functional form of $\phi$ in our theoretical proofs.

---

> > ### Comment · Reviewer_tRrE · 2024-11-21
> >
> > Thank you for your rebuttal and revised paper. I am reviewing the latest draft of the paper.

---

> > > ### Author Response · Authors · 2024-11-21
> > >
> > > Certainly. Please don't hesitate to ask for additional clarifications which help with the understanding of the paper!

---

> ### Author Response · Authors · 2024-11-25
> **Possible further improvements to the paper?**
>
> Dear Reviewer tRrE,
>
> We once again thank you for your comments. We have responded to all your questions sequentially and incorporated the additional analysis in the revised version of the paper. Is there anything else you would like to see in the revised version or answer via rebuttals that can help further enhance the paper and increase the score?

---

### Official Review · Reviewer_795H · 2024-11-03

**Soundness:** 3
**Presentation:** 2
**Contribution:** 2
**Rating:** 6
**Confidence:** 2

**Summary:**

The authors consider the problem of off-policy evaluation in offline/batched reinforcement learning. The goal is to be able to determine the effectiveness of a policy different from the one that collected/generated the data. The paper proposes a concept-based approach. A concept is a higher-level feature of the states/actions etc. that is (ideally) more interpretable and can capture key aspects of the problem such as transition points, changing dynamics and so on. The paper constructs a sampling importance method using concepts -- either learned concepts or concepts defined by experts. They prove various properties of the estimators and compare against traditional importance sampling methods. Finally, they run experiments on the Windy GridWorld problem and the MIMIC dataset, both when having human-designed concepts and when learning concepts.

**Strengths:**

The paper proposes a useful idea in improving the interpretability of dynamics in RL which can give insights into the problem. Section 7 was very useful in illustrating this. The computational results overall are promising and show nice improvements over existing methods.

**Weaknesses:**

Theory: Additional discussion would be helpful for the theoretical results to explain the significance of these results. What kind of insights can we gain from the theory? Anything we can apply to improve/guide practical application and experimental results?

Additionally, how does the choice on the number of concepts affect the ability of evaluate policies? More concepts may be helpful in better partitioning state  space, but overall the process becomes less interpretable if we have too many concepts.

Experiments: More detail would be appreciated on the tasks. Please explain what the WindyGridworld and MIMIC problems are, discussion on the state/action space etc. How are the concepts you propose in section 5.2 related to these? How do you assume the data is generated? For example, at least cite what the PPO algorithm is in section 5.2.

It would be nice to have more synthetic experiments, besides only WindyGridworld, so we can observe bias, variance, mean squared error and the effective sample size (ESS) on more problems (since MIMIC is not, we can only compute variance -- it is still important to observe this variance metric on real-world data, so please do keep it).

Algorithm: The algorithm introduced (Algorithm 1) seems to be a fairly significant contribution of the paper. However, it is given very little attention, and as a result I found it difficult to follow. For example, from the discussion in section 6.1 I do not see how the term $c_t^i = w \cdot f(s_t)$ is connected to the algorithm. Please provide a clearer description of the algorithm.

**Questions:**

Please see my questions in the weaknesses section above. In addition,

1. How is learning concepts different from learning representations? For example the work in [1].

2. Overall, can you give more details and intuition behind definitions etc. so we can better follow.

a) For example, in equation (1) $\phi^{-1}$ is never defined. Also initially $\phi$ is defined as a function of $a, r, s$ but $\phi^{-1}$ only acts on $c_t$. You later say that concepts are only a function of $c_t$.

b) Moreover, what is the steady-state distribution $d_{\pi}$? Steady-state of what distribution?

3. Can you provide more background on importance sampling, which seems to be the main approach you extend? It would be useful to have in order to better gauge the contribution of your work.



[1] Representation Matters: Offline Pretraining for Sequential Decision Making, Mengjiao Yang, Ofir Nachum ICML 2021

---

> ### Author Response · Authors · 2024-11-16
> **Author Response to Weaknesses 1,2**
>
> We sincerely thank the reviewer 795H for the review and comments. We will sequentially address the questions.
>
> Theory: Additional discussion would be helpful for the theoretical results to explain the significance of these results. What kind of insights can we gain from the theory? Anything we can apply to improve/guide practical application and experimental results?
>
> Response: We provide more detailed explanations to each of the Assumptions and Theorems in Sections 5 and 6, in the revised draft. These also cater to insights from theory and help design concepts and their corresponding policies practically. As a summary,
>
> 5.1 (Lines 210-213): This talks about the absolute continuity assumption, that every state-action pair that appears in the batch policy is capable of being evaluated with a positive probability.
>
> 5.2 (Lines 214-220): The state and concept policies differ by most $\beta$, to ensure that the evaluation policy $π^c_e$ under concepts is reflective of the original policy $\pi_e$.
>
> 5.3 (Lines 221-222): Known concepts are unbiased -> This is because the change of measure theorem from $\pi_b$ to $\pi^c_b$ is possible. However, this holds for infinite trajectories, and bias creeps in when there are finite trajectories.
>
> 5.4 (Lines 224-230): Variance under known concepts is lower than traditional IS, under the covariance assumption. We discuss why the covariance assumption can be realized better under concept representations as opposed to the state representations, in the revised draft of the paper.
>
> 5.5 (Lines 232-234): Variance Comparison with the MIS Estimator: We discuss how the covariance assumption under concept IS-product ratios can result in lower variance, even compared to the more efficient state-distribution ratios, which don’t suffer from the curse of the horizon with increasing trajectory length.
>
> 5.6 (Lines 236-243): Cramer-Rao bounds of our estimators: We explicitly discuss the scenario when the worst-case bounds tighten when moving from states to concepts. The crux is in the ratio $\frac{\pi_e(a|s)}{\pi_b(a|s)}$ where $\pi_b(a|s)$ is very low due to some states being poorly sampled. Under concepts, this state is better characterized which leads to an improved value of $\pi^c_b(a|c)$, which improves the denominator and reduces the IS ratio, which in turn improves the bounds. (We provide a short explanation right after stating the theorem in the main text and a detailed explanation in Appendix D.1.7.
>
> 6.1 (Lines 362-366): Unknown concepts are biased. We discuss why unknown concepts are biased, as change of measure theorem is not applicable due to unknown probability $\pi^c_b$.
>
> 6.2, 6.3 (Lines 367-377): We discuss when variance under unknown concept OPEs are lower than traditional OPEs. More specifically, we discuss the covariance assumption can be used as a loss function in the algorithm section of the paper, unlike known concepts where the practitioner has to explicitly satisfy the assumption while designing the concept policies.
>
> 6.4 (Lines 390-397): We discuss how the confidence bounds loosen under unknown concepts as there is additional bias, and quantify the worst-case additional bias to be the value function $\mathbb{E}_{\pi^c_e}[\hat{V}_{\pi_e}]$ of the unknown concept-based estimator.
>
> Additionally, how does the choice on the number of concepts affect the ability of evaluate policies? More concepts may be helpful in better partitioning state space, but overall the process becomes less interpretable if we have too many concepts.
>
> Response: The choice of the cardinality of the concepts is important for the ability to interpret and intervene the underlying evaluation policies. Larger cardinality boosts the flexibility of capturing critical state information but comes at a cost of harder interventions. Lower cardinality helps with interpretations and interventions, however makes the critical information capture tougher. Nevertheless, even on large concept spaces, it’s possible to intervene just on a subset of those concepts and extract interpretations out of it, an advantage missing in state abstractions, which may have unknown un-interpretable dependencies between different dimensions of the abstraction.

---

> ### Author Response · Authors · 2024-11-16
> **Author Response to Weaknesses 3,4,5 and Question 1**
>
> Experiments: More detail would be appreciated on the tasks. Please explain what the WindyGridworld and MIMIC problems are, discussion on the state/action space etc. How are the concepts you propose in section 5.2 related to these? How do you assume the data is generated? For example, at least cite what the PPO algorithm is in section 5.2.
>
> Response:
> Environments: We provide a detailed description of environments in Appendix F. Known Concepts: We provide elaborated details on training tasks in section 5.2. We provide details about the known Windygridworld concepts, true concepts and intervened concepts in Appendix G.2. Unknown Concepts: Training and Hyperparameter details are elaborated in Appendix G.1 Interventions: Experiment details are consumed within Sections 7.1 and 7.2. The data is generated by training an on-policy PPO algorithm for the Windygridworld environment,(we add a citation for PPO in the revised version of the paper), while the MIMIC-III dataset is publicly available with preprocessing steps elaborated in Appendix F.
>
> It would be nice to have more synthetic experiments, besides only WindyGridworld, so we can observe bias, variance, mean squared error and the effective sample size (ESS) on more problems (since MIMIC is not, we can only compute variance -- it is still important to observe this variance metric on real-world data, so please do keep it).
>
> Response: We are experimenting on an additional cancer simulator environment in the next few days, we plan to add the results in the Appendix once we are done.
>
> Algorithm: The algorithm introduced (Algorithm 1) seems to be a fairly significant contribution of the paper. However, it is given very little attention, and as a result I found it difficult to follow. For example, from the discussion in section 6.1 I do not see how the term cti=w⋅f(st) is connected to the algorithm. Please provide a clearer description of the algorithm.
>
> Response: We modify section 6.1 of the paper to better explain the nuances in the algorithm, in the revised version of the draft, lines 316-356. Additional details in comparison to the previous draft are along the lines of explanations of the loss functions concerning the concept desiderata, proximity between the concept policies and the state policies, and choice of OPE-metric (Variance) to directly optimize for the Variance as a metric.
>
> The term $c_t^i=w⋅f(s_t)$ is a design choice, which specifies the concepts are linear functions of the interpretable state features. The weights w are embedded as part of the CBM and is a specific assumption we make to automatically satisfy the interpretability desiderata. Possible alternatives to these could be hierarchical concepts or symbolic representations.
>
> How is learning concepts different from learning representations?
>
> Response: Typically all the literature so far that talks about state-abstractions are of the form of neural embeddings. These neural embeddings which are seen as alternative representations of states have limited interpretability,  and hence can’t be used for further downstream tasks. Concepts are also a form of representations, but they can be generalized to capture more important information such as transition dynamics, high variance states, short-term rewards, etc using CBMs as the base architecture. Furthermore, since concepts are compositions of human interpretable functions, they stay interpretable, and thus allow for targetted interventions (Section 7). These targeted interventions allow us to isolate sources of variance, reduce bias, and make the overall OPE estimation better. Traditional representations obtained via clustering or predefined neural embeddings lack this advantage and thus are less suitable for interventions.

---

> ### Author Response · Authors · 2024-11-16
> **Author Response to Questions 2,3**
>
> Q: Overall, can you give more details and intuition behind definitions etc. so we can better follow.
> a) For example, in equation (1) ϕ−1 is never defined. Also initially ϕ is defined as a function of a,r,s but ϕ−1 only acts on ct. You later say that concepts are only a function of ct.
>
> Response: We acknowledge that section 4.1 could be better written to describe the notion of a concept function $\phi$. We make the changes in the latest version of the draft, section 4.1.
>
> The concept function $\phi$ maps trajectory histories $h_t$ to concepts $c_t$. This function $\phi$ can capture various vital information in the trajectory history such as transition dynamics, short-term rewards, influential states, interdependencies in actions across timesteps, etc. Without loss of generality, in this work, we consider concepts $c_t$ to be functions of current state $s_t$. This assumption considers the scenario where concepts capture important information based on the criticalness of the state. Furthermore, the concept function ϕ satisfies the following desiderata: explainability, conciseness, better trajectory coverage and diversity.
> We provide this improved interpretation of a concept function in the latest draft of the paper, section 4.1. The concept function $\phi$ satisfies the following desiderata: explainability, conciseness, better trajectory coverage and diversity. A detailed description of desiderata is provided in Appendix A.
>
> b) Moreover, what is the steady-state distribution dπ? Steady-state of what distribution?
>
> Response: The steady-state distribution dπ(s) of a state s is the probability of being in that state s at any given point in time, assuming the process has reached its equilibrium. It’s mathematically defined as  $d_\pi(s') = \sum_{s \in \mathcal{S}} \sum_{a \in \mathcal{A}} d_\pi(s) \pi(a \mid s) P(s' \mid s, a)$.
>
> We acknowledge there is an easier way to define a concept-based policy, without making it as complicated as Eqn 1 (without requiring steady-state distribution functions) and rectify notation. We make the changes in the latest version of the draft, section 4.1. Concept-based policies $\pi^c_e, \pi^c_b$ are policies that are conditioned on concepts instead of states, where the concepts satisfy the desiderata. We make an additional assumption (5.2) on these concept-based policies. This assumption states that given a state, the concept-based policies $\pi^c_e, \pi^c_b$ are allowed to differ from the traditional policies $\pi_e,\pi_b$ by at most a quantity of $\beta$, i.e. $|\pi^c_e(a|c)-\pi_e(a|s)|<\beta$ and $|\pi^c_b(a|c)-\pi_b(a|s)|<\beta$. This is to ensure that the evaluation policy
> $π^c_e$ under concepts is reflective of the original policy $π_e$, while allowing to satisfy additional desiderata depending on the value of $\beta$. The quantity $\beta$ is defined at the discretion of the practitioner.
>
> Can you provide more background on importance sampling, which seems to be the main approach you extend? It would be useful to have to better gauge the contribution of your work.
>
> Response: We provide background on most Importance Sampling works in the related work section of the paper (lines 88-100), in brevity of space, we will add more background on Importance sampling in the Appendix as an additional related work section.

---

> ### Author Response · Authors · 2024-11-24
> **Possible further improvements to the paper?**
>
> Dear Reviewer 795H,
>
> We once again thank you for your comments. We have responded to all your questions sequentially and incorporated the additional analysis in the revised version of the paper. Is there anything else you would like to see in the revised version or answer via rebuttals that can help further enhance the paper and increase the score?

---

### Official Review · Reviewer_n4XH · 2024-11-08

**Soundness:** 3
**Presentation:** 3
**Contribution:** 3
**Rating:** 6
**Confidence:** 4

**Summary:**

The works addresses the high variance of methods to estimate policy value in sequential decision-making problems from offline data. It proposes to summarize the state information into interpretable concepts, e.g. using concept bottleneck models. Aggregating states into concepts better aligns supports of evaluation and batch data and reduce variance of importance sampling estimators. The work proves unbiasedness and lower variance of concept-based estimates compared to state-based estimates for common estimators. Experiments on two environments, grid world and health data from MIMIC, show good improvements both in settings of known and learnt concepts.

---
## After the rebuttal

I acknowledging reading the rebuttal and the other reviewers' comments.

I thank the authors for a detailed rebuttal. It addresses most of my comments. However, I agree with Reviewer JpPf that the revision seems major which includes changes to the definition and introduction of a new Assumption 5.2 (which seems like it should appear in the theoretical results but it does not). It warrants a closer look at the paper which is outside the scope of the rebuttal phase in my view.

That said, I like the work and think it presents an innovative approach to reducing variance of OPE. Hence, I keep the score of 6 but given the need to look closely at the substantial updates I do not feel confident to recommend acceptance strongly.

**Strengths:**

Main strengths are
- Transforming states to concepts is an interesting and refreshing idea among the many variance reduction methods. It may improve applicability of methods to safety-critical domains via improved interpretability and ability to intervene to correct estimates of policy value.
- Proposal to use state abstractions was given by Pavse & Hanna 2022a, but the use of concept bottleneck models is novel to the best of my knowledge. Authors show that concepts are more intervenable and allows a human to correct the estimates.
- Evaluation is thoroughly done and clearly visualized. Metrics such as bias, variance, MSE, and ESS are reported which are good measures in context of OPEs.

**Weaknesses:**

Main weaknesses in my view are
- Presentation could be improved at places by providing more details on how concepts can help improve OPE estimation, definitions of concept-equivalent policies, and interventions on concepts.
- Comparisons to previous work on state abstractions by Pavse & Hanna 2022a, either in experiments or in terms of technical and conceptual contributions, should be clearly made.
- Some implementation details like the OPE cost in optimization and computing concept-equivalent policy for a given policy can be described.

**Questions:**

Questions for authors:

## On presentation,

The idea behind defining a policy as in eq (1) was not clear to me. It seems like a change of measure but there could have been other ways to define a concept-equivalent policy and a clear definition is missing. Does it take the same actions as the original policy would or the same value?

Section 7 on intervention is hard to follow as the goal of the section, and terms like human criteria and qualitative intervention are not clearly described.

What are the key differences from the theoretical results in Pavse & Hanna 2022a when adapted to concept-based representation?


## On implementation,

How is the concept-equivalent policy obtained, particularly how is the inverse in eq (1) computed for infinite state space as in experiments? Relatedly, meaning of aligning a policy in line 345 was unclear.

Describe how the cost for OPE-metric is defined in line 12 of Algorithm 1. Does some data have to held out to compute the cost?

## On approach,

How is the concept-equivalent policy supposed to help evaluation - does it trade-off variance with bias or reduce variance by removing reward-irrelevant state information? What are pros and cons of this design choice?

I was expecting much more discussion in Sec 4.1 on the goals and consequences of changing representation to concepts.

Important questions remain unanswered like how does imperfect concept discovery or inability to get a concept-equivalent policy affect the final estimates.

## On evaluation,

Evaluation against state abstractions e.g. method by Pavse & Hanna 2022a seemed necessary to show that dimensionality reduction by concept bottlenecks has unique advantages, perhaps in intervention evaluations.

Similarly, does model-based methods equipped with the same domain knowledge as concept bottleneck models, say the same parameterization of feature space, perform worse?

Where does the improvement in estimation metrics come from? Was it because of collapsing states with skewed propensity scores that caused high variance? A plot of propensity scores in state and concept feature spaces might be helpful.

---

Minor comments (responses are not sought for the below):

Assumption 5.1 is explained backwards in text.

I would suggest to discuss balancing scores (https://www.jstor.org/stable/2288398), which like propensity scores are enough to get an unbiased value estimate when used to reweight rewards. Propensity scores are coarsest balancing score, however, it might be that concept-equivalent policies give another choice of balancing score.

Provide a reference for the meaning of Symbolic Representation in line 412.

Consider discussing interpretation of the four learnt concepts in MIMIC.

What is the reward in MIMIC? The term viral load was not clear and seems to refer to vitals and labs.

Authors should consider expanding the baselines and possibly the environments.

Correct typos like in line 481 cuase.

---

> ### Author Response · Authors · 2024-11-16
> **Author Response to presentation Questions 1,2**
>
> We sincerely thank the reviewer n4XH for the review and some really thought-provoking comments. We will plan to address the questions in a sequential manner.
>
> Q: The idea behind defining a policy as in eq (1) was not clear to me. It seems like a change of measure but there could have been other ways to define a concept-equivalent policy and a clear definition is missing.
>
> Response: We acknowledge there is an easier way to define a concept-based policy, without making it as complicated as Eqn 1 and rectifying notation. We make the changes in the latest version of the draft, section 4.1, lines 169-177. Concept-based policies $\pi^c_e, \pi^c_b$ are policies that are conditioned on concepts instead of states, where the concepts satisfy the desiderata. We make an additional assumption (5.2), lines 214-220 on these concept-based policies. This assumption states that given a state, the concept-based policies $\pi^c_e, \pi^c_b$ are allowed to differ from the traditional policies $\pi_e,\pi_b$ by at most a quantity of $\beta$, i.e. $|\pi^c_e(a|c)-\pi_e(a|s)|<\beta$ and $|\pi^c_b(a|c)-\pi_b(a|s)|<\beta$. This is to ensure that the evaluation policy
> $π^c_e$ under concepts is reflective of the original policy $π_e$, while allowing to satisfy additional desiderata depending on the value of $\beta$. The quantity $\beta$ is defined at the discretion of the practitioner.
>
> Q: Is it a change of measure?
> One possible interpretation of concept policies can be viewed as a change of measure, from states to concepts. More specifically, the change of measure is actually the distribution ratios under states and concepts, i.e. $\frac{d_{\pi^c_e}(c)}{d_{\pi_e}(s)}$. However, under the simpler definition we propose now, concept-policies are merely policies with actions conditioned on concepts, which satisfy the desiderata and are bounded from the original policies by $\beta$. It's possible to unify the two definitions when the distribution function under concepts can be evaluated and the concept function $\phi$ is invertible, allowing the reverse mapping from concepts to states (or concepts to trajectories in the general definition).
>
> Q: Does it take the same actions as the original policy would or the same value?
>
> Response: The action space remains the same as the original state-conditioned policies, but the policies under concepts and states differ, depending on $\beta$. For lower values of $\beta$, the concept-policies closely align with the traditional policies, with the divergence increasing with higher values of $\beta$. If the practitioner is confident in the state representations, they may set a lower $\beta$ to find concepts that align closely with state policies. Conversely, he/she may set a higher value of $\beta$ to allow for deviations and prioritize over a different objective, eg: Reducing the variance of OPE.
>
> Q: Section 7 on intervention is hard to follow as the goal of the section, and terms like human criteria and qualitative intervention are not clearly described.
>
> Response: We provide a clearer description in the revised version of the submission, section 7.1. Formal definition is elaborated in lines 440-446 while the corresponding examples to the descriptions are elaborated in lines 447-454.
> 1. We define $c^{int}_t$ as the intervention (alternative) concept a practitioner proposes at time t.
> 2. We define human criteria $h_c: (h_t, c_t) \rightarrow \{0, 1\}$ as a function constructed from domain expertise that takes in $\(h_t,c_t\)$ as input and outputs a boolean value. Basically, this function determines whether an intervention needs to be conducted over the current concept $c_t$ or not. As an example, if a practitioner has access to true on-policy values, he/she can estimate which concepts suffer from bias. If a concept doesn't suffer from bias, the human criteria $h_c(h_t, c_t)=1$ is satisfied and the concept is not intervened upon, else $h_c(h_t, c_t)=0$ and the intervened concept  c{\text{int}}_t is used instead.
> 3. (Described in Section 7.1: Qualitative concept intervention). Qualitative intervention is the scenario where the concepts are intervened with a concept defined by the practitioner. This intervention can be anything based on the domain under consideration. In the case of WindyGridworld, we consider the oracle concepts as our qualitative concept, while for MIMIC, we consider the learned C-PDIS estimator concepts as our qualitative concept while intervening on the C-IS estimator.

---

> ### Author Response · Authors · 2024-11-16
> **Author Response to presentation Question 3 and Implementation questions 1,2**
>
> What are the key differences from the theoretical results in Pavse & Hanna 2022a when adapted to concept-based representation?
>
> Response: Pavse & Hanna 2022a (referred to as PH in future) have the following 2 main theoretical results:
>
> 1. IS-based OPE estimators under state abstractions are always unbiased
> 2. Under certain covariance conditions, the Variance of the OPE estimators under state abstractions is lower than the Variance of OPE under the traditional state.
>
> Our theoretical results differ in the following way:
> We divide our studies into known and unknown concepts. (PH don't consider the possibility of unknown abstraction functions)
>
> Our first key result: IS-Estimators under known concepts are always unbiased (Theorem 5.3), whereas IS-Estimators under unknown concepts are always biased (Theorem 6.1). This is because when the concepts are unknown, a change of measure theorem to change the trajectory probabilities sampled under the behavior policy isn’t applicable (Appendix E.1, line 1274). The bias proof of PH can be seen as a special case of known concepts when the change of measure theorem is applicable and thus, is unbiased. In addition to proving unknown-concept IS-estimators are unbiased, we also learn upper bounds for them (Appendix E.1.5, E.2.4), not present in PH.
>
> Our 2nd key result: Under similar co-variance assumptions as PH over concepts, the Variance of Concept-based estimators (both known and unknown) are lower than the Variance of traditional estimators. In addition to the comparison, we also provide upper bounds on the Variance of the said estimators in (Appendix D.1.5, D.2.4, E.1.6, E.2.5), which is absent in the proofs of PH.
>
> Our 3rd key result: We make comparisons with the MIS estimator, which is the gold standard for least-variance OPE estimators, and show under specific covariance conditions, it is possible to have concept-based estimators that have lower variance than the MIS estimators. (Appendix D.1.7, D.2.7, E.1.8, E.2.7). This comparison is absent in PH.
>
> How is the concept-equivalent policy obtained, particularly how is the inverse in eq (1) computed for infinite state space as in experiments? Relatedly, the meaning of aligning a policy in line 345 was unclear.
>
> Response: In the unknown-concept scenario, alongside Concept Bottleneck Models (CBMs), the concept policies are parameterized using a neural network (NN) that maps concepts to actions. These policies are optimized using five loss functions:
> Line 5: Loss function based on what the concepts are trying to capture (Eg: In our work, transition dynamics by mapping current state to next state).
> Lines 6-7: Two loss functions based on the desiderata of the concepts: Interpretability and Diversity
> Line 8: This loss function minimizes the difference between the concept-policies and state-policies to satisfy Assumption 5.2. (Assn 5.2 is stated in the revised version of the paper)
> Line 12: The final loss function optimizes the OPE-metric variance.
> In the previous version of the paper, it was impractical to compute the inverse for large state spaces as defined in Equation 1. To address this, as we discussed earlier, we redefine concept-based policies by making a simplifying assumption that the concept policies do not deviate significantly from the state policies, now formalized as Assumption 5.2 in the revised version of the paper. This allows for computational feasibility without having to evaluate the inverse of the function and instead use a soft constraint over the loss function in Line 8. We provide all the additional details related to training in Appendix G.1 of our paper.
>
> Describe how the cost for OPE-metric is defined in line 12 of Algorithm 1.
>
> Response: The cost for the OPE-metric is defined as the OPE metric a practitioner is trying to improve. In our experiments, we consider Variance to be that metric. Alternative possibilities could be 1. Bias 2. MSE, in case the on-policy metric is known, 3. Any user-defined metric related to the evaluation depending on the practitioner and the domain.
>
> Does some data have to held out to compute the cost?
>
> Response: For conducting the unknown concept experiments, we split the batch of trajectories into 2, the first batch is used for learning the concepts (which also includes the OPE-metric cost, as it's one of our loss functions), and the other batch of data to perform evaluation.

---

> > ### Author Response · Authors · 2024-11-16
> > **Author response to Approach questions 1,2,3 and Evaluation questions 1,3**
> >
> > How is the concept-equivalent policy supposed to help evaluation - does it trade off variance with bias or reduce variance by removing reward-irrelevant state information? What are the pros and cons of this design choice?
> >
> > Response: The concept-equivalent policy helps evaluations based on the following factors.
> >
> > First, in our methodology, the learned concepts and the corresponding policies optimize for variance directly via $C_{OPE-metric}$ (Line 12 of our algorithm). Incorporating variance as a loss metric directly in the algorithm helps the concepts and the policies optimize for it and thus help evaluation.
> >
> > Second, the other loss terms in the methodology are designed to optimize for additional desiderata such as interpretability, diversity, and state-transition dynamics (as described in Algorithm line 5). These loss terms implicitly capture key properties of the states by promoting diversity in the concepts (Algorithm line 7) and accounting for state-transition dynamics (Algorithm line 5). This approach effectively eliminates redundant states that contribute to high OPE variance.
> >
> > Third, having concept-based representations allows for interventions, to inspect reasons and isolate sources of variance into why a particular concept is contributing to high variance in an OPE, as discussed in the Interventions section of the paper.
> >
> > I was expecting much more discussion in Sec 4.1 on the goals and consequences of changing representation to concepts.
> >
> > Response: We have modified section 4.1 and discussed more on the goals and consequences of using concept representations. These concept representations can capture various vital information in history, such as transition dynamics, short-term rewards, influential states, interdependencies in actions across timesteps, etc. We also provide a more elaborate description of the concept desiderata using a running medical example in Appendix A and B of our paper. We had to move this discussion to the Appendix in the brevity of space.
> >
> > Important questions remain unanswered like how does imperfect concept discovery or inability to get a concept-equivalent policy affect the final estimates.
> >
> > Response: We conduct an ablation study (Appendix H.2, line 1832) in the revised version of the paper where the concepts are imperfect. (This can happen when there are suboptimal experts in the case of known concepts and earlier stages of the training algorithm in unknown concepts). The experiment design is as follows: In WindyGridworld, the concepts are defined based on only the vertical distance to the target. This disregards important information like horizontal distance, winds, negative penalties in the regions, etc. We observe that the OPE performance is poor as compared to the original known concepts, with higher variance, higher MSE and lower ESS.
> > This further underscores the importance of the ability of our algorithm to possibly obtain concepts that are better than human-defined concepts. Additionally, as these concepts are interpretable, the cause of the high variance OPE can be tied back to these imperfect concepts and thus be intervened upon. This advantage isn’t available in traditional state abstractions.
> >
> > Evaluation against state abstractions e.g. method by Pavse & Hanna 2022a seemed necessary to show that dimensionality reduction by concept bottlenecks has unique advantages, perhaps in intervention evaluations.
> >
> > Response: We conduct an ablation study (Appendix H.1, line 1799) where we perform K-means clustering over the state space. (This is our baseline for the state abstractions). First, we plot the MSE for our OPE performance against state clusters with varying numbers of clusters and then plot the clusters for the value of K(33) with minimum MSE. We observe a large number of clusters (As K=33), spread locally across, which is quite different from the true concepts. These clusters lack correspondence with the true oracle concepts or the optimized concepts, highlighting that learned concepts capture more meaningful and
> > useful information compared to state abstractions. Furthermore, these aren’t readily interpretable and thus intervenable, underscoring the importance of performing OPE under concept representations.
> >
> > Where does the improvement in estimation metrics come from? Was it because of collapsing states with skewed propensity scores that caused high variance? A plot of propensity scores in state and concept feature spaces might be helpful.
> >
> > Response: We make a plot of the Inverse Propensity Scores (Appendix H.1, line 1799, figure on page 37) in the concept space and the traditional state space. We observe the IPS scores in the concept space are shifted towards the left as compared to IPS scores of the states, with around 2-3 orders of lowering in terms of magnitude. This indicates the improvement in the estimation metrics results from lowered IPS scores in the concept representation, reducing the overall variance.

---

> ### Author Response · Authors · 2024-11-16
> **Author response to Evaluation question 2 and minor questions**
>
> Similarly, does model-based methods equipped with the same domain knowledge as concept bottleneck models, say the same parameterization of feature space, perform worse?
>
> Response:  We list out the Doubly Robust estimator for both traditional and concept representations first.
>
> $\hat{V}_{\text{DR}} $
>
> $ = \frac{1}{N} \sum_{i=1}^N \sum_{t=0}^T \prod_{k=0}^t \frac{\pi_e(a_k^{(i)} \mid s_k^{(i)})}{\pi_b(a_k^{(i)} \mid s_k^{(i)})} \left( r_t^{(i)} - \hat{Q}(s_t^{(i)}, a_t^{(i)}) \right) + \hat{V}(s_t^{(i)}) $
>
> $\hat{V}_{\text{CDR}} $
>
> $$
> = \frac{1}{N} \sum_{i=1}^N \sum_{t=0}^T \prod_{k=0}^t \frac{\pi_e(a_k^{(i)} \mid c_k^{(i)})}{\pi_b(a_k^{(i)} \mid c_k^{(i)})} \left( r_t^{(i)} - \hat{Q}(s_t^{(i)}, a_t^{(i)}) \right) + \hat{V}(s_t^{(i)})
> $$
>
> It’s important to note the concept representation only appears in the Importance Sampling ratios and not in the actual model-based estimates. The model-based estimates are still a function of states. As the concepts only appear in the IS ratios, they don’t have to be necessarily markovian, and the bellman equation is still satisfied as the burden of satisfying markovianity still lies on the states.
> Now, let’s compare the variance of the two variants of DR estimators. Since variance comparison is preserved under addition of scalar quantities, the terms $\hat{V}(s_t^{(i)})$ and $\hat{Q}(s_t^{(i)},a_t^{(i)})$ are inconsequential, and it simplifies to comparing IS estimators under concepts and states. Under the same covariance assumption (Theorem 5.4), concept-based IS estimators have lower variance, and thus, concept-based DR estimators also have lower variance compared to traditional DR estimators.
> The overall quality of the estimation however, lies on the quality of the model-based estimates $\hat{V}(s_t^{(i)})$ and $\hat{Q}(s_t^{(i)},a_t^{(i)})$.
>
> Assumption 5.1 is explained backwards in text.
>
> Response: Thanks for pointing it out! We have made the corrections. (Line 212)
>
> Provide a reference for the meaning of Symbolic Representation.
>
> Response: We have cited a reference which uses symbolic representation using context free grammar parameterized using Differentiable programs. (Line 431)
>
> What is the reward in MIMIC? The term viral load was not clear and seems to refer to vitals and labs.
>
> Response: The term “viral load” was a typing error on our behalf, and we actually meant “vital signs”. This has been rectified in the revised version of the paper. The reward design in MIMIC is elaborated in Appendix F of the paper.

---

> > ### Author Response · Authors · 2024-11-24
> > **Possible further improvements to the paper?**
> >
> > Dear Reviewer n4XH,
> >
> > We once again thank you for your comments and thought-provoking questions. We have responded to all your questions sequentially and incorporated the additional analysis in the revised version of the paper. Is there anything else you would like to see in the revised version or answer via rebuttals that can help further enhance the paper and increase the score?

---

### Meta-Review · Area_Chair_uAz1 · 2024-12-21

**Metareview:**

The paper introduces concept-based off-policy evaluation that maps states to interpretable concepts to reduce variance in policy evaluation.

Strengths:

+ Proposes to use concept bottleneck models for variance reduction in OPE

Weaknesses:

+ Lacks some technical details and some mathematical foundations need clarification

+ Lacks strong experimental evaluation; it could benefit from more synthetic experiments and clearer baselines

+ Lacks clarity on some algorithm details, theoretical results interpretation, and overall clarity of technical concepts

**Additional Comments On Reviewer Discussion:**

The reviewers generally view this as a promising paper with good contributions, but it needs substantial improvement in presentation and experimental validation. The rebuttal addressed some concerns but introduced substantial changes that warrant further review.

---

### Decision · Program_Chairs · 2025-01-22

Reject